# SpeedCP: Fast Kernel-based Conditional Conformal Prediction

## Abstract

Conformal prediction provides distribution-free prediction sets with finite-sample conditional guarantees. We build upon the RKHS-based framework of Gibbs et al. (2023), which leverages families of covariate shifts to provide approximate conditional conformal prediction intervals, an approach with strong theoretical promise, but with prohibitive computational cost. To bridge this gap, we develop a stable and efficient algorithm that computes the full solution path of the regularized RKHS conformal optimization problem, at essentially the same cost as a single kernel quantile fit. Our path-tracing framework simultaneously tunes hyperparameters, providing smoothness control and data-adaptive calibration. To extend the method to high-dimensional settings, we further integrate our approach with low-rank latent embeddings that capture conditional validity in a data-driven latent space. Empirically, our method provides reliable conditional coverage across a variety of modern black-box predictors, improving the interval length of Gibbs et al. (2023) by 30%, while achieving a 40-fold speedup.

## 1 Introduction

Conformal prediction is a framework for constructing prediction sets that are valid under minimal distributional assumptions. Given a trained predictor $\hat{\mu}(X)$, and calibration data $(X_i, Y_i)_{i \in [n]}$ together with a test point $X_{n+1}$, all drawn i.i.d. (or more generally, exchangeable) from an unknown and arbitrary distribution $P$, conformal methods such as split conformal prediction (SplitCP) (Papadopoulos et al., 2002) calculate conformity scores $\{S_i\}_{i \in [n]}$ on the calibration data to construct a prediction set $\hat{C}(X_{n+1})$. This procedure guarantees *marginal coverage*, ensuring that the resulting set includes the true label $Y_{n+1}$ with probability at least $1 - \alpha$, for any specified $\alpha \in (0, 1)$.

However, marginal coverage does not preclude significant variability in *conditional coverage* on the test input $X_{n+1}$, defined as $\mathbb{P}(Y_{n+1} \in \hat{C}(X_{n+1}) \mid X_{n+1} = x) = 1 - \alpha$ for all $x$. This limitation can be particularly problematic in high-stakes applications such as drug discovery or socially sensitive decision-making, where systematic under-coverage on critical subgroups may lead to unreliable or even harmful outcomes. Unfortunately, prior works (Vovk, 2012; Barber et al., 2021) have shown that in distribution-free settings, any interval satisfying conditional coverage must have an infinite expected length, $\hat{C}(X_{n+1}) = \mathbb{R}$, making meaningful prediction impossible without further assumptions.

To address this issue, Gibbs et al. (2023) note that the conditional coverage can be equivalently reformulated as a marginal guarantee over any measurable function $f$, i.e., $\mathbb{E}[f(X_{n+1}) \cdot (\mathbf{1}\{Y_{n+1} \in \hat{C}(X_{n+1})\} - (1 - \alpha))] = 0$. This observation motivates them to relax the objective by restricting the requirement to a user-specified function class $\mathcal{F}$:

$$\mathbb{E}\big[f(X_{n+1}) \cdot \big(\mathbf{1}\{Y_{n+1} \in \hat{C}(X_{n+1})\} - (1 - \alpha)\big)\big] = 0, \text{ for all } f \in \mathcal{F}. \tag{1}$$

Different choices of $\mathcal{F}$ yield different notions of conditional validity. For example, taking $\mathcal{F}^0 = \{\eta : \eta \in \mathbb{R}\}$ to be the set of all constant functions in equation 1 is equivalent to guaranteeing marginal coverage. Taking $\mathcal{F}^g$ to be the set of piecewise constant functions over a set of pre-specified (potentially overlapping) groups $\mathcal{G}$, so that $\mathcal{F}^g = \{\sum_{G \in \mathcal{G}} \eta_G \mathbf{1}\{x \in G\} : \eta \in \mathbb{R}^{|\mathcal{G}|}\}$, yields group-conditional coverage (Vovk et al., 2003; Jung et al., 2022), i.e., $\mathbb{P}(Y_{n+1} \in \hat{C}(X_{n+1}) \mid X_{n+1} \in G) = 1 - \alpha$ for all $G \in \mathcal{G}$.

In this paper, we consider a more flexible class associated with a reproducing kernel Hilbert space (RKHS) that is capable of achieving coverage guarantees under *complex, nonlinear covariate shifts*:

$$\mathcal{F}^{RKHS} = \left\{ f_\psi(\cdot) + \Phi(\cdot)^\top \eta : f_\psi \in \mathcal{F}_\psi, \eta \in \mathbb{R}^d \right\} {}^1, \tag{2}$$

with a given positive definite kernel $\psi : \mathcal{X} \times \mathcal{X} \to \mathbb{R}$ and any covariate representation $\Phi : \mathcal{X} \to \mathbb{R}^d$. The linear component $\Phi(\cdot)^\top \eta$ enables marginal, group-conditional, or other linear adjustments, while the RKHS component $f_\psi(\cdot)$ controls smoothness over complex data structures. Notably, both $\mathcal{F}^0, \mathcal{F}^g$ are special cases of $\mathcal{F}^{RKHS}$. For instance, setting $f_\psi = 0$ and choosing $\Phi(X) = \mathbf{1}\{X \in G\}$ for a group $G \in \mathcal{G}$ in equation 2 recovers group-conditional coverage.

Although RKHS function classes provide a promising surrogate for exact conditional coverage in equation 1, their practical use remains limited. Gibbs et al. (2023) established theoretical guarantees under RKHS classes, but, contrary to simpler classes of shifts for which their method is fast, the computational cost of RKHS classes is so prohibitive that the approach is not deployable at scale.

To construct prediction sets, Gibbs et al. (2023) fit an RKHS quantile regression on the $n$ calibration points $(X_i, S_i)_{i \in [n]}$, augmented with the test point $(X_{n+1}, S)$, where $S$ is an imputed score for unseen value $S_{n+1}$. The imputation of $S$ is carried out via a binary search, with each candidate value requiring a fresh RKHS regression on the $n + 1$ points. Because of this already prohibitive computational burden, the authors fix the kernel bandwidth $\gamma$ and restrict hyperparameter selection to cross-validation over a pre-specified grid for the regularization parameter $\lambda$. While they demonstrate that $(\lambda, \gamma)$ do not affect marginal coverage, these hyperparameters crucially shape the smoothness of the regression fit and thus the tightness of the resulting prediction sets.

The primary objective of this paper is to improve upon the algorithm of Gibbs et al. (2023) in order to achieve conditional validity in the RKHS function class in reasonable time, guaranteeing coverage under complex covariate shifts. Like (Gibbs et al., 2023), we frame the problem as regularized RKHS quantile regression to recover score cutoffs for constructing prediction sets. To address the previous limitations, we introduce two new $(\lambda, S)$-*path* algorithms. Our method builds solution paths of regression parameters that are piecewise-linear in either the smoothness parameter $\lambda$ (the $\lambda$–path) or in the candidate score $S$ (the $S$–path). The algorithm decides the next $\lambda$ or $S$ by updating these parameters only when there is a change in the elbow set. At each step, the solution is based on the current elbow set, a subset dramatically smaller than $n+1$, yielding substantial computational savings. This formulation makes conditional conformal prediction with RKHS both tractable and tunable, providing prediction sets that are not only valid but also adaptively tight.

Our second objective is to deploy our method in high-dimensional settings when $X \in \mathbb{R}^p$ with $p \gg n$. In such cases, conditional coverage on low-rank representation is often more interpretable and relevant. Using raw covariates in kernel methods is often ineffective, as distance-based similarities become less discriminative. Accordingly, we approximate each covariate vector $X$ using a $K$-dimensional latent embedding (i.e., latent mixture, principal component, or layer embedding of a predictor network model) via a low-rank map $\hat{\pi} : \mathbb{R}^p \to \mathbb{R}^K$ with $K \ll p$. We define the kernel of the RKHS function class $\mathcal{F}^{RKHS}$ on this representation, resulting in improved signal-to-noise ratios and enhanced predictive performance (Hastie et al., 2009; Udell & Townsend, 2019). *This yields a different notion of conditional coverage: rather than directly guaranteeing* $\mathbb{P}(Y_{n+1} \in \hat{C}(X_{n+1})|X_{n+1})$, *we wish to condition on* $\mathbb{P}(Y_{n+1} \in \hat{C}(X_{n+1})|\hat{\pi}(X_{n+1}))$.

**Contributions**   Our contributions in this work are threefold:

- *Methods:* We extend conditional conformal prediction (Gibbs et al., 2023) to high-dimensional settings by conditioning on learned low-rank embeddings $\hat{\pi}(X)$ within an RKHS, and thus improving signal-to-noise and yielding better-calibrated prediction sets, particularly in low-density data regions.
- *Algorithm:* We explore the affine relationship between the imputed score $S$ and the RKHS coefficients $\upsilon, \eta$, and leverage this structure to design a fast, stable solution-path algorithm for RKHS-based conformal prediction, yielding a closed-form solution for hyperparameter selection and higher-quality prediction sets.

---

[1] Given a positive definite kernel $\psi : \mathcal{X} \times \mathcal{X} \to \mathbb{R}$, let $\mathcal{F}_\psi$ denote the associated RKHS with an inner product $\langle \cdot, \cdot \rangle_\psi$ and a norm $\| \cdot \|_\psi$. Using the representer theorem (Kimeldorf & Wahba, 1971), any function $f_\psi \in \mathcal{F}_\psi$ has a finite form $f_\psi(X) = \sum_{i \in [n+1]} \upsilon_i \psi(X, X_i)$ for some coefficient vector $\upsilon \in \mathbb{R}^{n+1}$. The norm has form $\|f_\psi\|_\psi^2 = \langle f_\psi, f_\psi \rangle_\psi = \sum_{i,j} \upsilon_i \upsilon_j \psi(X_j, X_i)$. We provide notations used in the paper in Appendix A.1.

- *Theory:* We establish finite-sample guarantees for approximate conditional coverage with respect to data-driven latent embeddings, instead of pre-specified functions of $X$, and characterize how embedding estimation error affects validity in high-dimensional inference.

We illustrate our contributions in Figure 1. **SpeedCP** achieves uniform 0.9 coverage across the 2D simplex, delivering smaller prediction sets while running nearly 50 times faster than CondCP (Gibbs et al., 2023). A detailed comparison with other conformal methods is provided in Appendix A.2, and further results are discussed in Section 3.

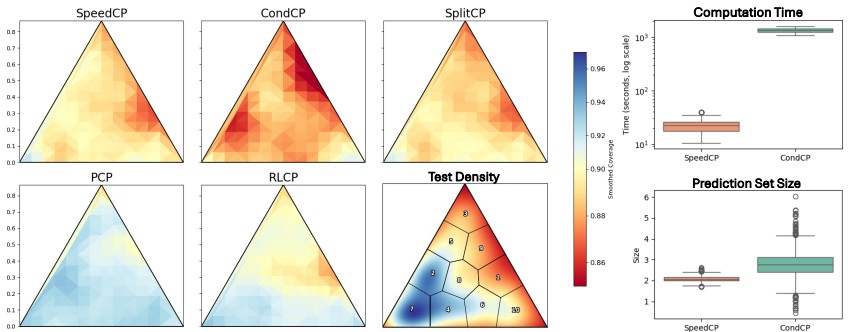

Figure 1: Mean coverage on fine-gridded partitions on the latent space (a 2D simplex). The results are aggregated over 50 random generations. **SpeedCP** shows the most uniform 0.9 (pale yellow) coverage across the simplex.

## 2 METHODS

We begin by introducing preliminary notation. We partition the dataset $\{(X_i, Y_i)\}_{i \in \mathcal{D}}$ into three disjoint subsets: $\mathcal{D}_{train}$, $\mathcal{D}_{calib}$, and $\mathcal{D}_{test}$. A single test input is denoted as $X_{n+1}$, since $Y_{n+1}$ is unobserved. The training set $\mathcal{D}_{train}$ is used to train a predictive model $\hat{\mu}(\cdot)$ while the calibration set $\mathcal{D}_{calib}$ provides conformity scores $S_i = S(X_i, Y_i)$ for $i \in \mathcal{D}_{calib}$ (We also use $i \in [n]$ to denote calibration points as $|\mathcal{D}_{calib}| = n$). For high-dimensional covariates $\mathbf{X} \in \mathbb{R}^{n \times p}$ with $p \gg n$, we denote a low-rank embedding map by $\hat{\pi} : \mathcal{X} \to \mathbb{R}^K$ with $K \ll p$. Our procedure can accommodate any low-rank embedding $\hat{\pi}(X)$, provided that $\hat{\pi}(\cdot)$ is fitted symmetrically across the calibration and test set. We provide experiments on different low-rank methods in Section 3. When dimensionality reduction is unnecessary, the identity map $\hat{\pi}(X) = X$ may be used.

Our goal is to construct prediction intervals for test points $X_{n+1}$ that achieve conditional coverage defined in equation 1 within the RKHS function class $\mathcal{F}^{RKHS}$ (equation 2). In the high-dimensional setting, we instead define the kernel on low-rank embeddings yielding a subclass $\mathcal{F}^* \subset \mathcal{F}^{RKHS}$ tailored to the latent space. The associated kernel $\psi^*$ is designed to emphasize local coverage in the latent embedding space:

$$\psi^*(X_1, X_2) = \exp\left\{-\gamma \cdot d_\pi\left(\hat{\pi}(X_1), \hat{\pi}(X_2)\right)\right\}, \tag{3}$$

where $\gamma$ is the kernel bandwidth and $d_\pi(\cdot, \cdot)$ is a distance metric between the low-dimensional embeddings (we detail this distance in Appendix B.1). The feature map $\Phi^* : \mathbb{R}^K \to \mathbb{R}^d$ is defined using the estimated embedding space generated by $\hat{\pi}(\cdot)$, thereby allowing linear modeling of covariate shifts within the latent representation space. The specific form of $\Phi^*$ depends on the application and will be specified in the theorem statements and experimental settings later.

### 2.1 ALGORITHM: SPEEDCP

In this section, we present our method for constructing conditionally valid prediction sets. We fit a regularized quantile regression in the RKHS class $\mathcal{F}^*$ with fixed $\gamma > 0$. Recalling that the rank of a test point is uniformly distributed over the calibration set plus the test point, we fit using $n$ calibration covariate-score pairs $(X_i, S_i)_{i \in [n]}$ plus the test point $(X_{n+1}, S_{n+1})$. Because $S_{n+1}$ is unobserved, we impute it with an arbitrary candidate value $S$, which yields a regression function parameterized by $S$,

$$\hat{g}_S := \arg\min_{g \in \mathcal{F}^*} \frac{1}{n+1} \sum_{i \in [n]} \ell_\alpha(S_i - g(X_i)) + \frac{1}{n+1} \ell_\alpha(S - g(X_{n+1})) + \frac{\lambda}{2} \|g_{\psi^*}\|^2_{\psi^*}, {}^2 \quad (4)$$

where $\lambda > 0$ is the regularization parameter and $\ell_\alpha(z) = (1-\alpha)[z]_+ + \alpha[z]_-$ denotes the pinball loss at level $\alpha \in (0,1)$. The regularization penalty rules out the meaningless prediction set $\hat{C}(X_{n+1}) = \mathbb{R}$ that can arise in infinite-dimensional classes. Accordingly, the prediction set takes the form,

$$\hat{C}^*(X_{n+1}) = \{y : S(X_{n+1}, y) \le \hat{g}_{S(X_{n+1}, y)}(X_{n+1})\}. \quad (5)$$

Note that the RKHS class $\mathcal{F}^*$ and the corresponding quantile regressor in equation 4 are well defined for any fixed pair $(\gamma, \lambda)$. Our method proceeds in two stages. First, we aim to select a sufficiently good pair of hyperparameters $(\gamma, \lambda)$ for the RKHS quantile regression. Using a separate data set, $\mathcal{D}_{split}$, disjoint from both the calibration and test sets, we trace the $\lambda$-path for each prefixed $\gamma$, which provides a solution path of RKHS regression along the regularization parameter $\lambda$. This yields an efficient way to explore different levels of smoothness without repeatedly solving the full optimization problem. We then cross-validate on the bandwidth $\gamma$ of the kernel $\psi^*$ to choose the optimal $(\hat{\gamma}, \hat{\lambda})$ pair. We therefore define the RKHS class $\mathcal{F}^*$ in equation 4 and in all subsequent theorems with respect to this selected pair $(\hat{\gamma}, \hat{\lambda})$. Second, integrating with the test set, we construct the $S$-path, which traces maximum score cutoff $S$ that satisfies the condition in equation 5. The full procedure is detailed in Algorithm 1. We begin by outlining the setup before introducing the $(\lambda, S)$-paths.

For a given $\lambda$, the solution to equation 4 has the following closed form:

$$\hat{g}_S(X) = \Phi^*(X)^\top \hat{\eta}_S + \frac{1}{\lambda} \sum_{i=1}^{n+1} \hat{v}_{S,i} \psi^*(X, X_i), \quad (6)$$

where $\hat{\eta}_S$, $\hat{v}_{S,i}$ are parameters when the score of the test point $S_{n+1}$ is set to $S$. For numerical stability of the algorithm, we assume the columns of $\Phi^*(X)$ are linearly independent. Plugging this in equation 4, the objective becomes,

$$\min_{\eta_S, v_S} \sum_{i=1}^{n+1} l_\alpha\Big(S_i - \Phi^*(X_i)^\top \eta_S - \frac{1}{\lambda} \sum_{i'=1}^{n+1} v_{S,i'} \psi^*(X_i, X_{i'})\Big) + \frac{1}{2\lambda} \sum_{i,i=1}^{n+1} v_{S,i} v_{S,i'} \psi^*(X_i, X_{i'}). \quad (7)$$

The Lagrangian formulation and the Karush–Kuhn–Tucker (KKT) conditions of equation 7 motivate us to define three index sets: the *Elbow*, *Left*, and *Right* set,

$$
\begin{aligned}
E &= \{i : S_i - g_S(X_i) = 0, v_{S,i} \in (-\alpha, 1-\alpha)\} \\
L &= \{i : S_i - g_S(X_i) < 0, v_{S,i} = -\alpha\} \\
R &= \{i : S_i - g_S(X_i) > 0, v_{S,i} = 1-\alpha\}.
\end{aligned} \quad (8)
$$

We observe that for the left and right sets, the kernel parameters $v_{S,i}$ are fixed to either $-\alpha$ or $1-\alpha$. Thus, we only need to solve for $v_{S,i}$'s in the elbow set, making the computation more efficient. The algorithm reduces to tracking changes in this set for different $\lambda$ or $S$ values: an *event occurs when there is a change in the index sets:* 1) a point leaves the elbow or 2) when a point from the left or right set enters it.

### 2.1.1 $\lambda$-PATH FOR SMOOTHNESS CONTROL

To select $\lambda$, we rely exclusively on the separate $m$ observations in $\mathcal{D}_{split}$ to choose optimal $\lambda$ independent of calibration and test sets. The equations 6-8 remain valid on this subset $m$, so we denote the index sets as $(E(\lambda), L(\lambda), R(\lambda))$ as the sets evolve with $\lambda$. Since no imputed score $S$ is required for $S_{n+1}$, we drop $S$ from the subscripts. We initialize $\lambda$ at the largest value for which at least two points are in the elbow, and define the step size to the next $\lambda$ as the smallest decrement that triggers an event. Importantly, the resulting coefficients $\{\hat{v}_{i'}(\lambda)\}_{i' \in [m]}$ and $\hat{\eta}(\lambda)$ evolve as a piecewise-linear function of $\lambda$, which we formalize in the following proposition.

---

[2]Let $f(\cdot) = f_{\psi^*}(\cdot) + \Phi^*(\cdot)^\top \eta \in \mathcal{F}^*$ denote the covariate-shift weighting of interest and $\hat{g}_S(\cdot) = \hat{g}_{\psi^*}(\cdot) + \Phi^*(\cdot)^\top \hat{\eta} \in \mathcal{F}^*$ be the fitted results using imputed $S$ over the same RKHS with kernel $\psi^*$ (with optimal $\hat{\gamma}$). The RKHS class is given by the optimal $\hat{\lambda}$ such that $\mathcal{F}_{\psi^*} = \{f_{\psi^*}(x) = \frac{1}{\hat{\lambda}} \sum_{i \in [n+1]} v_i \psi^*(x, X_i), v \in \mathbb{R}^{n+1}\}$.

**Proposition 1** *Let $\{\lambda^l\}_{l=1,2,3,\cdots}$ be the change points when an event occurs. For $\lambda^{l+1} \leq \lambda \leq \lambda^l$, denote $\{\hat{v}_{i'}(\lambda)\}_{i' \in [m]}$ and $\hat{\eta}(\lambda)$ as the solution of equation 7 given $\lambda$. Then, $\{\hat{v}_{i'}(\lambda)\}_{i' \in [m]}$ are affine in $\lambda$ and $\hat{\eta}(\lambda)$ is affine in $1/\lambda$.*

The piecewise linearity allows us to track the whole $\lambda$ solution path, not just at the change points. To select the optimal $(\gamma, \lambda)$-pair, we fix a grid of the kernel bandwidth values $\gamma$, and run the $\lambda$-path for each fixed $\gamma$. We then perform $k$-fold cross validation to choose the combination $(\gamma, \lambda)$ that minimizes the quantile loss. It is worth noting that our coverage guarantee, established in Section 2.2, holds for any $(\gamma, \lambda)$. However, we want to choose parameters that reflect an appropriate level of smoothness of $\hat{g}_S$, which leads to tighter prediction sets. We provide additional details on the derivation of the closed-form expressions for $\hat{v}_{i'}(\lambda)$'s and $\hat{\eta}(\lambda)$, as well as the effect of hyperparameter tuning in Appendix B.2, D.1.3, and D.1.4.

### 2.1.2 $S$-PATH FOR CONSTRUCTING PREDICTION SETS

We proceed to constructing prediction sets with $(\hat{\gamma}, \hat{\lambda})$ selected from the $\lambda$-path. We use the original notations of the regression parameters, $\hat{v}_{S,i}$ and $\hat{\eta}_S$, since conditions 4–8 now only depend on the imputed test score $S$. Recall that the prediction set is defined as a set of $y$ such that $S(X_{n+1}, y) \leq \hat{g}_{S(X_{n+1}, y)}(X_{n+1})$. By equation 8, this is equivalent to $\hat{v}_{S(X_{n+1}, y), n+1} < 1 - \alpha$. Moreover, the mapping $S \mapsto \hat{v}_S$ is nondecreasing (which we prove in Proposition 3 in Appendix C). Thus, the problem reduces to finding the largest value $S^*(X_{n+1})$ such that $\hat{v}_{S^*(X_{n+1}), n+1} < 1 - \alpha$ holds.

Conceptually, the $S$-path mirrors the $\lambda$-path: it traces the evolution of the score cutoff $S$ through a sequence of events, where events are defined identically as before. The sets in equation 8 now evolve with $S$. We initialize the $S$-path with the smallest $S^1$ such that the test point enters the elbow set (i.e., $\hat{v}_{S^1, n+1} \in (-\alpha, 1 - \alpha)$) and then increment $S$ to the next value at which an event occurs while the test point is still in the elbow. We iterate until the test point exits the elbow and set the final $S$ as $S^*(X_{n+1})$. Similar to the $\lambda$-path, we prove that $\hat{v}_{S,i}$'s and $\hat{\eta}_S$ evolve as an affine function of $S$ between any two change points:

**Proposition 2** *Let $\{S^l\}_{l=1,2,3,\cdots}$ be the change points when an event occurs. For $S^l \leq S \leq S^{l+1}$, denote $\{\hat{v}_{S,i}\}_{i \in [n+1]}$ and $\hat{\eta}_S$ as the solution of equation 7. Then, $\{\hat{v}_{S,i}\}_{i \in [n+1]}$ and $\hat{\eta}_S$ are affine in $S$.*

As shown in Appendix Lemmas 3 and 4, using the threshold $S^*(X_{n+1})$ can inflate the conditional coverage. To mitigate this, we instead prefer the randomized cutoff $S^{rand}(X_{n+1}) = \sup\{S \mid \hat{v}_{S,n+1} < U\}$, where $1 - \alpha$ is replaced by $U \sim Unif(-\alpha, 1 - \alpha)$. The final prediction set is then defined as:

$$\hat{C}^*_{rand}(X_{n+1}) = \{y : S(X_{n+1}, y) \leq S^{rand}(X_{n+1})\}. \tag{9}$$

**Computational complexity** At each iteration of the $\lambda$- and $S$-paths, we solve the inverse of $\begin{pmatrix} \mathbf{\Phi}^*_E & \frac{1}{\lambda}\mathbf{\Psi}^*_{EE} \\ \mathbf{0} & \mathbf{\Phi}^{*\top}_E \end{pmatrix}$. Here, $\mathbf{\Phi}^*_E \in \mathbb{R}^{|E| \times d}$ and $\mathbf{\Psi}^*_{EE} \in \mathbb{R}^{|E| \times |E|}$ denote submatrices with row indices and both row and column indices in the current elbow set $E$, respectively. This requires inverting a $(|E| + d) \times (|E| + d)$ matrix at each iteration. While the worst-case complexity is $O((n + d)^3)$, in practice $|E| \ll n$, making our procedure more efficient than refitting the full RKHS quantile regression at every step. We detail the initialization and update functions of the $\lambda$-, and $S$-paths as well as the proofs of Proposition 1,2 in Appendix B.2.

## 2.2 COVERAGE UNDER COVARIATE SHIFT

In our setting, covariate shift is encoded by a tilting function $f \in \mathcal{F}^*$, which reweights the original distribution $P$ to emphasize specific regions or subpopulations of the embedding space on which we seek to condition, $dP_f(x) = \frac{f(x)}{\mathbb{E}_P[f(X)]} dP(x)$. Since the solution-path formulation allows us to fit the RKHS-based quantile regression model for any pre-selected $\lambda$ and $\gamma$, we can apply Theorem 3 of Gibbs et al. (2023) to obtain a conditional guarantee with respect to all such tilts $f \in \mathcal{F}^*$ under selected $(\hat{\gamma}, \hat{\lambda})$ (as shown in Appendix C). Because $\mathcal{F}^*$ is defined in terms of an estimated low-rank projection $\hat{\pi}(\cdot)$ directly rather than the unknown true embedding of the covariates, the coverage

---

**Algorithm 1** SpeedCP

---

**Input:** $\mathcal{D}_{train}, \mathcal{D}_{split}, \mathcal{D}_{calib}, \mathcal{D}_{test}$, latent map $\hat{\pi} : \mathcal{X} \to \mathbb{R}^K, \ (K \ll p)$, kernel bandwidth grid $\Gamma$, miscoverage level $\alpha$

**Output:** Conditionally calibrated prediction set for each test point

1. Train $\hat{\mu}$ on $\mathcal{D}_{train}$ and get calibration scores: $S_i = S(X_i, Y_i), i \in \mathcal{D}_{cal\_total}$.

2. Get latent embeddings: $\hat{\pi}_{cal\_total} = \hat{\pi}(X_{cal\_total})$, $\hat{\pi}_{test} = \hat{\pi}(X_{test})$.

3. Optimize for hyperparameter pair $(\hat{\gamma}, \hat{\lambda})$ using $\mathcal{D}_{split}$ Solve $(\hat{\gamma}, \hat{\lambda}) = \arg\min_{(\gamma, \lambda)} \mathrm{CV}(\gamma, \lambda)$,

**for** $\gamma \in \Gamma$ **do** using,

    **for** $j = 1, \cdots k$ **do**

        $\{\hat{v}_1^\gamma(\lambda^l), \cdots \hat{v}_m^\gamma(\lambda^l), \hat{\eta}^\gamma(\lambda^l)\}_{l=1,2,\cdots} = \lambda\text{-}path\big((\hat{\pi}_{split \backslash fold_j}, S_{split \backslash fold_j}); \gamma\big)$

        $\hat{g}^l(X) = \Phi^*(X)^\top \hat{\eta}^\gamma(\lambda^l) + \frac{1}{\lambda^l} \sum_{i \in \mathcal{D}_{split} \backslash fold_j} \hat{v}_i^\gamma(\lambda^l) \psi^*(X, X_i)$

        $\mathrm{CV}_j(\gamma, \lambda^l) = \sum_{i \in fold_j} \big((1 - \alpha)[S_i - \hat{g}^l(X_i)]_+ + \alpha[S_i - \hat{g}^l(X_i)]_-\big)$ for $l = 1, 2, \cdots$

    **end for**

    $\mathrm{CV}(\gamma, \lambda^l) = \frac{1}{k} \sum_{j=1}^k \mathrm{CV}_j(\gamma, \lambda^l)$ for $l = 1, 2, \cdots$

**end for**

4. For each test point $X_{n+1}$, find the maximum score $S^*$ such that $S^* \leq \hat{g}_{S^*}(X_{n+1})$. Use $U \sim \mathrm{Unif}[-\alpha, 1 - \alpha]$ to get the corresponding score $S^{rand}$ for a randomized prediction set,

    **for** $X_{n+1} \in \mathcal{D}_{test}$ **do**

        $S^{rand} = S\text{-}path(X_{n+1}, \mathcal{D}_{calib}; \hat{\gamma}, \hat{\lambda}, U)$

        $\hat{C}^*_{rand}(X_{n+1}) = \{y \in \mathcal{Y} : S(X_{n+1}, y) \leq S^{rand}\}$

    **end for**

---

validity is robust to errors in $\hat{\pi}(\cdot)$. Estimation error only impacts the effectiveness of prediction set size and the deviation from conditional guarantee given the true embedding $\pi(\cdot)$ directly. We illustrate this further via the following results. To do so, we need the following assumptions:

**Assumption 1** $\{(X_i, S_i)\}_{i \in [n+1]}$ *are exchangeable and* $\{Y_i \mid X_i\}_{i \in [n+1]} \overset{i.i.d.}{\sim} P_{Y|X}$.

**Assumption 2** *The projection* $\hat{\pi}(\cdot)$ *is computed symmetrically with respect to the* $n + 1$ *inputs.*

Assumption 1 relaxes the i.i.d. condition used in Gibbs et al. (2023) to exchangeability, which is standard in conformal inference and accommodates latent-variable generative structures (e.g., admixture models such as LDA (Blei et al., 2003)) that induce dependence among $\{X_i\}$ while preserving exchangeability (see Theorem 2 for details). Assumption 2 ensures the validity of the tilt function $f$ and exchangeability of $n + 1$ samples under $P_f$.

To achieve a distribution-free guarantee for $\mathbb{P}(Y_{n+1} \in \hat{C}^*_{rand}(X_{n+1}) | \hat{\pi}(X_{n+1}))$ without overly wide intervals, we consider one standard relaxation of conditional coverage using kernel reweighting such that the tilt $f(x) := \psi^*(x, x')$ with a given fixed point $x'$, that emphasizes coverage in a neighborhood around the latent embeddings of $x'$. In this analysis, we focus purely on the RKHS component and set $\Phi^*(\cdot) \equiv 0$.

**Theorem 1** *Suppose* $\{(X_i, S_i)\}_{i \in [n+1]} \overset{i.i.d}{\sim} P$ *and Assumption 2 holds. Assume there exists a density kernel* $\psi_W^*(w, \cdot)$ *on the latent space such that, for all* $x_1, x_2 \in \mathcal{X}$, $\psi_W^*(\hat{\pi}(x_1), \hat{\pi}(x_2)) = \psi^*(x_1, x_2)$. *Let* $W' \mid X_{n+1} = x \sim \psi_W^*(\hat{\pi}(x), \cdot)$, *then we have*

$$\mathbb{P}(Y_{n+1} \in \hat{C}^*_{rand}(X_{n+1}) \mid W') = 1 - \alpha - \frac{\mathbb{E}[\sum_{i \in [n+1]} \hat{v}_{S^{rand}, i} \psi_W^*(W', \hat{\pi}(X_i))]}{\mathbb{E}[\psi_W^*(W', \hat{\pi}(X))]}. \tag{10}$$

This localized version of conformal prediction can be viewed as an approximation of conditional coverage on the event that $W' \approx \hat{\pi}(X_{n+1})$. The coverage gap on the right-hand side of equation 10 quantifies the difficulty of achieving conditional coverage in the neighborhood emphasized

by $W = w'$ after projecting on the RKHS. When $W'$ (or said $\hat{\pi}(X_{n+1})$) lies in a dense region of the embedding, this gap is small. It requires a stronger i.i.d. assumption than exchangeability in Assumption 1 in order to give more relevance to data points closer to the test point in the latent space. In other words, the coverage gap term becomes more stable. Rather than showing the gap is asymptotically zero (Guan, 2023) with strong distribution and modeling assumption, this decomposition makes the source of deviation explicit and directly estimable (see the method in Appendix C.4.2 using $n$-points quantile regression). Note, however, that equation 10 is stated for neighborhoods centered at the estimated embedding $\hat{\pi}(X_{n+1})$, not the true one. When $\hat{\pi}(\cdot)$ is a good approximation of the true embedding $\pi(\cdot)$, the guarantee in equation 10 closely matches the conditional guarantee under localization by the true latent representation, as shown in Appendix C.4.1.

In addition, the guarantee can be generalized to any finite collection of groups encoded by the feature map $\Phi^*(\cdot)$. When the covariates are generated from an underlying latent structure, such as mixture components or clusters, our approach yields tighter and more informative prediction intervals whenever the feature map captures the most informative low-rank projection of that structure. In the ideal setting where a true embedding $\pi(\cdot)$ is available, we run the quantile regression in equation 4 directly based on $\pi(\cdot)$ to obtain the conditional coverage as below.

**Theorem 2 (Oracle setting)** *Fix $K \geq 2$ and consider the latent mixture weights $\{W_i \in \Delta^{K-1}\}_{i \in [n]} \overset{i.i.d}{\sim} P_W$[3], and observations $\{X_i \mid W_i\}_{i \in [n]} \overset{i.i.d}{\sim} P_{X|W}$. Define $\pi(X) := \mathbb{E}[W \mid X] \in \Delta^{K-1}$ to be the true embedding representatives. Let $T(X) := \arg\max_{k \in [K]} \pi_k(X)$. Suppose the Assumptions 1 and 2 are both satisfied. Assume $\mathbb{P}(T(X) = k) > 0$ for any $k \in [K]$. Let $\hat{C}^*_{rand}(\cdot)$ be the randomized conformal set calibrated with the linear term $\Phi^*(X) = (\mathbf{1}\{T(X) = 1\}, \ldots, \mathbf{1}\{T(X) = K\})^\top$. Then for every $k \in [K]$,*

$$\mathbb{P}\Big(Y_{n+1} \in \hat{C}^*_{rand}(X_{n+1}) \,\Big|\, T(X_{n+1}) = k\Big) = 1 - \alpha. \tag{11}$$

Note that $\{(X_i, S_i)\}_{i \in [n]}$ in Theorem 2 are exchangeable but not independent because $\{X_i\}_{i \in [n]}$ are generated conditionally on latent variables $\{W_i\}_{i \in [n]}$. This structure violates the i.i.d. assumption on $\{(X_i, S_i)\}$ in Gibbs et al. (2023), so we need to adapt their conformal guarantee to the case with unobserved variables $W$. In practice, neither $W$ nor $\pi(\cdot)$ is observed, so we condition on the estimated representation $\hat{\pi}(\cdot)$ and its induced group $\hat{T}(X)$. The finite-sample guarantee with respect to estimated groups $\hat{T}(X_{n+1})$ hold for any low-rank projection $\hat{\pi}(\cdot)$ (see Corollary 6). In the Appendix, we further quantify how the finite-sample guarantee based on the estimated embedding $\hat{\pi}(X)$ deviates from this oracle guarantee. The accuracy of $\hat{\pi}(X)$ relative to $\pi(X)$ does not affect coverage guarantee, but affect prediction sets. In particular, a coarser embedding still maintains coverage but may yield wider and less efficient sets. Compared with the near-nominal conditional coverage given by Posterior Conformal Prediction (PCP) (Meng, 1994) which relies on a latent structure built from residuals and requires the embeddings $\{\pi(X_i)\}_{i \in [n]}$ to be highly concentrated around $\pi(X_{n+1})$, our guarantee in equation 11 is on the latent space induced by covariates directly and thus remains robust even under highly heterogeneous mixture proportions.

## 3 EXPERIMENTS

In this section, we evaluate SpeedCP across four diverse settings: synthetic admixture data, molecular property prediction with GNNs, and brain tumor MRI analysis with a CNN. We also analyze citation-count prediction on the arXiv dataset using topic-modeling features (see Appendix D.2.1).

**Synthetic experiments** We evaluate the performance of our method using synthetic datasets in the admixture setting where $X$ is generated from a mixture of $K = 3$ latent distributions. We use the mixture proportion $\hat{\pi}(X)$ as an input to all CP methods. In this case, $\sum_{k=1}^{K} \hat{\pi}_k(X) = 1$ and $\hat{\pi}_k(X) > 0$, yielding the latent space as a simplex.

---

[3]$\Delta^{K-1} = \{x \in \mathbb{R}^K : 0 \leq x_k \leq 1, \sum_{k \in [K]} x_k = 1\}$ is the $(K-1)$-dimensional simplex.

To test whether a method can effectively adapt to a covariate shift, we symmetrically sample the calibration mixture proportions over the simplex, but sample the test mixture proportions highly concentrated near one vertex (see the density plots in Figure 5). We also consider two different predictors, a linear regression and a two-layer neural network, to assess the model-agnostic behavior of the conditional conformal methods. We assess conditional coverage by dividing the simplex into 10 bins and evaluating coverage in each bin as in Figure 1. We summarize the results of SpeedCP and compare them with four other benchmarks: CondCP (Gibbs et al., 2023), SplitCP (Papadopoulos et al., 2002), PCP (Zhang & Candès, 2024), and RLCP (Hore & Barber, 2023) in Figure 2.

Table 1: Summary of parameters

| **Data** | $n$ | $p$ | $K$ | $d$ |
|---|---|---|---|---|
| Synthetic | 2K | 1K | 3 | 3 |
| Molecule | 2K | 64 | 3 | 1 |
| MRI | 2K | 256 | 8 | 2 |

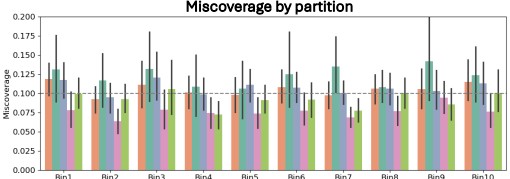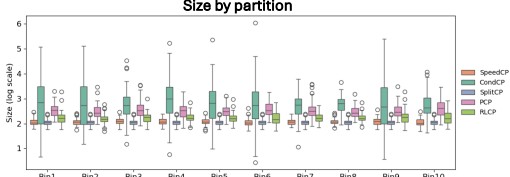

Figure 2: Conditional miscoverage and prediction set size on each fixed partition on the latent space when the predictor $\hat{\mu}$ is a linear regression. SpeedCP achieves 0.1 miscoverage across bins consistently with the smallest prediction sets.

Table 2: Marginal miscoverage, prediction set size, and computation time.

| Method | **Marginal miscoverage** ($\alpha = 0.1$) | | **Prediction set size** | **Time (seconds)** |
|---|---|---|---|---|
| | LR | NN | NN | LR |
| **SpeedCP** | **0.105** $\pm$ **0.07** | **0.098** $\pm$ **0.02** | **0.804** $\pm$ **0.06** | 22.05 $\pm$ 6.22 |
| CondCP | 0.123 $\pm$ 0.13 | 0.124 $\pm$ 0.05 | 2.126 $\pm$ 0.31 | 1332.67 $\pm$ 129.93 |
| SplitCP | 0.107 $\pm$ 0.07 | 0.114 $\pm$ 0.02 | **0.780** $\pm$ **0.06** | < 0.01 |
| PCP | 0.076 $\pm$ 0.06 | 0.088 $\pm$ 0.02 | 0.910 $\pm$ 0.13 | 141.64 $\pm$ 14.48 |
| RLCP | 0.092 $\pm$ 0.07 | 0.089 $\pm$ 0.02 | 0.864 $\pm$ 0.07 | 22.05 $\pm$ 0.07 |

Overall, SpeedCP achieves miscoverage closest to the target level of 0.1 while producing the smallest prediction sets. SplitCP attains near-target miscoverage in many bins because a single global threshold works reasonably well when the calibration points are approximately uniform over the simplex. However, near the vertex (bins 3 and 5. See also Figure 4 in Appendix), sparsity makes the global threshold governed by denser regions, weakening its ability to capture local scores under covariate shift. CondCP solves the same regularized quantile regression problem, but it fails to achieve reasonable coverage in several bins and produces overly wide intervals. This happens because the optimization solver it relies on does not return exact solutions and output conservative approximations. In contrast, our path algorithm uses a stable piecewise-linear structure of the problem and tracks boundary events precisely, yielding tighter and more accurate prediction sets. Both PCP and RLCP tend to overcover in most bins and produce large prediction sets, as their performance is sensitive to the quality of the base predictor $\hat{\mu}(\cdot)$. We provide additional details of the experimental design, the full NN results, and comparisons across different values of $n$ in Appendix D.1.

**Molecule Graphs** We evaluate our method on three molecular property prediction benchmarks: QM9, QM7b, and ESOL (Wu et al., 2018). For each dataset, we train a GNN to predict a molecular property: the HOMO–LUMO gap for QM9, polarizability for QM7b, and solubility for ESOL. We extract the last 64-dimensional graph embedding after pooling, and reduce it to 3 dimensions via PCA. Our objective is to achieve nominal 0.9 coverage across this low-dimensional representation of the molecular graphs. To assess conditional coverage, we partition the PC space into 6–8 regions using Voronoi tessellation, and compute coverage within each region. We aggregate results over 50 random subsamples of 2000 graphs, and report the results in Figure 3 and Table 3. We observe that SpeedCP achieves nominal coverage consistently across all partitions, while achieving sharp prediction sets. We provide additional results for each dataset in Appendix D.2.

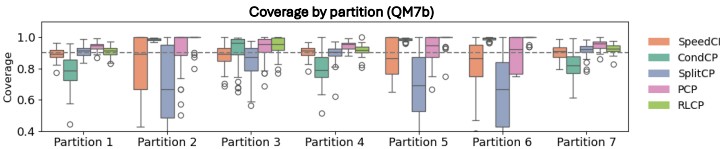

Figure 3: Coverage on fixed partitions of the PC space for QM7b. We use PCA on the last layer embeddings of GNN with $K = 3$ dimensions. The dashed line denotes the target coverage rate $1 - \alpha = 0.9$.

Table 3: Mean prediction set size and computation time of QM9, QM7b, and ESOL

| Method | Prediction set size | | | Computation time (seconds) | | |
|---|---|---|---|---|---|---|
| | QM9 | QM7b | ESOL | QM9 | QM7b | ESOL |
| **SpeedCP** | 1.135 ±0.25 | **0.902 ±0.44** | **1.789 ±0.38** | 31.061 ±2.94 | 33.056 ±7.23 | 15.442 ±1.55 |
| CondCP | 1.922 ±0.40 | 1.447 ±1.17 | 2.683 ±0.42 | 1531.15 ±195.60 | 1890.38 ±166.62 | 625.06 ±64.54 |
| SplitCP | **1.122 ±0.122** | 0.999 ±0.37 | 1.800±0.17 | < 0.01 | < 0.01 | < 0.01 |
| PCP | 1.530 ±0.87 | 1.303 ±1.07 | 2.261 ±1.00 | 38.018 ±3.48 | 47.218 ±6.50 | 21.659 ±2.36 |
| RLCP | 1.554 ±0.89 | 1.286 ±1.04 | 2.248±1.02 | 1.157 ±0.02 | 1.148 ±0.01 | 0.668 ±0.00 |

**Brain Tumor MRI** We evaluate on a brain–tumor MRI dataset from Kaggle[4] with labels $\{healthy, tumor\}$. We train a CNN classifier $\hat{\mu}(\cdot)$ on 2,000 images and extract NN features from the last layer for calibration (training details in Appendix D.2.3). Table 4 shows that even with intercept-only calibration ($\Phi^*(X) = 1$), our RKHS component alone gives a good approximation for predicted-label coverage. When covariate shift aligns with label groups, adding linear terms for the predicted label, $\Phi^*(X) = (\mathbf{1}\{\hat{\mu}(X) = healthy\}, \mathbf{1}\{\hat{\mu}(X) = tumor\})^\top$, provides better conditional coverage. In contrast, SplitCP achieves comparable coverage but requires more conservative sets than ours, while RLCP fails to exploit locality in the 256-dimensional feature space and effectively reduces to uniform weighting, thus converging to Split CP. PCP tends to overcover, especially for the healthy group, and their cutoffs are unstable with high variance and frequent near-zero values (see Appendix Table 7), thereby producing overly conservative conditional coverage.

Table 4: Mean coverage and prediction set size across predicted labels in the MRI dataset.

| Method | Target coverage ($1 - \alpha = 0.9$) | | | Prediction set size | | | Time (seconds) |
|---|---|---|---|---|---|---|---|
| | Marginal | Healthy | Tumor | Marginal | Healthy | Tumor | |
| **SpeedCP(1)**[5] | 0.910 ±0.01 | **0.902 ±0.02** | 0.914 ±0.02 | 0.262 ±0.09 | **0.250 ±0.09** | 0.275 ±0.08 | 244.1 ±9.2 |
| **SpeedCP($\Phi^*$)** | 0.908 ±0.02 | **0.902 ±0.02** | **0.901 ±0.02** | 0.282 ±0.08 | 0.266 ±0.08 | 0.295 ±0.08 | 270.5 ±13.9 |
| SplitCP | **0.898 ±0.01** | 0.888 ±0.02 | 0.903 ±0.02 | 0.348 ±0.00 | 0.348 ±0.00 | 0.348 ±0.00 | < 0.01 |
| PCP | 0.918 ±0.01 | 0.945 ±0.02 | 0.902 ±0.02 | **0.231 ±0.27** | 0.281 ±0.26 | **0.201 ±0.28** | 162.1 ± 13.9 |
| RLCP | **0.898 ±0.01** | 0.888 ±0.02 | 0.903 ±0.02 | 0.348 ±0.00 | 0.348 ±0.00 | 0.348 ±0.00 | 3.48± 0.08 |

## 4 LIMITATIONS AND FUTURE DIRECTIONS

While we believe our algorithm can be broadly applicable in high-dimensional problems, especially when prior knowledge is limited, we highlight several limitations and directions for future work: (1) We currently fix the miscoverage level $\alpha$ for all test points. However, $\alpha$ could be made adaptive based on latent structure or user-specified utility. For example, one might use a stricter $\alpha$ for subpopulations deemed more critical (Takeuchi et al., 2006; Gauthier et al., 2025), thereby allocating tighter guarantees where they matter most. (2) Incorporating weights into our quantile regression based on uncertainty or embeddings' importance could further refine coverage and interpretability (Jang & Candès, 2023). Although we focus on scalar regression tasks, the RKHS-based framework can be extended to structured prediction problems such as text generation (Sun et al., 2023; Farquhar et al., 2024; Su et al., 2024; Shahrokhi et al., 2025), image completion (Angelopoulos et al., 2020; Wieslander et al., 2020), and other multivariate problems (Messoudi et al., 2021; Johnstone & Ndiaye, 2022; Xu et al., 2024) where uncertainty quantification over complex outputs is crucial.

---

[4]https://www.kaggle.com/datasets/murtozalikhon/brain-tumor-multimodal-image-ct-and-mri

[5]For the Brain Tumor MRI data, we use **SpeedCP($\Phi^*$)** to denote calibration with a linear term that includes predicted labels, whereas **SpeedCP(1)** uses an intercept-only linear term with $\Phi^*(X) = 1$.

**Ethics Statement.**   This work adheres to the ICLR Code of Ethics. Our research does not involve human subjects, sensitive personal data, or applications with foreseeable risks of misuse. The datasets employed are publicly available and widely used in prior work. We have carefully considered issues of fairness, privacy, and security, and do not anticipate any ethical concerns arising from our methodology or findings.

**Reproducibility Statement.**   We have taken significant steps to ensure the reproducibility of our results. All theoretical results are stated with clear assumptions and complete proofs provided in the appendix. The experimental setup, including data preprocessing procedures, hyperparameter choices, and evaluation metrics, is described in detail in the main text and appendix. Anonymized code and instructions to reproduce all experiments will be made available in the supplementary material. Together, these resources allow independent researchers to fully reproduce and validate our findings.

**Use of Large Language Models (LLMs).**   In preparing this work, we used large language models (LLMs) only as general-purpose assistive tools. Specifically, LLMs were employed to help with tasks such as grammar correction, polishing the clarity of exposition, rephrasing sentences for readability, adjusting mathematical notation for consistency, and correcting minor issues in code formatting. All research ideas, methodological contributions, theoretical results, and experimental designs were conceived and executed by the authors. We carefully verified all LLM-assisted text and code to ensure correctness and originality, and we take full responsibility for the content of this paper. LLMs were not used for generating research insights, proofs, experiments, or results, and therefore are not considered contributors or authors.

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

## A  NOTATION AND RELATED WORKS

### A.1  NOTATION

For any set $\mathcal{G}$, let $|\mathcal{G}|$ denote its cardinality. Given a vector $\eta \in \mathbb{R}^p$, we use $\eta(i)$ or $\eta_i$ to represent the $i$-th entry. For any $n \in \mathbb{N}$, let $[n]$ denote the index set $\{1, \ldots, n\}$. Throughout this paper, we denote the sets of variables with simple bold letters (e.g. $\mathbf{X} \in \mathbb{R}^{n \times p} = (X_1, X_2, \ldots, X_n)^\top$). Let capital letter $P$ denote the joint distribution and $P_X$ denote the marginal distribution of $X$.

Given a value $z$, let $[z]_+ = \max(z, 0)$ and $[z]_- = \max(-z, 0)$. Let $\mathcal{P}_{\mathcal{B}_n}, \mathcal{P}_{\mathcal{B}_\infty} : \mathbb{R}^K \to \mathbb{R}^K$ denote the projection operators onto sets $\mathcal{B}_n, \mathcal{B}_\infty$, respectively. We use $Q_{1-\alpha}$ to denote the empirical $1 - \alpha$ quantile of the conformal scores.

Let $a_n$ and $b_n$ be sequences of real-valued random variables or deterministic quantities indexed by $n \in \mathbb{N}$. We use the following asymptotic notation:$a_n = O(b_n)$ means there exists a constant $c > 0$ such that $|a_n| \le c|b_n|$ for all sufficiently large $n$. $a_n = O_{\mathbb{P}}(b_n)$ means that for any $\epsilon > 0$, there exists $c_\epsilon > 0$ and $N_\epsilon \in \mathbb{N}$ such that $\mathbb{P}(|a_n| > c_\epsilon |b_n|) < \epsilon$, for all $n \ge N_\epsilon$. We use small $c$ to represent a constant, which may vary line by line.

### A.2  RELATED WORKS ON CONFORMAL PREDICTION

In standard split conformal prediction, the data is partitioned into three sets: the training set which is used to train a predictive model $\hat{\mu}(\cdot)$, the calibration set $\{X_i, Y_i\}_{i \in [n]}$ which is used to calibrate conformity scores, and finally, the test point $X_{n+1}$ with unknown response $Y_{n+1}$. Throughout this paper, we work with split conformal prediction, which generates the prediction interval for $Y_{n+1}$ as:

$$\hat{C}(X_{n+1}) = \{y : S(X_{n+1}, y) \le q^*\}, \tag{12}$$

where $q^*$ is chosen as the $(1 - \alpha)$-quantile of the set $\{S_i\}_{i \in [n+1]}$. The resulting prediction set contains all values $y$ for which the conformity score $S(X_{n+1}, y)$ is sufficiently small.

We demonstrate below how the various coverage can be achieved depending on the information available about the predictive model $\hat{\mu}(\cdot)$.

**Marginal coverage**  Suppose we know that the predictive model performs equally well across the entire feature space, and the $(n + 1)$-th conformity score is drawn i.i.d. from the same distribution as the first $n$ scores. By the replacement lemma in Angelopoulos et al. (2024), the prediction set in equation 12 can be obtained by the threshold $q^0 = Q_{1-\alpha}(\sum_{i \in [n]} \frac{1}{n+1} \delta_{S_i} + \frac{1}{n+1} \delta_{+\infty})$. It is well known that the set $\hat{C}^0(X_{n+1})$ given by $q^0$ has marginal validity such that $\mathbb{P}(Y_{n+1} \in \hat{C}^0(X_{n+1})) \ge 1 - \alpha$ (Papadopoulos et al., 2002). As an alternative strategy, Gibbs et al. (2023) proposed obtaining coverage threshold $q^0$ in equation 12 using an intercept-only quantile regression within the constant function class $\mathcal{F}^0$. Let $S$ denote an imputed value for the unknown score $S_{n+1}$ and define the pinball loss for level $\alpha$ as $\ell_\alpha(z) = (1 - \alpha)[z]_+ + \alpha[z]_-$. Then they fit

$$\hat{q}_S^0 := \arg\min_{q \in \mathcal{F}^0} \frac{1}{n+1} \sum_{i \in [n]} \ell_\alpha(S_i - q) + \frac{1}{n+1} \ell_\alpha(S - q), \tag{13}$$

and output the nonrandomized prediction set $\hat{C}^0(X_{n+1}) = \{y : S(X_{n+1}, y) \le \hat{q}_{S(X_{n+1}, y)}^0\}$. They show that this procedure also satisfies the marginal validity guarantee.

Applying conformal prediction in settings with latent structure is nontrivial. There exist several challenges for conformal prediction with low-rank structure: (1) misspecification of $\hat{\mu}(\cdot)$ may prevent the latent structure of $X$ from being faithfully reflected in the distribution of $S \mid X$; (2) if the embedding $\hat{\pi}(\cdot)$ is inaccurate or incomplete so that there are few neighbors near the test point in the embedding space, prediction intervals can become overly conservative or excessively wide; and (3) an inappropriate choice of rank $K$ may undermine the conditional validity.

One prominent approach is Posterior Conformal Prediction (PCP) (Zhang & Candès, 2024), which has been detailed as follows.

**Posterior conformal prediction**  Zhang et al. Zhang & Candès (2024) proposed a posterior conformal prediction (PCP) framework under the assumption that $X$ exhibits a latent low-rank structure,

and the predictive model $\hat{\mu}(\cdot)$ is well-specified. Specifically, they assume the conditional distribution of the conformity score $S \mid X$ follows a mixture model:

$$S_i \mid X_i \sim \sum_{k \in [K]} \pi_k(X_i)\zeta_k,$$

where $\zeta_1, \ldots, \zeta_K$ are distinct probability densities, and $\pi_k(X_i)$ represent cluster membership probabilities. Adapting ideas from weighted conformal prediction, the prediction set is constructed as:

$$\hat{C}^{\text{PCP}}(X_{n+1}) = \left\{ y : S(X_{n+1}, y) \leq Q_{1-\alpha}\left( \sum_{i \in [n]} w_i \delta_{S_i} + w_{n+1}\delta_{+\infty} \right) \right\}.$$

where weights $\{w_i\}_{\in [n+1]}$ are determined by the similarity between latent structures. Let $m\hat{\pi} \sim$ Multinomial$(m, \pi(X_{n+1}))$. In the randomized setting, the weights $w_{i,rand}$ are proportional to $\exp\left\{ -\sum_{k=1}^{K} m\hat{\pi}_k \cdot \log \frac{\pi_k(X_{n+1})}{\pi_k(X_i)} \right\}$. In the nonrandomized setting, weights are proportional to $\exp\left\{ -mD_{\text{KL}}\left(\pi(X_{n+1}) \,\|\, \pi(X_i)\right) \right\}$. Under the randomized setting, Zhang & Candès (2024) show that PCP provides conservative conditional coverage guarantees.

$$1 - \alpha \leq \mathbb{P}\left( Y_{n+1} \in \hat{C}^{\text{PCP}}_{rand}(X_{n+1}) \mid \hat{\pi} \right) \leq 1 - \alpha + \mathbb{E}\left[ \max_{i \in [n+1]} w_{i,rand} \mid \hat{\pi} \right]. \tag{14}$$

This approach relies on the assumption that the predictive model $\hat{\mu}(\cdot)$ is well-specified, so that the latent structure of $\mathbf{X}$ can be faithfully reflected in the mixture structure of the conditional distribution of the scores given $X$. When $\hat{\mu}(\cdot)$ is inaccurate, the scores $S$ can exhibit higher variability, and the distribution of $S \mid X$ may not display a meaningful latent structure.

Instead of assuming latent structure in the noise model $S \mid X$, we directly leverage latent embeddings in the covariates $\mathbf{X}$. By calibrating conformity scores as a function of $\hat{\pi}(X)$ within an RKHS, rather than assuming their relationships a priori, our method remains robust under model misspecification and provides reliable uncertainty quantification.

**Localized conformal prediction** Another related method is randomly-localized conformal prediction (RLCP) Hore & Barber (2023), which aims to capture heterogeneity in the conformity score by adjusting the distribution based on proximity to the test point $X_{n+1}$. Specifically, LCP assigns higher weights, instead of $1/(n + 1)$ for $q^0$ in marginal coverage, to data points closer to the test point $X_{n+1}$. These weights on $\delta_{S_i}$, for instance, are proportional to the kernel distance $\exp(-\gamma \|X_i - X_{n+1}\|^2)$ for a bandwidth parameter $\gamma > 0$. While Hore & Barber (2023) showed LCP achieves marginal validity under a randomization step, increasing the bandwidth parameter $\gamma$ can significantly widen the prediction interval, especially in high-dimensional settings.

To do the low-rank projection, RLCP applies a Gaussian reweighting to conformity scores based on distances in a latent embedding space between the test point and calibration points. This approach relies on carefully chosen embeddings that maximize the mutual information between conformity scores and covariates. When either $\hat{\mu}(\cdot)$ or $\hat{\pi}(\cdot)$ is inaccurate or incomplete so that there are few neighbors near the test point in the embedding space, RLCP often produces overly conservative or excessively wide prediction intervals by increasing $\gamma$.

In contrast, our method uses $\lambda$-path adapted to the local calibration density, allowing greater flexibility in sparse regions. This selects $(\gamma, \lambda)$ to leverage the global low-rank structure and produce more stable, calibrated prediction intervals (See Figure 1).

**Conditional conformal** Suppose no prior information is available about the covariate shift, unlike the settings discussed in LCP and PCP. In this general setting, let $\psi : \mathcal{X} \times \mathcal{X} \to \mathbb{R}$ be a positive definite kernel, and let $\mathcal{F}_\psi$ denote the associated RKHS with an inner product $\langle \cdot, \cdot \rangle_\psi$ and a norm $\| \cdot \|_\psi$. Gibbs et al. (2023) proposed the regularized kernel quantile regression for class $\mathcal{F}^{RKHS}$ in equation 2 with a fixed hyperparameter $\lambda > 0$:

$$\hat{g}_S^{CC} := \arg\min_{g \in \mathcal{F}^{RKHS}} \frac{1}{n+1} \sum_{i \in [n]} \ell_\alpha(S_i - g(X_i)) + \frac{1}{n+1}\ell_\alpha(S - g(X_{n+1})) + \lambda\|g_\psi\|_\psi^2. \tag{15}$$

They constructed the nonrandomized prediction set as $\hat{C}^{cc}(X_{n+1}) := \{y : S(X_{n+1}, y) \leq \hat{g}_{S(X_{n+1},y)}^{CC}(X_{n+1})\}$

**Lemma 3 (Theorem 3 in Gibbs et al. (2023))** *Let $\psi : \mathcal{X} \times \mathcal{X} \to \mathbb{R}$ be a positive definite kernel, and $\Phi : \mathcal{X} \to \mathbb{R}^d$ a finite dimensional feature map. Consider the RKHS-based function class $\mathcal{F}^{RKHS}$ associated with $\psi$ and $\Phi$. Assume that $\{(X_i, S_i)\}_{i \in [n+1]}$ are exchangeable. Then for all $f \in \mathcal{F}^{RKHS}$, we have*

$$\mathbb{E}\left[ f(X_{n+1}) \cdot \left( \mathbf{1}\{Y_{n+1} \in \hat{C}^{CC}(X_{n+1})\} - (1 - \alpha) \right) \right] = -2\lambda \mathbb{E}\left[ \langle \hat{g}^{CC}_{S_{n+1}, \psi}, f_\psi \rangle \right] + |\epsilon_{int}|,$$

*where the interpolation error $\epsilon_{int}$ satisfies $|\epsilon_{int}| \leq \mathbb{E}\left[ f(X_i) \mathbf{1}\{S_i = \hat{g}^{CC}_{S_{n+1}}(X_i)\} \right]$.*

The interpolation term $\epsilon_{int}$ can be removed when randomized prediction sets are used (see Lemma 4).

Similar to the challenges faced in localized conformal prediction, solving the optimization problem equation 4 using a kernel $\psi$ defined over the original high-dimensional feature space can lead to oversmoothing and wider prediction interval. In particular, when $p \gg n$ the RKHS norm $\|g_\psi\|_\psi$ becomes large unless regularization $\lambda$ is increased significantly, which in turn flattens the estimated quantile function $\hat{g}_S(\cdot)$ As a result, the prediction set may have poor local adaptivity, leading to wider intervals and coverage gaps.

# B COMPUTATIONAL DETAILS FOR SPEEDCP

## B.1 LOW-RANK PROJECTION USING ADMIXTURE MODEL

In this work, we consider high-dimensional covariates $X \in \mathbb{R}^p$ with $p \gg n$ and denote its low-rank representation map as $\hat{\pi} : \mathcal{X} \to \mathbb{R}^K$ with $K \ll p$. A simple choice of $\hat{\pi}(\cdot)$ is principal component analysis (PCA), where $\hat{\pi}(X) = X^\top \mathbf{V}$, with $K$ principal directions $\mathbf{V} \in \mathbb{P}^{p \times K}$. Alternatively, probabilistic models such as latent Dirichlet allocation (LDA) (Blei et al., 2003) provide interpretable embeddings, representing each $X$ as a mixture of latent components $\{\zeta_k\}_{k \in [K]}$. In deep learning models, one can also consider applying low-rank projections on layer embeddings. For the simulation experiments and experiments with ArXiv abstracts, we consider the admixture model under the probabilistic Latent Semantic Indexing (pLSI)(Hofmann, 1999),

$$mX_i \mid W_i = w_i \sim \text{Multinomial}(m, \sum\nolimits_{k \in [K]} w_i(k)\zeta_k) \tag{16}$$

where $W_i \in \Delta^{K-1}$ denotes the latent mixture proportions and $\zeta_k$ represents the latent distribution. $m$ denotes the document length. This shows $\mathbb{E}[X_i \mid W_i] = \boldsymbol{\zeta}^\top W_i$. However, this decomposition in general may not be unique, but under the separability condition Donoho & Stodden (2003) or anchor word condition Arora et al. (2012), $\boldsymbol{\zeta}$ is identifiable.

When applying RKHS methods to compositional data such as mixture proportions $\hat{\pi}(X)$, it is essential to first transform the simplex into Euclidean space. If we perform kernel regression or smoothing over $\hat{\pi}$ directly, the output might be outside the simplex. Suppose $\hat{\pi}(X_i)$ lies in the open simplex such that all entries are positive, then the log-ratio transformation (such as additive, centered, and isometric log-ratio transformations)(Aitchison, 1982) can be used.

**Centered log-ratio transformation (clr)** If $\hat{\pi}_k(X_i) > 0$ for all $i, k$,

$$\hat{\theta}_{ik} := \log \hat{\pi}_k(X_i) - \frac{1}{K} \sum_{j \in [K]} \log \hat{\pi}_j(X_i)$$

Given this transformation, we define the kernel similarity between points as:

$$d_\pi(X_i, X_j) := \|\hat{\theta}_i - \hat{\theta}_j\|^2; \qquad \psi^*(X_i, X_j) = \exp\left\{ -\gamma \|\hat{\theta}_i - \hat{\theta}_j\|^2 \right\}$$

**pLSI using SVD** Let $\mathbf{X} := \mathbf{X}_{train} \cup \mathbf{X}_{calib} \cup \mathbf{X}_{test} \in \mathbb{R}^{n_{all} \times p}$. Here, we present one of the algorithms used to estimate the latent embeddings $\boldsymbol{\pi} := \boldsymbol{\pi}(\mathbf{X}) = \mathbb{E}[\mathbf{W} \mid \mathbf{X}]$ from $\mathbf{X}$. When $m \to \infty$, the posterior mean $\mathbb{E}[W_i \mid X_i]$ concentrates around the true mixture proportion $w_i$.

Assume $\boldsymbol{\pi}$ and $\boldsymbol{\zeta}$ are full-rank matrices and the $K$-th largest singular value satisfies $\lambda_K(\boldsymbol{\pi}\boldsymbol{\zeta}^\top) > 0$, we start with the singular value decomposition of matrix $\boldsymbol{\pi}\boldsymbol{\zeta}^\top$:

$$\boldsymbol{\pi}\boldsymbol{\zeta}^\top = \boldsymbol{\Xi}\boldsymbol{\Lambda}\mathbf{V}^\top \implies \boldsymbol{\Xi} = \boldsymbol{\pi}\boldsymbol{\zeta}^\top\mathbf{V}\boldsymbol{\Lambda}^{-1} := \boldsymbol{\pi}\mathbf{H}$$

with some nonsingular matrix $H$. Notice that each row of $\boldsymbol{\pi} \in \mathbb{R}^{n_{all} \times K}$ is a probability vector (i.e., nonnegative and sums to 1). Given this simplex structure, we can recover the matrix $H$ from $\boldsymbol{\Xi}$ using nonnegative matrix factorization techniques. In particular, methods such as the *Successive Projection Algorithm* (SPA) Araújo et al. (2001); Gillis & Vavasis (2013) and *Archetypal Analysis* Javadi & Montanari (2020) are effective in recovering the extreme points (vertices) of the convex hull.

---

**Algorithm 2** pLSI using SVD Klopp et al. (2021)

---

**Input:** $\mathbf{X} \in \mathbb{R}^{n_{all} \times p}$, latent dimension $K$

**Output:** $\widehat{\boldsymbol{\pi}}_{train}, \widehat{\boldsymbol{\pi}}_{calib}, \widehat{\boldsymbol{\pi}}_{test} = \hat{\pi}(\mathbf{X}, K)$

1. Get the rank-$K$ SVD of $\mathbf{X} = \hat{\boldsymbol{\Xi}}\hat{\boldsymbol{\Lambda}}\hat{\mathbf{V}}^\top$

2. (Vertex hunting algorithm) Apply the vertex hunting algorithm on the rows of $\hat{\boldsymbol{\Xi}}$ to get the vertices $\hat{\mathbf{H}}$

3. Set $\hat{\pi}(\mathbf{X}) = \hat{\boldsymbol{\Xi}}\hat{\mathbf{H}}^{-1}$ and thus $\hat{\pi}(X_i) = (\hat{\mathbf{H}}^{-1})^\top \Xi_i$.

---

### B.2 DERIVATION OF $\lambda$-PATH AND $S$-PATH

In this section, we provide technical details on our path-tracing approaches of $\lambda$ and $S$. Our approach for $\lambda$-path is inspired by the work of Li et al. (2007), who derives the solution path of $\lambda$ in a RKHS quantile regression setting. Similar approaches have been studied extensively for the lasso Tibshirani (1996; 2011), generalized linear models Friedman et al. (2010), and quantile regression Koenker (2005); Li et al. (2007). In our work, we build on the solution path algorithm for RKHS quantile regression developed by Li et al. (2007) and adapt it to our RKHS function class $\mathcal{F}^*$, which has an extra linear component $\Phi^*(X)^\top \eta_S$,

$$\mathcal{F}^* = \left\{ f_{\psi^*}(\cdot) + \Phi^*(\cdot)^\top \eta : f_{\psi^*} \in \mathcal{F}_{\psi^*}, \eta \in \mathbb{R}^d \right\}. \tag{17}$$

We begin with some preliminaries.

Denote $S_i = S(X_i, Y_i)$ as the score of the $i^{th}$ point in the calibration set for $i \in [n]$ and $S_{n+1}$ as the score of a test point. To decide the score cutoff we use for a prediction set, we proceed to fit a RKHS quantile regression on $n$ calibration points together with the test point. Since the true score of the test point, $S_{n+1}$ is unknown, we set the score of the test point, $S_{n+1}$, as an arbitrary value $S$. Let $\alpha \in (0, 1)$ be a user-specified miscoverage level. The objective then becomes,

$$\hat{g}_S = \arg\min_{g \in \mathcal{F}^*} \frac{1}{n+1} \sum_{i \in [n]} \ell_\alpha(S_i - g(X_i)) + \frac{1}{n+1} \ell_\alpha(S - g(X_{n+1})) + \frac{\lambda}{2} \|g_{\psi^*}\|_{\psi^*}^2, \tag{18}$$

with the known solution in finite form:

$$\hat{g}_S(X) = \Phi^*(X)^\top \hat{\eta}_S + \frac{1}{\lambda} \sum_{i=1}^{n+1} \hat{v}_{S,i} \psi^*(X, X_i), \tag{19}$$

We define $\Phi^*(X) \in \mathbb{R}^d$ as any feature representation of $X$ and $\eta_{S,j}$ as the coefficient of $\Phi^*(X)_j$, $j \in [d]$. Plugging this in, the objective becomes,

$$\min_{\eta_S, v_S} \sum_{i=1}^{n+1} l_\alpha \left( S_i - \Phi^*(X_i)^\top \eta_S - \frac{1}{\lambda} \sum_{i'=1}^{n+1} v_{S,i'} \psi^*(X_i, X_j) \right) + \frac{1}{2\lambda} \sum_{i,i'=1}^{n+1} v_{S,i} v_{S,i'} \psi^*(X_i, X_{i'}).$$

with the Lagrangian primal function as

$$L_p = (1 - \alpha) \sum_{i=1}^{n+1} p_i + \alpha \sum_{i=1}^{n+1} q_i + \frac{1}{2\lambda} \upsilon_S^\top \boldsymbol{\Psi}^* \upsilon_S$$

$$+ \sum_{i=1}^{n+1} \sigma_i(S_i - g_S(X_i) - p_i) - \sum_{i=1}^{n+1} \tau_i(S_i - g_S(X_i) + q_i) \tag{20}$$

$$- \sum_{i=1}^{n+1} \kappa_i p_i - \sum_{i=1}^{n+1} \rho_i q_i,$$

and $\sigma, \tau, \kappa, \rho$ are nonnegative Lagrangian multipliers. Here, $\Psi^* \in \mathbb{R}^{(n+1) \times (n+1)}$ denotes the kernel matrix where its $(i, i')$ element denotes $\psi^*(X_i, X_{i'})$. Setting the derivatives of $L_p$ at 0,

$$\frac{\partial L_p}{\partial \upsilon_{S,i}} : \upsilon_{S,i} = \sigma_i - \tau_i$$

$$\frac{\partial L_p}{\partial \eta_{S,j}} : \sum_{i=1}^{n+1} \sigma_i \Phi^*(X_i)_j = \sum_{i=1}^{n+1} \tau_i \Phi^*(X_i)_j, \;\; j \in [d]$$

$$\frac{\partial L_p}{\partial p_i} : \sigma_i = 1 - \alpha - \kappa_i \tag{21}$$

$$\frac{\partial L_p}{\partial q_i} : \tau_i = \alpha - \rho_i.$$

The Karush–Kuhn–Tucker (KKT) conditions give

$$\sigma_i(S_i - g_S(X_i) - p_i) = 0$$
$$\tau_i(S_i - g_S(X_i) + q_i) = 0$$
$$\kappa_i p_i = 0 \tag{22}$$
$$\rho_i q_i = 0$$

Since Lagrangian multipliers are nonnegative, $0 \leq \sigma_i \leq 1 - \alpha$ and $0 \leq \tau_i \leq \alpha$, combining equation 21 and equation 22, we can easily see that,

$$S_i - g_S(X_i) > 0 \;\; \Rightarrow \;\; p_i > 0, \; \kappa_i = 0, \; \sigma_i = 1 - \alpha, \; \tau_i = 0 \;\; \Rightarrow \;\; \upsilon_{S,i} = 1 - \alpha$$
$$S_i - g_S(X_i) < 0 \;\; \Rightarrow \;\; q_i > 0, \; \rho_i = 0, \; \tau_i = \alpha, \; \sigma_i = 0 \;\; \Rightarrow \;\; \upsilon_{S,i} = -\alpha \tag{23}$$
$$S_i - g_S(X_i) = 0 \;\; \Rightarrow \;\; p_i = q_i = 0, \; \sigma_i \in (0, 1 - \alpha], \; \tau_i \in (0, \alpha] \;\; \Rightarrow \;\; \upsilon_{S,i} \in (-\alpha, 1 - \alpha)$$

With $\widehat{r}_{S,i} := S_i - \hat{g}_S(X_i)$, the KKT conditions induce three index sets:

$$E := \left\{ i : \; \widehat{r}_{S,i} = 0, \; \hat{\upsilon}_{S,i} \in (-\alpha, 1 - \alpha) \right\}, \tag{24}$$

$$L := \left\{ i : \; \widehat{r}_{S,i} < 0, \; \hat{\upsilon}_{S,i} = -\alpha \right\}, \tag{25}$$

$$R := \left\{ i : \; \widehat{r}_{S,i} > 0, \; \hat{\upsilon}_{S,i} = 1 - \alpha \right\}. \tag{26}$$

## B.3 Derivation of $\lambda$-path

We use $\lambda$-path to tune the regularization (or smoothness) parameter $\lambda$, which we combine with cross validation on the kernel bandwidth $\gamma$ to determine the optimal hyperparameter pair. The same equation 18-equation 24 hold, but the RKHS quantile regression is now estimated with m separate points. The motivation for this is to fix the hyperparameters before constructing prediction sets, which is necessary for our theoretical guarantees. The index sets $(E, L, R)$ evolve with different $\lambda$ values. We denote them as $(E(\lambda), L(\lambda), R(\lambda))$.

We start with a sufficiently large initial value $\lambda^1$ and decrease it toward 0. As $\lambda$ decreases, data points move from the left of the elbow, stay in the elbow, then move to the right of the elbow (or vice versa). Any change in the elbow set is denoted as an "event". The next $\lambda$ is updated as the largest value where such event occurs. At each update, we calculate $\hat{\upsilon}_i$ for the points in $E(\lambda)$ since $\{\hat{\upsilon}_{i'}\}_{i' \in [m]}$ in $L(\lambda), R(\lambda)$ are fixed.

In this section, we let the columns of $m$ points projection $\boldsymbol{\Phi}^* \in \mathbb{R}^{m \times d}$ are linearly independent. Denote $\boldsymbol{\Phi}_A^*$ as a submatrix of $\boldsymbol{\Phi}^*$ whose row indices are in set $A$. Also denote $\boldsymbol{\Psi}_{AB}^*$ as a submatrix of $\boldsymbol{\Psi}^* \in \mathbb{R}^{m \times m}$ whose row indices are in set $A$ and column indices are in set $B$.

### B.3.1 PROOF OF PROPOSITION 1

We now prove Proposition 1, which states affine relationship of $\hat{v}_{i'}(\lambda)$'s and $\hat{\eta}(\lambda)$ on $\lambda$ between two change points of $\lambda$. If $\hat{v}_{i'}(\lambda)$'s and $\hat{\eta}(\lambda)$ are affine in $\lambda$ between any change points, then it holds that they are piecewise-linear on $\lambda$, which makes the solution path tractable for any $\lambda \leq \lambda^1$. We provide a more detailed version of the proof in Section B.4.1, which has identical steps as Proposition 1.

**Proof.** Let $\{\lambda^l\}_{l=1,2,3,\cdots}$ be the change points when an event occurs. Consider an interval $\lambda^{l+1} \leq \lambda \leq \lambda^l$ during which the sets stay the same, i.e., $(E(\lambda), L(\lambda), R(\lambda)) = (E(\lambda^l), L(\lambda^l), R(\lambda^l))$. Denote $\hat{v}_{i'}(\lambda)$ and $\hat{\eta}(\lambda)$ as the solution of equation 18 given $\lambda$. In this proof, denote $E = E(\lambda) = E(\lambda^l)$, $L = L(\lambda) = L(\lambda^l)$, and $R = R(\lambda) = R(\lambda^l)$. Define two quantities.

$$d_E := \frac{1}{\lambda}((-\alpha)\,\boldsymbol{\Psi}_{EL}^* \mathbf{1}_L + (1-\alpha)\,\boldsymbol{\Psi}_{ER}^* \mathbf{1}_R), \qquad \boldsymbol{\Pi}_E := I_{|E|} - \boldsymbol{\Phi}_E^*\big(\boldsymbol{\Phi}_E^{*\top}\boldsymbol{\Phi}_E^*\big)^{-1}\boldsymbol{\Phi}_E^{*\top}.$$

Let $S_E := (S_{i'})_{i' \in E}, d_E := (d_{i'})_{i' \in E}, \boldsymbol{\Phi}_E^* \in \mathbb{R}^{|E| \times p}, \boldsymbol{\Psi}_{EE}^* \in \mathbb{R}^{|E| \times |E|}$. By the definition of the elbow set combined with equation 19,

$$S_E = \boldsymbol{\Phi}_E^* \hat{\eta}(\lambda) + \frac{1}{\lambda}\,\boldsymbol{\Psi}_{EE}^*\,\hat{v}_E(\lambda) + d_E. \tag{27}$$

Projecting with $\boldsymbol{\Pi}_E$ eliminates $\hat{\eta}(\lambda)$,

$$\boldsymbol{\Pi}_E \boldsymbol{\Psi}_{EE}^* \hat{v}_E(\lambda) \; = \; \lambda\,\boldsymbol{\Pi}_E(S_E - d_E). \tag{28}$$

Moreover, the second KKT constraint in equation 21 gives $\boldsymbol{\Phi}^{*\top}\hat{v} = 0$. This is equivalent to,

$$\boldsymbol{\Phi}_E^{*\top}\hat{v}_E(\lambda) = \alpha\boldsymbol{\Phi}_L^{*\top}\mathbf{1}_L - (1-\alpha)\boldsymbol{\Phi}_R^{*\top}\mathbf{1}_R.$$

Define $\mathbf{A} := \boldsymbol{\Pi}_E \boldsymbol{\Psi}_{EE}^* \boldsymbol{\Pi}_E$. Using its Moore–Penrose inverse (denoted by superscript $\dagger$),

$$\begin{aligned}
\boldsymbol{\Pi}_E \hat{v}_E(\lambda) = \; & \lambda\mathbf{A}^\dagger\,\boldsymbol{\Pi}_E\big(S_E - d_E\big) \\
& - \alpha\,\mathbf{A}^\dagger\boldsymbol{\Pi}_E\boldsymbol{\Psi}_{EE}^*\boldsymbol{\Phi}_E^*(\boldsymbol{\Phi}_E^{*\top}\boldsymbol{\Phi}_E^*)^\dagger\boldsymbol{\Phi}_L^{*\top}\mathbf{1}_L \\
& + (1-\alpha)\,\mathbf{A}^\dagger\boldsymbol{\Pi}_E\boldsymbol{\Psi}_{EE}^*\boldsymbol{\Phi}_E^*(\boldsymbol{\Phi}_E^{*\top}\boldsymbol{\Phi}_E^*)^\dagger\boldsymbol{\Phi}_R^{*\top}\mathbf{1}_R.
\end{aligned} \tag{29}$$

Thus, the minimum–norm solution on $\mathrm{Im}(\boldsymbol{\Pi}_E)$ is,

$$\begin{aligned}
\hat{v}_E(\lambda) = \; & \lambda\mathbf{A}^\dagger\boldsymbol{\Pi}_E\big(S_E - d_E\big) \\
& + [I_{|E|} - \mathbf{A}^\dagger\boldsymbol{\Pi}_E\boldsymbol{\Psi}_{EE}^*]\boldsymbol{\Phi}_E^*\big(\boldsymbol{\Phi}_E^{*\top}\boldsymbol{\Phi}_E^*\big)^\dagger\big[\alpha\,\boldsymbol{\Phi}_L^{*\top}\mathbf{1}_L - (1-\alpha)\,\boldsymbol{\Phi}_R^{*\top}\mathbf{1}_R\big].
\end{aligned} \tag{30}$$

Thus, $\hat{v}_E(\lambda)$ is *affine in $\lambda$* on the interval. From equation 27,

$$\hat{\eta}(\lambda) = (\boldsymbol{\Phi}_E^{*\top}\boldsymbol{\Phi}_E^*)^{-1}\boldsymbol{\Phi}_E^{*\top}\Big[S_E - d_E - \frac{1}{\lambda}\,\boldsymbol{\Psi}_{EE}^*\,\hat{v}_E(\lambda)\Big], \tag{31}$$

hence $\hat{\eta}(\lambda)$ is *affine in $1/\lambda$*. We have shown that,

$$\hat{v}_E(\lambda) = a + \lambda a, \qquad \hat{\eta}(\lambda) = a^{(1)} + \frac{a^{(1)}}{\lambda}. \tag{32}$$

with $a, b \in \mathbb{R}^{|E|}, a^{(1)}, b^{(1)} \in \mathbb{R}^d$ constant on the segment. For $i \in L(\lambda), R(\lambda)$, $\hat{v}_{i'}$ is constant, making it affine in $\lambda$ as well. Finally, for any $i' \in [m]$,

$$\begin{aligned}
\hat{g}(X_{i'}) &= \boldsymbol{\Phi}_{i'}^*\big(a^{(1)} + \frac{a^{(1)}}{\lambda}\big) + \frac{1}{\lambda}\boldsymbol{\Psi}_{i',E}^*(a + \lambda a) + d_{i'} \\
&= \frac{1}{\lambda}(\boldsymbol{\Phi}_{i'}^* a^{(1)} + \boldsymbol{\Psi}_{i',E}^* a) + \boldsymbol{\Phi}_{i'}^* a^{(1)} + \boldsymbol{\Psi}_{i',E}^* a + d_{i'},
\end{aligned} \tag{33}$$

which makes the residual $r_{i'}(\lambda) = S_{i'} - \hat{g}(X_{i'})$ affine again in $1/\lambda$ for $i' \in [m]$ on the interval. $\blacksquare$

### B.3.2 UPDATE OF $\lambda^l$

Let $\lambda^l$ denote the value after the $l^{th}$ event. The elbow set $E(\lambda^l)$ is updated when one of the following events occurs,

- A point $i$ in either $L(\lambda^l)$ or $R(\lambda^l)$ enters the elbow set (residual $S_i - \hat{g}(X_i)$ becomes 0).
- A point $i$ in $E(\lambda^l)$ leaves to the left or right set ($\hat{v}_i(\lambda), i \in E(\lambda^l)$ becomes $-\alpha$ or $1 - \alpha$).

We take $\lambda^{l+1}$ as the largest $\lambda \leq \lambda^l$ that triggers one of the events and update $(E, L, R)$ accordingly. Here, let $E = E(\lambda) = E(\lambda^l)$. Denote the linear parameter $\hat{\eta}_j^\lambda = \lambda \hat{\eta}_j(\lambda)$ for $j \in [d]$. From equation 19, for $\lambda^{l+1} \leq \lambda \leq \lambda^l$, the fit at $\lambda$ is,

$$\hat{g}(X_{i'}) = \boldsymbol{\Phi}_{i'.}^* \hat{\eta}(\lambda) + \frac{1}{\lambda} \boldsymbol{\Psi}_{i',.}^* \hat{v}(\lambda)$$

$$= \boldsymbol{\Phi}_{i'.}^* \hat{\eta}(\lambda) + \frac{1}{\lambda} \left( \boldsymbol{\Psi}_{i',E}^* \hat{v}_E(\lambda) - \alpha \boldsymbol{\Psi}_{i',L}^* \mathbf{1}_L + (1-\alpha) \boldsymbol{\Psi}_{i',R}^* \mathbf{1}_R \right)$$

$$= \frac{1}{\lambda} \left( \boldsymbol{\Phi}_{i'.}^* \hat{\eta}^\lambda + \boldsymbol{\Psi}_{i',E}^* \hat{v}_E(\lambda) + d_{i'} \right),$$

where

$$d_{i'} := -\alpha \boldsymbol{\Psi}_{i',L}^* \mathbf{1}_L + (1-\alpha) \boldsymbol{\Psi}_{i',R}^* \mathbf{1}_R.$$

Let $\hat{g}^l(X_{i'})$ be the estimated function with $\lambda^l$. Now, we can express $\hat{g}(X_{i'})$ with $\lambda^l$ and $\hat{g}^l(X_{i'})$,

$$\hat{g}(X_{i'}) = \hat{g}(X_{i'}) - \frac{\lambda^l}{\lambda} \hat{g}^l(X_{i'}) + \frac{\lambda^l}{\lambda} \hat{g}^l(X_{i'})$$

$$= \frac{1}{\lambda} \left[ \boldsymbol{\Phi}_{i'.}^* (\hat{\eta}^\lambda - \hat{\eta}^{\lambda^l}) + \boldsymbol{\Psi}_{i',E}^* (\hat{v}(\lambda) - \hat{v}^l(\lambda)) + d_{i'} - d_{i'} + \lambda^l \hat{g}^l(X_{i'}) \right] \qquad (34)$$

$$= \frac{1}{\lambda} \left[ \boldsymbol{\Phi}_{i'.}^* (\hat{\eta}^\lambda - \hat{\eta}^{\lambda^l}) + \boldsymbol{\Psi}_{i',E}^* (\hat{v}(\lambda) - \hat{v}^l(\lambda)) + \lambda^l \hat{g}^l(X_{i'}) \right].$$

Recall from the second KKT condition equation 21, we have $v_{i'} = \sigma_{i'} - \tau_{i'}$ and $\sum_{i'=1}^m (\sigma_{i'} - \tau_i) \boldsymbol{\Phi}_{i',j}^* = \sum_{i'=1}^m v_{i'} \boldsymbol{\Phi}_{i',j}^* = 0$ for $j = 1, \cdots, d$.

Component-wise,

$$\boldsymbol{\Phi}_E^{*\top} \hat{v}_E(\lambda) - \alpha \boldsymbol{\Phi}_L^{*\top} \mathbf{1}_L + (1-\alpha) \boldsymbol{\Phi}_R^{*\top} \mathbf{1}_R = 0,$$

and

$$\boldsymbol{\Phi}_E^{*\top} \hat{v}_E(\lambda^l) - \alpha \boldsymbol{\Phi}_L^{*\top} \mathbf{1}_L + (1-\alpha) \boldsymbol{\Phi}_R^{*\top} \mathbf{1}_R = 0,$$

leading to

$$\boldsymbol{\Phi}_E^{*\top} (\hat{v}_E(\lambda) - \hat{v}_E(\lambda^l)) = 0. \qquad (35)$$

Denote $\bar{v}_{i'} = \hat{v}_{i'}(\lambda) - \hat{v}_{i'}(\lambda^l)$ for $i' \in E$ and $\bar{\eta}_j = \hat{\eta}_j^\lambda - \hat{\eta}_j^{\lambda^l}$ for $j \in [d]$. For any $i \in E^l$, $\hat{g}(X_i) = S_i$. Let $S_E$ be the stacked scores for $E$. Then, equation 34 becomes,

$$\boldsymbol{\Phi}_E^* \bar{\eta} + \boldsymbol{\Psi}_{EE}^* \bar{v} = (\lambda - \lambda^l) S_E$$

Combining with equation 35 and representing in a matrix form,

$$\begin{pmatrix} \boldsymbol{\Phi}_E^* & \boldsymbol{\Psi}_{EE}^* \\ \mathbf{0} & \boldsymbol{\Phi}_E^{*\top} \end{pmatrix} \begin{pmatrix} \bar{\eta} \\ \bar{v} \end{pmatrix} = (\lambda - \lambda^l) \begin{pmatrix} S_E \\ \mathbf{0} \end{pmatrix}$$

$$\mathbf{A}^l \beta = (\lambda - \lambda^l) \mathbf{S}_0$$

$$a = (\mathbf{A}^l)^{-1} \mathbf{S}_0,$$

where $a = \beta/(\lambda - \lambda^l)$. Let $a_u = \bar{\eta}/(\lambda - \lambda^l)$ and $a_v = \bar{v}/(\lambda - \lambda^l)$. Plugging $a_u, a_v$ back to equation 34, we reexpress the estimated function as a function of $a$,

$$\hat{g}(X_{i'}) = \frac{\lambda^l}{\lambda}[\hat{g}^l(X_{i'}) - h^l(X_{i'})] + h^l(X_{i'})$$

where

$$h^l(X_{i'}) = \mathbf{\Phi}^*_{i'.} a_u + \mathbf{\Psi}^*_{i', E} a_v$$

for $i' \in E$. Finally, to decide $\lambda^{l+1}$, we choose which event (whether a point enters or exits the elbow set). The first event will happen for $\lambda$ such that a point in $L(\lambda^l)$ or $R(\lambda^l)$ set satisfies $\hat{g}(X_{i'}) = S_{i'}$, leading to,

$$\lambda^{l+1,hit} = \max_{i' \in L(\lambda^l), R(\lambda^l)} \lambda^l \frac{\hat{g}^l(X_{i'}) - h^l(X_{i'})}{S_{i'} - h^l(X_{i'})} \mathbf{1}\left\{ \frac{\hat{g}^l(X_{i'}) - h^l(X_{i'})}{S_{i'} - h^l(X_{i'})} \leq 1 \right\}.$$

Here, the indicator is to ensure that the updated $\lambda$ is smaller than $\lambda^l$ so that the path is monotonically decreasing. To find $\lambda$ such that a point leaves $E(\lambda^l)$,

$$\lambda^{l+1,leave} = \lambda^l + \max_{i' \in E(\lambda^l)} \left\{ x \in \left\{ \frac{-\alpha - \hat{v}_{i'}(\lambda^l)}{a_{v,i'}}, \frac{1 - \alpha - \hat{v}_{i'}(\lambda^l)}{a_{v,i'}} \right\} \mid x \leq 0 \right\}$$

We then take $\lambda^{l+1} = \max\left\{\lambda^{l+1,hit}, \lambda^{l+1,leave}\right\}$. We also update $(E, L, R) = (E(\lambda^{l+1}), L(\lambda^{l+1}), R(\lambda^{l+1}))$ accordingly based on which event occurred. Finally, parameters $\hat{v}_{i'}(\lambda^{l+1})$'s, $\hat{\eta}(\lambda^{l+1})$ can be updated by solving for the new elbow,

$$\begin{pmatrix} \frac{1}{\lambda}\mathbf{\Psi}^\star_{EE} & \mathbf{\Phi}^*_E \\ \mathbf{\Phi}^{*\top}_E & 0 \end{pmatrix} \begin{pmatrix} v \\ \eta \end{pmatrix} = \begin{pmatrix} S_E - \frac{1}{\lambda}(-\alpha\mathbf{\Psi}^\star_{EL}\mathbf{1}_L + (1-\alpha)\mathbf{\Psi}^\star_{ER}\mathbf{1}_R) \\ \alpha\mathbf{\Phi}^{*\top}_L\mathbf{1}_L - (1-\alpha)\mathbf{\Phi}^{*\top}_R\mathbf{1}_R \end{pmatrix} \tag{36}$$

### B.3.3 INITIALIZATION OF $\lambda$

We describe our strategy for selecting a sufficiently large initial value $\lambda^1$. At $\lambda^0 = \infty$, from equation 19, we can see that $\hat{g}(X_{i'}) = \mathbf{\Phi}^*_{i',.}\hat{\eta}$. In this case, we have only one point in the elbow, which we denote as $i^0$, that satisfies $S_{i^0} = \hat{g}(X_{i^0}) = \mathbf{\Phi}^*_{i^0.}\hat{\eta}$. We choose $i^0$ as the $(1-\alpha)$th quantile of scores, i.e. $S_{i^0} = S_{\lceil(m)(1-\alpha)\rceil}$. Then, points that satisfy $S_{i'} < S_{i^0}$ are in $L(\lambda^0)$, and points such that $S_{i'} > S_{i^0}$ are in $R(\lambda^0)$.

To make the parameters identifiable, we set $\hat{\eta}_{j^*}(\lambda^0) = S_{i^0}/\mathbf{\Phi}^*_{i^0,j^*}$ for one $j^* \in [d]$ and set other parameters $\hat{\eta}_j(\lambda^0)$, $j \neq j^*$ to 0. When $\mathbf{\Phi}^*_{i^0.}$ is one-hot encoded, $j^*$ is any index such that $\mathbf{\Phi}^*_{i^0,j^*} = 1$. If $\mathbf{\Phi}^*_{i^0.}$ is continuous, we choose $j^*$ to be any arbitrary index. From equation 21, we have the condition $\sum_{i'=1}^m \hat{v}_{i'}\mathbf{\Phi}^*_{i',j} = 0$ for $j \in [d]$. Since $i^0$ is the only point in $E(\lambda^0)$. This leads to,

$$\hat{v}_{i^0}(\lambda^0) = \frac{\alpha \sum_{i \in L(\lambda^0), R(\lambda^0)} \mathbf{\Phi}^*_{i,j^*} - \sum_{i \in R(\lambda^0)} \mathbf{\Phi}^*_{i,j^*}}{\mathbf{\Phi}^*_{i^0,j^*}} \tag{37}$$

Next, we find the next $\lambda^1$, which will be the initial value of our solution path. This will be the largest $\lambda < \infty$ such that another point from either $L(\lambda^0), R(\lambda^0)$ enters the elbow. Let $i^1$ be the new point entering the elbow. Then, $i^1$ satisfies,

$$S_{i^1} = \mathbf{\Phi}^*_{i^1,j^*}\hat{\eta}_{j^*}(\lambda^0) + \frac{1}{\lambda^1}\left( \mathbf{\Psi}^*_{i^1,i^0}\hat{v}_{i^0}(\lambda^0) - \alpha\mathbf{\Psi}^*_{i^1,L(\lambda^0)}\mathbf{1}_{L(\lambda^0)} + (1-\alpha)\mathbf{\Psi}^*_{i^1,R(\lambda^0)}\mathbf{1}_{R(\lambda^0)} \right)$$

$$= \mathbf{\Phi}^*_{i^1,j^*}\hat{\eta}_{j^*}(\lambda^0) + \frac{1}{\lambda^1}f(X_{i^1})$$

Since $i^0$ is still in the elbow set, it should also satisfy,

$$S_{i^0} = \mathbf{\Phi}^*_{i^0,j^*}\hat{\eta}_{j^*}(\lambda^0) + \frac{1}{\lambda^1}\left( \mathbf{\Psi}^*_{i^0,i^0}\hat{v}_{i^0}(\lambda^0) - \alpha\mathbf{\Psi}^*_{i^0,L(\lambda^0)}\mathbf{1}_{L(\lambda^0)} + (1-\alpha)\mathbf{\Psi}^*_{i^0,R(\lambda^0)}\mathbf{1}_{R(\lambda^0)} \right)$$

$$= \mathbf{\Phi}^*_{i^0,j^*}\hat{\eta}_{j^*}(\lambda^0) + \frac{1}{\lambda^1}f(X_{i^0})$$

Putting it all together, we can choose $\lambda^1$ as,

$$\lambda^1 = \max_{i' \neq i^0, i' \in [m]} \frac{f(X_{i'}) - (\mathbf{\Phi}^*_{i',j^*}/\mathbf{\Phi}^*_{i^0,j^*})f(X_{i^0})}{S_{i'} - (\mathbf{\Phi}^*_{i',j^*}/\Phi^*_{i^0,j^*})S_{i^0}} \tag{38}$$

and the corresponding $i'$ that maximizes equation 38 becomes $i^1$. We proceed with the same $\hat{v}(\lambda^0)$, $\hat{\eta}(\lambda^0)$ as our initial parameters and our initial elbow set as $E(\lambda^1) = \{i^0, i^1\}$.

## B.4 DERIVATION OF $S$-PATH

We fix the hyperparameters $\hat{\gamma}, \hat{\lambda}$ selected by the $\lambda$-path. Conceptually, the $S$-path mirrors the $\lambda$-path, and the conditions 18–24 apply. Now recall the prediction set we defined for a test point $X_{n+1}$,

$$\hat{C}^*(X_{n+1}) = \{y : S(X_{n+1},y) \leq \hat{g}_{S(X_{n+1},y)}(X_{n+1})\}.$$

By equation 24, this is equivalent to,

$$\hat{C}^*(X_{n+1}) = \{y : \hat{v}_{S(X_{n+1},y),n+1} < 1 - \alpha\}.$$

The problem reduces to finding the largest test score $S^*(X_{n+1})$ such that $\hat{v}_{S^*(X_{n+1}),n+1} < 1 - \alpha$. By Proposition 3, the mapping $S \mapsto \hat{v}_S$ is monotone, which allows us to recover the prediction set as,

$$\hat{C}^*(X_{n+1}) = \{y : S(X_{n+1},y) \leq S^*(X_{n+1})\}.$$

It remains to find the maximum $S^*(X_{n+1})$, the test score cutoff, such that $\hat{v}_{S^*(X_{n+1}),n+1} < 1 - \alpha$ holds, i.e., $S^*(X_{n+1}) = \sup\{S \mid \hat{v}_{S,n+1} < 1 - \alpha\}$ which is the role of $S$-path. Denote the index sets,

$$E(S) := \{i : \widehat{r}_{S,i} = 0, \ \hat{v}_{S,i} \in (-\alpha, 1 - \alpha)\}, \tag{39}$$
$$L(S) := \{i : \widehat{r}_{S,i} < 0, \ \hat{v}_{S,i} = -\alpha\}, \tag{40}$$
$$R(S) := \{i : \widehat{r}_{S,i} > 0, \ \hat{v}_{S,i} = 1 - \alpha\}. \tag{41}$$

These sets now *evolve with $S$*. We initialize $S$-path with the smallest $S^1$ such that the test point is in the elbow set (i.e., $S^1 = \hat{g}_{S^1}(X_{n+1})$) and find the smallest increment to the next $S$ such that an event occurs while the test point is still in the elbow. We use the same notion of an "event" as before—any change in the elbow set. We iterate until the test point exits the elbow and use the final $S$ as $S^*(X_{n+1})$.

In this section, we assume the columns of $\mathbf{\Phi}^* \in \mathbb{R}^{(n+1) \times d}$ are linearly independent. The dimension of $\mathbf{\Psi}^*$ is now $\mathbb{R}^{(n+1) \times (n+1)}$.

### B.4.1 PROOF OF PROPOSITION 2

**Proof.** Let $\{S^l\}_{l=1,2,3,\dots}$ be the change points when an event occurs. Consider an interval $S^l \leq S \leq S^{l+1}$ during which the sets stay the same, i.e., $(E(S), L(\lambda), R(S)) = (E(S^l), L(S^l), R(S^l))$. Denote $\hat{v}_{S,i}$ and $\hat{\eta}_S$ as the solution of equation 18 given $S$. In this proof, denote $E = E(S) = E(S^l)$, $L = L(S) = L(S^l)$, and $R = R(S) = R(S^l)$. Here, $\lambda$ is fixed as the selected hyperparameter from the previous step.

For every index $i$ we have,

$$\hat{g}_S(X_i) = \mathbf{\Phi}_{i\cdot}^* \hat{\eta}_S + \frac{1}{\lambda} \mathbf{\Psi}_{i,\cdot}^* \hat{v}_S$$

$$= \mathbf{\Phi}_{i\cdot}^* \hat{\eta}_S + \frac{1}{\lambda} \left( \mathbf{\Psi}_{i,E}^* \hat{v}_{S,E} - \alpha \mathbf{\Psi}_{i,L}^* \mathbf{1}_L + (1-\alpha) \mathbf{\Psi}_{i,R}^* \mathbf{1}_R \right) \qquad (42)$$

$$= \mathbf{\Phi}_{i\cdot}^* \hat{\eta}_S + \frac{1}{\lambda} \mathbf{\Psi}_{i,E}^* \hat{v}_{S,E} + d_i,$$

where

$$d_i := \frac{1}{\lambda}(-\alpha \mathbf{\Psi}_{i,L}^* \mathbf{1}_L + (1-\alpha) \mathbf{\Psi}_{i,R}^* \mathbf{1}_R).$$

From the second KKT condition in equation 21, we have $v_i = \sigma_i - \tau_i$ and $\sum_{i=1}^n (\sigma_i - \tau_i) \mathbf{\Phi}_{i,j}^* = \sum_{i=1}^n v_i \mathbf{\Phi}_{i,j}^* = 0$ for $j = 1, \cdots, d$. In compact form,

$$\mathbf{\Phi}_E^{*\top} \hat{v}_{S,E} = \alpha \, \mathbf{\Phi}_L^{*\top} \mathbf{1}_L - (1-\alpha) \, \mathbf{\Phi}_R^{*\top} \mathbf{1}_R.$$

This means that,

$$\mathbf{\Phi}_E^*(\mathbf{\Phi}_E^{*\top} \mathbf{\Phi}_E^*)^\dagger \mathbf{\Phi}_E^{*\top} \hat{v}_{S,E} = \alpha \, \mathbf{\Phi}_E^*(\mathbf{\Phi}_E^{*\top} \mathbf{\Phi}_E^*)^\dagger \mathbf{\Phi}_L^{*\top} \mathbf{1}_L - (1-\alpha) \, \mathbf{\Phi}_E^*(\mathbf{\Phi}_E^{*\top} \mathbf{\Phi}_E^*)^\dagger \mathbf{\Phi}_R^{*\top} \mathbf{1}_R.$$

Let $S_E := (S_i)_{i \in E}, d_E := (d_i)_{i \in E}, \mathbf{\Phi}_E^* \in \mathbb{R}^{|E| \times p}, \mathbf{\Psi}_{EE}^* \in \mathbb{R}^{|E| \times |E|}$. Equation 42 for $i \in E$ becomes,

$$S_E = \mathbf{\Phi}_E^* \hat{\eta}_S + \frac{1}{\lambda} \mathbf{\Psi}_{EE}^* \hat{v}_{S,E} + d_E. \qquad (43)$$

Define the orthogonal projector, $\mathbf{\Pi}_E := I_{|E|} - \mathbf{\Phi}_E^* \left( \mathbf{\Phi}_E^{*\top} \mathbf{\Phi}_E^* \right)^\dagger \mathbf{\Phi}_E^{*\top}$. Because $\mathbf{\Pi}_E \mathbf{\Phi}_E^* = 0$, multiplying equation 43 by $\mathbf{\Pi}_E$ gives,

$$\mathbf{\Pi}_E S_E = \frac{1}{\lambda} \mathbf{\Pi}_E \mathbf{\Psi}_{EE}^\star \hat{v}_{S,E} + \mathbf{\Pi}_E d_E. \qquad (44)$$

Write $S_E = S_E^{\text{fixed}} + S \, \mathbf{e}_{n+1}$, where $S_E^{\text{fixed}}$ has a zero in the $(n+1)$-st row and $\mathbf{e}_{n+1}$ selects that row. Equation 44 becomes,

$$\mathbf{\Pi}_E \mathbf{\Psi}_{EE}^\star \hat{v}_{S,E} = \lambda \, \mathbf{\Pi}_E \left( S_E^{\text{fixed}} - d_E \right) + \lambda \, S \, \mathbf{\Pi}_E \mathbf{e}_{n+1}. \qquad (45)$$

Since $I_E = \mathbf{\Phi}_E^* \left( \mathbf{\Phi}_E^{*\top} \mathbf{\Phi}_E^* \right)^\dagger \mathbf{\Phi}_E^{*\top} + \mathbf{\Pi}_E$, the previous equation yields,

$$\mathbf{\Pi}_E \mathbf{\Psi}_{EE}^\star \mathbf{\Pi}_E \hat{v}_{S,E} = -\mathbf{\Pi}_E \mathbf{\Psi}_{EE}^\star \mathbf{\Phi}_E^*(\mathbf{\Phi}_E^{*\top} \mathbf{\Phi}_E^*)^\dagger \mathbf{\Phi}_E^{*\top} \hat{v}_{S,E} + \lambda \, \mathbf{\Pi}_E \left( S_E^{\text{fixed}} - d_E \right) + \lambda \, S \, \mathbf{\Pi}_E \mathbf{e}_{n+1}. \qquad (46)$$

Now, we know that:

$$\mathbf{\Pi}_E \mathbf{\Psi}_{EE}^\star \mathbf{\Phi}_E^*(\mathbf{\Phi}_E^{*\top} \mathbf{\Phi}_E^*)^\dagger \mathbf{\Phi}_E^{*\top} \hat{v}_{S,E} = \alpha \, \mathbf{\Pi}_E \mathbf{\Psi}_{EE}^\star \mathbf{\Phi}_E^*(\mathbf{\Phi}_E^{*\top} \mathbf{\Phi}_E^*)^\dagger \mathbf{\Phi}_L^{*\top} \mathbf{1}_L - (1-\alpha) \, \mathbf{\Pi}_E \mathbf{\Psi}_{EE}^\star \mathbf{\Phi}_E^*(\mathbf{\Phi}_E^{*\top} \mathbf{\Phi}_E^*)^\dagger \mathbf{\Phi}_R^\top \mathbf{1}_R.$$

Because $\mathbf{\Pi}_E$ is an orthogonal projector ($\mathbf{\Pi}_E^2 = \mathbf{\Pi}_E$), the matrix $\mathbf{\Pi}_E \mathbf{\Psi}_{EE}^\star \mathbf{\Pi}_E$ is positive definite on the image of $\mathbf{\Pi}_E$. Using its Moore–Penrose inverse (denoted by superscript $\dagger$) gives the unique minimum-norm solution,

$$\mathbf{\Pi}_E \hat{v}_{S,E} = \lambda \left( \mathbf{\Pi}_E \mathbf{\Psi}_{EE}^\star \mathbf{\Pi}_E \right)^\dagger \mathbf{\Pi}_E \left( S_E^{\text{fixed}} - d_E + S \, \mathbf{e}_{n+1} \right)$$

$$- \alpha \left( \mathbf{\Pi}_E \mathbf{\Psi}_{EE}^\star \mathbf{\Pi}_E \right)^\dagger \mathbf{\Pi}_E \mathbf{\Psi}_{EE}^\star \mathbf{\Phi}_E^*(\mathbf{\Phi}_E^{*\top} \mathbf{\Phi}_E^*)^\dagger \mathbf{\Phi}_L^{*\top} \mathbf{1}_L \qquad (47)$$

$$+ (1-\alpha) \left( \mathbf{\Pi}_E \mathbf{\Psi}_{EE}^\star \mathbf{\Pi}_E \right)^\dagger \mathbf{\Pi}_E \mathbf{\Psi}_{EE}^\star \mathbf{\Phi}_E^*(\mathbf{\Phi}_E^{*\top} \mathbf{\Phi}_E^*)^\dagger \mathbf{\Phi}_R^{*\top} \mathbf{1}_R$$

Therefore, since $\hat{v}_{S,E} = \mathbf{\Pi}_E \hat{v}_{S,E} + \mathbf{\Phi}_E^*(\mathbf{\Phi}_E^{*\top} \mathbf{\Phi}_E^*)^\dagger \mathbf{\Phi}_E^{*\top} \hat{v}_{S,E}$:

$$\hat{v}_{S,E} = \lambda \left(\mathbf{\Pi}_E \mathbf{\Psi}_{EE}^\star \mathbf{\Pi}_E\right)^\dagger \mathbf{\Pi}_E \left(S_E^{\text{fixed}} - d_E + S\,\mathbf{e}_{n+1}\right)$$
$$- \alpha \left(\mathbf{\Pi}_E \mathbf{\Psi}_{EE}^\star \mathbf{\Pi}_E\right)^\dagger \mathbf{\Pi}_E \mathbf{\Psi}_{EE}^\star \mathbf{\Phi}_E^* (\mathbf{\Phi}_E^{*\top} \mathbf{\Phi}_E^*)^\dagger \mathbf{\Phi}_L^{*\top} \mathbf{1}_L$$
$$+ (1-\alpha) \left(\mathbf{\Pi}_E \mathbf{\Psi}_{EE}^\star \mathbf{\Pi}_E\right)^\dagger \mathbf{\Pi}_E \mathbf{\Psi}_{EE}^\star \mathbf{\Phi}_E^* (\mathbf{\Phi}_E^{*\top} \mathbf{\Phi}_E^*)^\dagger \mathbf{\Phi}_R^{*\top} \mathbf{1}_R$$
$$+ \alpha\, \mathbf{\Phi}_E^* (\mathbf{\Phi}_E^{*\top} \mathbf{\Phi}_E^*)^\dagger \mathbf{\Phi}_L^{*\top} \mathbf{1}_L$$
$$- (1-\alpha)\, \mathbf{\Phi}_E^* (\mathbf{\Phi}_E^{*\top} \mathbf{\Phi}_E^*)^\dagger \mathbf{\Phi}_R^{*\top} \mathbf{1}_R \tag{48}$$
$$= \lambda \left(\mathbf{\Pi}_E \mathbf{\Psi}_{EE}^\star \mathbf{\Pi}_E\right)^\dagger \mathbf{\Pi}_E \left(S_E^{\text{fixed}} - d_E + S\,\mathbf{e}_{n+1}\right)$$
$$+ \alpha \left[\, I_{|E|} - (\mathbf{\Pi}_E \mathbf{\Psi}_{EE}^\star \mathbf{\Pi}_E)^\dagger \mathbf{\Pi}_E \mathbf{\Psi}_{EE}^\star \right] \mathbf{\Phi}_E^* (\mathbf{\Phi}_E^{*\top} \mathbf{\Phi}_E^*)^\dagger \mathbf{\Phi}_L^{*\top} \mathbf{1}_L$$
$$- (1-\alpha) \left[\, I_{|E|} - (\mathbf{\Pi}_E \mathbf{\Psi}_{EE}^\star \mathbf{\Pi}_E)^\dagger \mathbf{\Pi}_E \mathbf{\Psi}_{EE}^\star \right] \mathbf{\Phi}_E^* (\mathbf{\Phi}_E^{*\top} \mathbf{\Phi}_E^*)^\dagger \mathbf{\Phi}_R^{*\top} \mathbf{1}_R$$

In particular, the kernel parameter of the test point, $\hat{v}_{S,n+1}$, is affine in $S$ on every segment where the index sets $(E, L, R)$ stay unchanged. Likewise, the linear coefficient satisfies,

$$S_E = \mathbf{\Phi}_E^* \hat{\eta}_S + \frac{1}{\lambda} \mathbf{\Psi}_{EE}^\star \hat{v}_{S,E} + d_E$$
$$\mathbf{\Phi}_E^{*\top} S_E = \mathbf{\Phi}_E^{*\top} \mathbf{\Phi}_E^* \hat{\eta}_S + \frac{1}{\lambda} \mathbf{\Phi}_E^{*\top} \mathbf{\Psi}_{EE}^\star \hat{v}_{S,E} + \mathbf{\Phi}_E^{*\top} d_E$$
$$\implies \hat{\eta}_S = (\mathbf{\Phi}_E^{*\top} \mathbf{\Phi}_E^*)^\dagger \mathbf{\Phi}_E^{*\top} S_E - \frac{1}{\lambda} (\mathbf{\Phi}_E^{*\top} \mathbf{\Phi}_E^*)^\dagger \mathbf{\Phi}_E^{*\top} \mathbf{\Psi}_{EE}^\star \hat{v}_{S,E} - (\mathbf{\Phi}_E^{*\top} \mathbf{\Phi}_E^*)^\dagger \mathbf{\Phi}_E^{*\top} d_E \tag{49}$$
$$\implies \hat{\eta}_S = (\mathbf{\Phi}_E^{*\top} \mathbf{\Phi}_E^*)^\dagger \mathbf{\Phi}_E^{*\top} S_E^{\text{fixed}} + S(\mathbf{\Phi}_E^{*\top} \mathbf{\Phi}_E^*)^\dagger \mathbf{\Phi}_E^{*\top} \mathbf{e}_{n+1}$$
$$- \frac{1}{\lambda} (\mathbf{\Phi}_E^{*\top} \mathbf{\Phi}_E^*)^\dagger \mathbf{\Phi}_E^{*\top} \mathbf{\Psi}_{EE}^\star \hat{v}_{S,E} - (\mathbf{\Phi}_E^{*\top} \mathbf{\Phi}_E^*)^\dagger \mathbf{\Phi}_E^{*\top} d_E$$
$$= (\mathbf{\Phi}_E^{*\top} \mathbf{\Phi}_E^*)^\dagger \mathbf{\Phi}_E^{*\top} \left[ S_E^{\text{fixed}} + S\mathbf{e}_{n+1} - d_E - \frac{1}{\lambda} \mathbf{\Psi}_{EE}^\star \hat{v}_{S,E} \right],$$

and thus, we have shown that,

$$\hat{v}_{S,E} = c + Sd, \qquad \hat{\eta}_S = c^{(1)} + Sd^{(1)}. \tag{50}$$

with $c, d \in \mathbb{R}^{|E|}, c^{(1)}, d^{(1)} \in \mathbb{R}^d$ constant on the segment $S^l \le S \le S^{l+1}$.

Insert equation 50 back to equation 42. For any $i \in [n+1]$,

$$\hat{g}_S(X_i) = \mathbf{\Phi}_{i\cdot} \hat{\eta}_S + \frac{1}{\lambda} \mathbf{\Psi}_{iE}^\star \hat{v}_{S,E} + d_i \tag{51}$$

$$= \mathbf{\Phi}_{i\cdot} \left(c^{(1)} + Sd^{(1)}\right) + \frac{1}{\lambda} \mathbf{\Psi}_{iE}^\star \left(c + Sd\right) + d_i \tag{52}$$

$$= \underbrace{\left(\mathbf{\Phi}_{i\cdot} c^{(1)} + \frac{1}{\lambda} \mathbf{\Psi}_{iE}^\star c + d_i\right)}_{=:g_i^{(0)}} + S \underbrace{\left(\mathbf{\Phi}_{i\cdot} d^{(1)} + \frac{1}{\lambda} \mathbf{\Psi}_{iE}^\star d\right)}_{=:g_i^{(1)}}. \tag{53}$$

Thus $g_S(X_i) = g_i^{(0)} + S\,g_i^{(1)}$ is affine in $S$. There are two cases for the residual $r_i(S) = S_i - \hat{g}_S(X_i)$:

1. *Calibration index $i \le n$.* The score $S_i$ is fixed, hence
$$r_i(S) = \left[S_i - g_i^{(0)}\right] - S\,g_i^{(1)}.$$
Both $S_i - g_i^{(0)}$ and $g_i^{(1)}$ are constants on the segment.

2. *Test index $i = n + 1$.* Here $S_{n+1} = S$, so
$$r_{n+1}(S) = S - g_{n+1}^{(0)} - S\,g_{n+1}^{(1)} = \left[1 - g_{n+1}^{(1)}\right] S - g_{n+1}^{(0)},$$
which is again affine in $S$.

Because every $r_i(S)$ is an affine function, each index outside the elbow can cross the zero-residual line at most once on the segment. Likewise each $v_i(S)$ in equation 50 can hit the bounds $1 - \alpha$ or $-\alpha$ at most once. Hence the overall solution path is *piecewise affine* with break-points occurring exactly when either (a) a coefficient in $E$ hits its bound, or (b) a residual for an index in $L \cup R$ reaches zero, completing the argument used by the S-path algorithm.

$\blacksquare$

### B.4.2 UPDATE OF $S^l$

Let $S^l$ the value after the $l^{th}$ event. The elbow set $E(S^l)$ is updated when one of the following events occurs:

- A point $i$ in either $L(S^l)$ or $R(S^l)$ enters the elbow set (residual $S_i - \hat{g}_{S^l}(X_i)$ becomes 0).
- A point $i$ in $E(S^l)$ leaves to the left or right set ($\hat{v}_{S^l,i}$ becomes $-\alpha$ or $1 - \alpha$).

For the first event, note that $\frac{\partial r_i(S)}{\partial S} = -(\mathbf{\Phi}_{i\cdot} d^{(1)} + \frac{1}{\lambda} \mathbf{\Psi}^\star_{iE} d)$ in equation 53. Then we have,

$$S^{l+1,hit} = S^l + \min_{i \in L(S^l), R(S^l)} r_i(S^l) / \left( \mathbf{\Phi}_{i\cdot} d^{(1)} + \frac{1}{\lambda} \mathbf{\Psi}^\star_{iE} d \right) \mathbf{1} \left( r_i(S^l) / \left( \mathbf{\Phi}_{i\cdot} d^{(1)} + \frac{1}{\lambda} \mathbf{\Psi}^\star_{iE} d \right) \geq 0 \right)$$

To find $S$ such that a point leaves $E(S^l)$, recall $\frac{\partial \hat{v}_{S,i}}{\partial S} = d_i$ for $i \in E(S^l)$ (equation 53).

$$S^{l+1,leave} = S^l + \min_{i \in E(S^l)} \left\{ x \in \left\{ \frac{-\alpha - \hat{v}_{S^l,i}}{d_i}, \frac{1 - \alpha - \hat{v}_{S^l,i}}{d_i} \right\} \mid x \leq 0 \right\}$$

We then take $S^{l+1} = \min\{S^{l+1,hit}, S^{l+1,leave}\}$. We also update $(E, L, R) = (E(S^{l+1}), L(S^{l+1}), R(S^{l+1}))$ accordingly based on which event occurred. Parameters $\hat{v}_{S^{l+1},i}$'s, $\hat{\eta}_{S^{l+1}}$ can be updated by solving for the new elbow,

$$\begin{pmatrix} \frac{1}{\lambda} \mathbf{\Psi}^\star_{EE} & \mathbf{\Phi}^*_E \\ \mathbf{\Phi}^{*\top}_E & 0 \end{pmatrix} \begin{pmatrix} v \\ \eta \end{pmatrix} = \begin{pmatrix} S_E - \frac{1}{\lambda}(-\alpha \mathbf{\Psi}^\star_{EL} \mathbf{1}_L + (1-\alpha) \mathbf{\Psi}^\star_{ER} \mathbf{1}_R) \\ \alpha \mathbf{\Phi}^{*\top}_L \mathbf{1}_L - (1-\alpha) \mathbf{\Phi}^{*\top}_R \mathbf{1}_R \end{pmatrix} \tag{54}$$

### B.4.3 INITIALIZATION OF $S$

Let us first assume that the imputed test score $S$ is small enough so that $S < \hat{g}_S(X_{n+1})$. Then, the test point $n + 1 \in L(S)$. We use the notation $\hat{v}_{small}$ and $\hat{\eta}_{small}$ (instead of $\hat{v}_S$ and $\hat{\eta}_S$), to denote the regression parameters. In this case, the residual $r_{n+1}(S) = S - \hat{g}_{small}(X_{n+1}) = S - \mathbf{\Phi}^*_{n+1,\cdot} \hat{\eta}_{small} - \frac{1}{\lambda} \mathbf{\Psi}^\star_{n+1,\cdot} \hat{v}_{small}$ is linear in $S$. We can therefore track the moment when it enters the elbow set $E(S)$. This happens as soon as:

$$S = \mathbf{\Phi}^*_{n+1,\cdot} \hat{\eta}_{small} + \frac{1}{\lambda} \mathbf{\Psi}^\star_{n+1,\cdot} \hat{v}_{small}.$$

We thus solve for $\hat{v}_{small}$ and $\hat{\eta}_{small}$ with $v_{small,n+1} = -\alpha$ (as the test point is in the left set). Let $v^{fixed} \in \mathbb{R}^{n+1}$ be the vector defined as equal to $v_S$ on all entries except the $(n+1)^{th}$, where it is set to 0:

$$v_i^{fixed} = \begin{cases} v_{S,i} \text{ if } i \neq n + 1 \\ 0 \text{ if } i = n + 1. \end{cases}$$

This allows us to write,

$$v_{small} = v^{fixed} - \alpha \mathbf{e}_{n+1},$$

where $\mathbf{e}_{n+1}$ is the indicator vector of the $(n+1)^{th}$ coefficient,

$$\mathbf{e}_{n+1,i} = 0 \text{ if } i \neq n + 1, \quad \mathbf{e}_{n+1,n+1} = 1.$$

The problem then becomes,

$$
\begin{aligned}
L_p &= (1-\alpha)\sum_{i=1}^{n+1} p_i + \alpha\sum_{i=1}^{n+1} q_i + \frac{1}{2\lambda}\upsilon_{\mathrm{small}}^\top \boldsymbol{\Psi}^\star \upsilon_{\mathrm{small}} \\
&\quad + \sum_{i=1}^{n+1}\sigma_i(S_i - g_{\mathrm{small}}(X_i) - p_i) - \sum_{i=1}^{n+1}\tau_i(S_i - g_{\mathrm{small}}(X_i) + q_i) \\
&\quad - \sum_{i=1}^{n+1}\kappa_i p_i - \sum_{i=1}^{n+1}\rho_i q_i \\
&= \sum_{i=1}^{n+1}\sigma_i p_i + \sum_{i=1}^{n+1}\tau_i q_i + \frac{1}{2\lambda}\upsilon_{\mathrm{small}}^\top \boldsymbol{\Psi}^\star \upsilon_{\mathrm{small}} \\
&\quad + \sum_{i=1}^{n+1}(\sigma_i - \tau_i)(S_i - \boldsymbol{\Phi}_{i,.}^*\eta_{\mathrm{small}} - \frac{1}{\lambda}\boldsymbol{\Psi}_{i.}^\star\upsilon_{\mathrm{small}}) - \sum_{i=1}^{n+1}(\sigma_i p_i + \tau_i q_i) \\
&= \frac{1}{2\lambda}(\upsilon_{\mathrm{small}}^{\mathrm{fixed}} - \alpha\mathbf{e}_{n+1})^\top \boldsymbol{\Psi}^\star (\upsilon_{\mathrm{small}}^{\mathrm{fixed}} - \alpha\mathbf{e}_{n+1}) \\
&\quad + \sum_{i=1}^{n}(\sigma_i - \tau_i)(S_i - \boldsymbol{\Phi}_{i,.}^*\eta_{\mathrm{small}} - \frac{1}{\lambda}\boldsymbol{\Psi}_{i.}^\star(\upsilon_{\mathrm{small}}^{\mathrm{fixed}} - \alpha\mathbf{e}_{n+1})) \\
&\quad - \alpha(S - \boldsymbol{\Phi}_{n+1,.}^*\eta_{\mathrm{small}} - \frac{1}{\lambda}\boldsymbol{\Psi}_{n+1,1:n}^\star(\upsilon_{\mathrm{small}}^{\mathrm{fixed}} - \alpha\mathbf{e}_{n+1})) \\
&= \frac{1}{2\lambda}\delta^\top \boldsymbol{\Psi}_{1:n,1:n}^\star\delta - \frac{\alpha}{\lambda}\boldsymbol{\Psi}_{n+1,1:n}^\star\delta + \frac{\alpha^2}{2\lambda}\boldsymbol{\Psi}_{n+1,n+1}^\star \\
&\quad + \delta^T(S_{1:n} - \boldsymbol{\Phi}_{1:n,.}^*\eta_{\mathrm{small}} - \frac{1}{\lambda}\boldsymbol{\Psi}_{1:n,1:n}^\star\delta + \frac{\alpha}{\lambda}\boldsymbol{\Psi}_{1:n,n+1}^\star) \\
&\quad - \alpha(S - \boldsymbol{\Phi}_{n+1,.}^*\eta_{\mathrm{small}} - \frac{1}{\lambda}\Psi_{n+1,1:n}^\star\delta + \frac{\alpha}{\lambda}\boldsymbol{\Psi}_{n+1,n+1}^\star)),
\end{aligned}
\tag{55}
$$

with $\delta \in \mathbb{R}^n$ the vector whose entries are defined as: $\delta_i = \sigma_i - \tau_i = \upsilon_{\mathrm{small},i}$.

We also know that $(\sigma - \tau)^\top \boldsymbol{\Phi}^* = 0 \implies \forall j \in [d], \sum_{i=1}^{n}(\sigma_i - \tau_i)\Phi_{i,j}^* = -(\sigma_{n+1} - \tau_{n+1})\Phi_{n+1,j}^* = \alpha\Phi_{n+1,j}^*$.

Therefore, taking derivatives with respect to $\delta$ yields the following system:

$$
\begin{pmatrix} \frac{1}{\lambda}\boldsymbol{\Psi}_{EE}^\star & \boldsymbol{\Phi}_E^* \\ \boldsymbol{\Phi}_E^{*\top} & 0 \end{pmatrix}\begin{pmatrix} \delta_E \\ \eta_{\mathrm{small}} \end{pmatrix} = \begin{pmatrix} S_E - \frac{1}{\lambda}(-\alpha\boldsymbol{\Psi}_{E,n+1}^\star - \alpha\boldsymbol{\Psi}_{EL}^\star\mathbf{1}_L + (1-\alpha)\boldsymbol{\Psi}_{ER}^\star\mathbf{1}_R) \\ \alpha\boldsymbol{\Phi}_{n+1}^{*\top} + \alpha\boldsymbol{\Phi}_L^{*\top}\mathbf{1}_L - (1-\alpha)\boldsymbol{\Phi}_R^{*\top}\mathbf{1}_R \end{pmatrix}
\tag{56}
$$

We can therefore solve for both $\delta_E$ and $\eta_{\mathrm{small}}$ by inverting the previous system of equations.

**When $S > g(X_{n+1})$.** Similarly, when we start with a large $S$ so that the test point is in the right set, we can derive the coefficients for both $\upsilon$ and $\eta$ by the same derivations as for the small case:

$$
\begin{pmatrix} \frac{1}{\lambda}\boldsymbol{\Psi}_{EE}^\star & \boldsymbol{\Phi}_E^* \\ \boldsymbol{\Phi}_E^{*\top} & 0 \end{pmatrix}\begin{pmatrix} \delta_E \\ \eta_{\mathrm{small}} \end{pmatrix} = \begin{pmatrix} S_E - \frac{1}{\lambda}((1-\alpha)\boldsymbol{\Psi}_{E,n+1}^\star - \alpha\boldsymbol{\Psi}_{EL}^\star\mathbf{1}_L + (1-\alpha)\boldsymbol{\Psi}_{ER}^\star\mathbf{1}_R) \\ (\alpha - 1)\boldsymbol{\Phi}_{n+1}^{*\top} + \alpha\boldsymbol{\Phi}_L^{*\top}\mathbf{1}_L - (1-\alpha)\boldsymbol{\Phi}_R^{*\top}\mathbf{1}_R \end{pmatrix}
\tag{57}
$$

### B.4.4 COMPUTATIONAL COMPLEXITY

**Complexity** At each step of $\lambda$- and $S$-path we take inverse of matrices whose size are at most $|E| + d$. The overall cost is $\mathcal{O}((n + d)^3)$ in the worst case, but empirical paths have $|E| \ll n$ and at most $2(n + 1)$ break-points, making the routine fast in practice.

**Practical consequences**

- **Threshold evaluation** Because $S \mapsto v_{S,n+1}$ is affine on each segment, the conformal threshold $\hat{g}(X_{n+1}) = \sup\{S : v_{S,n+1} < 1 - \alpha\}$ is found by a single root computation, *not* by binary search.

- **Randomization** By Lemma 4, using the final update of $S$-path, $S^*(X_{n+1})$, can inflate the conditional coverage. To mitigate this, we can use the randomized cutoff $S^{rand}(X_{n+1}) = \sup\{S \mid \hat{v}_{S,n+1} < U\}$, for $U \sim Unif(-\alpha, 1 - \alpha)$. The procedure of $S$-path stays the same but we stop the algorithm as soon as $\hat{v}_{S,n+1} \geq U$.

**Summary of modifications versus the $\lambda$–path**

| Component | $\lambda$–path | S–path (fixed $\lambda$) |
|---|---|---|
| Moving parameter | $\lambda \downarrow 0$ | $S \uparrow$ |
| Active sets | $E(\lambda), L(\lambda), R(\lambda)$ | $E(S), L(S), R(S)$ |
| Triggering event | Elbow set changes | Elbow set changes |
| Segment law | $\lambda \mapsto \hat{v}(\lambda)$ affine | $S \mapsto \hat{v}_S$ affine |
| Break-points | critical values of $\lambda$ | critical values of $S$ |
| Output | $\lambda \mapsto (v, \eta)$ | $S \mapsto (v, \eta)$ |

The resulting algorithm furnishes an explicit, efficient *score-path* for any fixed $\lambda$, enabling local density–adaptive conformal prediction and other post-hoc analyses.

## B.5 CROSS VALIDATION ON $\gamma$

In this section, we provide additional details on the selection of $(\hat{\gamma}, \hat{\lambda})$, described in Algorithm 1.

We first select a fixed grid of $\gamma$'s, denoted as $\Gamma$. For each $\gamma \in \Gamma$, we measure k-fold validation error. This is done by running a $\lambda$-path for each $\gamma$ that yields a sequence of $\lambda$ values, $(\lambda^1, \lambda^2, \cdots)$. The error of $(\gamma, \lambda^l)$ for the $j$-th fold is then defined as the quantile regression loss,

$$\mathrm{CV}_j(\gamma, \lambda^l) = \sum_{i' \in fold_j} \left( (1 - \alpha)[S_{i'} - \hat{g}^l(X_i)]_+ + \alpha[S_{i'} - \hat{g}^l(X_i)]_- \right)$$

A common practice is to get the mean of $\mathrm{CV}_j(\gamma, \lambda^l)$ over $k$ folds and choose the pair that minimizes it. However, running a $\lambda$-path for each $\gamma$ does not guarantee that the $\lambda$ values on the path will be all equivalent. To avoid this issue, we proceed with a two-step approach that first aggregates the error over all $\lambda$'s to get the optimal $\hat{\gamma}$,

$$\mathrm{CV}_j(\gamma) = \min_{\lambda^l} \mathrm{CV}_j(\gamma, \lambda^l)$$

$$\hat{\gamma} = \arg\min_\gamma \mathrm{CV}(\gamma) = \arg\min_\gamma \sum_{j=1}^{k} \mathrm{CV}_j(\gamma)$$

This selects $\hat{\gamma}$ that reflects the best smoothness of the function overall. Once we select $\hat{\gamma}$, we run the $\lambda$-path again on the full dataset ($m$ points) and choose $\hat{\lambda}$ that minimizes the Schwarz information criterion (SIC) (Schwarz, 1978), which is a commonly used alternative to cross validation error in kernel quantile regression for model selection (Li et al., 2007).

## C THEORETICAL PROOF

### C.1 GUARANTEE FOR RANDOMIZED INTERVAL

To incorporate the structured RKHS-based function in equation 6 into the conformal calibration framework in Gibbs et al. (2023), we need to show two propositions. Firstly, we show the monotonicity of the solution path for $S$. Namely, the mapping $S \mapsto v_{S,n+1}$ is nondecreasing in $S$. Second,

we require the low-rank projection $\hat{\pi}(\cdot)$ to be trained symmetrically across the input data. With these properties established, we are able to prove that our path algorithm satisfies exactly the same type of results as Gibbs et al. (2023):

**Lemma 4** *Consider the function class $\mathcal{F}^*$ in equation 17, where RKHS component is given with the optimal $\hat{\lambda}$ such that $\mathcal{F}_{\psi^*} = \{f_{\psi^*}(x) = \frac{1}{\hat{\lambda}} \sum_{i \in [n+1]} v_i \psi^*(x, X_i), v \in \mathbb{R}^{n+1}\}$. Suppose assumptions 1 and 2 are both satisfied. Then, for all $f \in \mathcal{F}^*$, SpeedCP gives*

$$\mathbb{E}\left[ f(X_{n+1}) \cdot \left( \mathbf{1}\{Y_{n+1} \in \hat{C}^*_{rand}(X_{n+1})\} - (1-\alpha) \right) \right] = -\hat{\lambda}\mathbb{E}\left[ \langle \hat{g}_{S^{rand}, \psi^*}, f_{\psi^*} \rangle \right],$$

*where $\hat{g}_{S^{rand}, \psi^*}(X) = \frac{1}{\hat{\lambda}} \sum_{i \in [n+1]} \hat{v}_{S^{rand}, i} \psi^*(X, X_i)$.*

This result aligns with the randomization version of Theorem 3 in Gibbs et al. (2023) – but adapted here to our algorithm and choice of RKHS class $\mathcal{F}_{\psi^*}$. While in Gibbs et al. (2023) $v_i$ can be any arbitrary value, we involve the optimal $\hat{\lambda}$ in the definition of $f_{\psi^*}$. In this type of RKHS class, the relationship between $S$ to $v_{S,n+1}$ is explicit, while Gibbs et al. (2023) depends on a dual analysis, making the parameter less interpretable. Furthermore, the coverage gap $\mathbb{E}[\langle \hat{g}_{S^{rand}, \psi^*}, f_{\psi^*} \rangle]$ arises because we have no prior information on the distribution shift and use a flexible RKHS-based function class instead. While it may lead to deviations from the nominal level $1 - \alpha$ when $f_{\psi^*} \neq 0$, this deviation is measurable as shown by Gibbs et al. (2023); we detail how to estimate this deviation in the latent-space setting in Appendix C.4.2.

**Proof.**

By Proposition 3, $S \mapsto v_{S,n+1}$ is non-decreasing in $S$. Furthermore, strong duality holds for the optimization problem in equation 7 (this has been shown in Gibbs et al. (2023)), and the KKT conditions are satisfied as shown in 22. Now consider a random variable $U \sim Unif(-\alpha, 1 - \alpha)$. Then we have the equivalence under the randomization for a given $S_{n+1} = S(X_{n+1}, y)$:

$$\mathbf{1}\{S_{n+1} \leq \hat{g}_{S_{n+1}}(X_{n+1})\} \iff \mathbf{1}\{\hat{v}_{S_{n+1}, n+1} \leq U\}$$

Thus,

$$\mathbb{E}\left[ f(X_{n+1}) \left( \mathbf{1}\left\{ \hat{v}_{S_{n+1}, n+1} \leq U \right\} - (1-\alpha) \right) \right]$$
$$= \mathbb{E}[\mathbb{E}_U \left[ f(X_{n+1}) \left( \mathbf{1}\left\{ \hat{v}_{S_{n+1}, n+1} \leq U \right\} - (1-\alpha) \right) \mid X_{n+1}, \hat{v}_{S_{n+1}, n+1}]]$$
$$= -\mathbb{E}\left[ f(X_{n+1})\hat{v}_{S_{n+1}, n+1} \right]$$

Using the Lagrangian in Proposition 3, we follow the calculation in the proof of Proposition 4 of Gibbs et al. (2023). By the exchangeability of the data and the symmetry of $\hat{g}_{S_{n+1}}$, we have

$$-\mathbb{E}\left[ f(X_{n+1})\hat{v}_{S_{n+1}, n+1} \right] = -2\mathbb{E}\left[ \lambda\langle \hat{g}_{S_{n+1}, \psi^*}, f_{\psi^*} \rangle \right].$$

Therefore, we replace $S_{n+1}$ with the randomized cutoff $S^{rand}$ and $\lambda$ with the optimal $\hat{\lambda}$ to obtain the desired result. ∎

**Proposition 3** *For all maximizers $\{v_{S,n+1}\}_{S \in \mathbb{R}}$ of the optimization problem in equation 7, the mapping $S \mapsto v_{S,n+1}$ is non-decreasing in $S$.*

**Proof.** Recall the objective in equation 7:

$$\min_{\eta_S, v_S} \sum_{i=1}^{n+1} l_\alpha\left(S_i - \Phi^*(X_i)^\top \eta_S - \frac{1}{\lambda} \sum_{i'=1}^{n+1} v_{S,i'}\psi^*(X_i, X_{i'})\right) + \frac{1}{2\lambda} \sum_{i,i=1}^{n+1} v_{S,i} v_{S,i'}\psi^*(X_i, X_{i'}).$$

Let $\mathbf{\Psi}^* = (\psi^*(x_i, x_j))_{i,j \in [n+1]} \in \mathbb{R}^{(n+1) \times (n+1)}$ be the positive semidefinite kernel matrix. Following the structure in Li et al. (2007), this objective is equivalent to the following quadratic program for a fixed imputed value $S$ (with $S_{n+1} = S$),

$$\min_{\eta_S, \boldsymbol{v}_S} (1-\alpha) \sum_{i=1}^{n} p_i + \alpha \sum_{i=1}^{n} q_i + \frac{1}{2\lambda_{n+1}} v_S^\top \mathbf{\Psi}^* v_S,$$

subject to

$$-q_i \leq S_i - g_S(x_i) \leq p_i,$$
$$q_i, p_i \geq 0, \quad i = 1, \ldots, n+1,$$

where

$$g_S(x_i) = \Phi^*(x)^\top \eta_S + \frac{1}{\lambda} \sum_{i'=1}^{n+1} \upsilon_{S,i'} \psi^*(x_i, x_{i'}), \quad i = 1, \ldots, n+1.$$

Note that the proof of Proposition 3 follows the argument structure of Theorem 4 in Gibbs et al. (2023), but with a key distinction that the function $g_S(x)$ in our case incorporates an RKHS-based component that depends on $\lambda$. The Lagrangian primal function is then defined as in equation 20. Setting the partial derivatives of $L_p$ with respect to $q$ and $p$ to zero, we obtain

$$\frac{\partial L_p}{\partial p_i} : \sigma_i = 1 - \alpha - \kappa_i$$
$$\frac{\partial L_p}{\partial q_i} : \tau_i = \alpha - \rho_i \tag{58}$$

Since minimizing with respect to $\upsilon$ yields $\upsilon_i = \sigma_i - \tau_i$, we can substitute this into the derivative expressions in Equation equation 58. We have

$$(1 - \alpha) \cdot \mathbf{1} - \upsilon = \kappa + \tau$$
$$\alpha \cdot \mathbf{1} + \upsilon = \rho + \sigma$$

Since $\kappa, \sigma, \tau, \rho$ are all non-negative, this can be simplified to

$$(1 - \alpha) \cdot \mathbf{1} \geq \upsilon$$
$$-\alpha \cdot \mathbf{1} \leq \upsilon$$

Let $Q^*(\upsilon) = -\min_{g \in \mathcal{F}^*} \frac{1}{2\lambda_{n+1}} \upsilon^\top \mathbf{\Psi}^* \upsilon - \sum_{i=1}^{n+1} \upsilon_i g(X_i)$. Therefore, the dual formulation for equation 20 is,

$$\text{maximize}_\upsilon \sum_{i=1}^{n} \upsilon_i S_i + \upsilon_{n+1} S - Q^*(\upsilon)$$

$$\text{subject to } -\alpha \leq \upsilon_i \leq 1 - \alpha, 1 \leq i \leq n+1$$

Note we use notation $\upsilon_S$ to denote the solution for a particular input $S$. Assume for the sake of contradiction that there exists $\tilde{S} > S$ such that

$$\upsilon_{\tilde{S}, n+1} < \upsilon_{S, n+1}.$$

Observe that $\sum_{i=1}^{n} \upsilon_i S_i - Q^*(\upsilon)$ does not depend on $S$. The contradiction assumption implies that

$$(\tilde{S} - S) \cdot \left( \upsilon_{\tilde{S}, n+1} - \upsilon_{S, n+1} \right) < 0,$$

or equivalently,

$$\tilde{S} \cdot \left( \upsilon_{\tilde{S}, n+1} - \upsilon_{S, n+1} \right) < S \cdot \left( \upsilon_{\tilde{S}, n+1} - \upsilon_{S, n+1} \right).$$

On the other hand, by the optimality of $\upsilon_S$, we have that

$$\sum_{i=1}^{n} \upsilon_{\tilde{S}, i} S_i - Q^*(\upsilon_{\tilde{S}}) + \tilde{S} \cdot \upsilon_{\tilde{S}, n+1} \geq \sum_{i=1}^{n} \upsilon_{S,i} S_i - Q^*(\upsilon_S) + S \cdot \upsilon_{S, n+1}$$

$$\iff \quad \tilde{S} \cdot \left( \upsilon_{\tilde{S}, n+1} - \upsilon_{S, n+1} \right) \geq \sum_{i=1}^{n} \upsilon_{S,i} S_i - Q^*(\upsilon_S) - \sum_{i=1}^{n} \upsilon_{\tilde{S}, i} S_i - Q^*(\upsilon_{\tilde{S}}).$$

Applying inequality given by assumption above, we conclude that

$$S \cdot \left( \upsilon_{\tilde{S}, n+1} - \upsilon_{S, n+1} \right) > \sum_{i=1}^{n} \upsilon_{S,i} S_i - Q^*(\upsilon_S) - \sum_{i=1}^{n} \upsilon_{\tilde{S}, i} S_i - Q^*(\upsilon_{\tilde{S}}),$$

which yields the contradiction

$$\sum_{i=1}^{n} \upsilon_{\tilde{S}, i} S_i - Q^*(\upsilon_{\tilde{S}}) + \tilde{S} \cdot \upsilon_{\tilde{S}, n+1} > \sum_{i=1}^{n} \upsilon_{S,i} S_i - Q^*(\upsilon_S) + S \cdot \upsilon_{S, n+1}$$

**Remark 5** *In this proof we treat $\lambda$ as fixed. Because $\lambda$ is pre-selected before entering the S-path, the nondecreasing property of $\hat{v}_S$ holds for each $\lambda$—including the optimal $\hat{\lambda}$ selected by the SIC criterion along the $\lambda$-path.*

∎

## C.2 PROOF OF THEOREM 1

First, we consider the setting that $(X_1, Y_1), \ldots, (X_n, Y_n)$ are independent of $(X_{n+1}, Y_{n+1}, W')$. Since $\hat{\pi}(\cdot)$ is a deterministic function (not a random variable), $\hat{\pi}(X_1), \ldots, \hat{\pi}(X_n)$ are also independent of $\hat{\pi}(X_{n+1})$. Since $\hat{\pi}(\cdot)$ is a pre-trained map from the covariate space to the latent space, we write $\hat{\pi} : \mathcal{X} \to \mathcal{W}$, where $\mathcal{W}$ denotes the latent representation space. Given this embedding, we define a kernel directly on the latent space $\psi_W^* : \mathcal{W} \times \mathcal{W} \to \mathbb{R}$. Consequently, $\psi^*(x, x') = \psi_W^*(\hat{\pi}(x), \hat{\pi}(x'))$. In our construction, $\psi^*$ is already normalized, so $\psi_W^*(w, \cdot)$ is a density kernel in its second argument with respect to a base measure on $\mathcal{W}$, i.e.,

$$\int_{\mathcal{W}} \psi_W^*(w, w') dw' = 1 \qquad \text{for each } w \in \mathcal{W}.$$

Let $P = P_X \times P_{Y|X}$. By the definition of $W'$, the joint distribution of $(X_{n+1}, Y_{n+1}, W')$ is defined by

$$X_{n+1} \sim P_X;$$
$$Y_{n+1} \mid X_{n+1} \sim P_{Y|X};$$
$$W' \mid (X_{n+1}, Y_{n+1}) \sim \psi^*(X_{n+1}, \cdot).$$

By definition of $\psi_W^*$, we equivalently have $W' \mid (\hat{\pi}(X_{n+1}), Y_{n+1}) \sim \psi_W^*(\hat{\pi}(X_{n+1}), \cdot)$. Then, the conditional distribution $(X_{n+1}, Y_{n+1}) \mid W'$ is given by

$$(X_{n+1}, Y_{n+1}) \mid W' = w' \sim \frac{(P_X \circ \psi_W^*(\hat{\pi}(X_{n+1}), w')) \times P_{Y|X}}{\int_{(x,y)} (P_X \circ \psi_W^*(\hat{\pi}(X_{n+1}), w')) \times P_{Y|X} dx dy}$$

$$\sim \frac{\psi_W^*(\hat{\pi}(x), w')}{\mathbb{E}[\psi_W^*(\hat{\pi}(X), w')]} dP_{(X,Y)}(x, y) := dP_f(x) \text{ by the symmetric of } \hat{\pi}(\cdot)$$

Thus conditioning on $W'$, we get

$$\mathbb{E}\left[ \mathbf{1}\{Y_{n+1} \in \hat{C}_{rand}^*(X_{n+1})\} - (1 - \alpha) \mid W' \right]$$

$$= \int \frac{\psi_W^*(\hat{\pi}(x), W')}{\mathbb{E}[\psi_W^*(\hat{\pi}(X), W')]} \left( \mathbf{1}\{y \in \hat{C}_{rand}^*(x)\} - (1 - \alpha) \right) dP_{X,Y}(x, y)$$

$$= \frac{\mathbb{E}\left[ \psi_W^*(\hat{\pi}(X_{n+1}), W') \cdot \left( \mathbf{1}\{Y_{n+1} \in \hat{C}_{rand}^*(X_{n+1})\} - (1 - \alpha) \right) \right]}{\mathbb{E}[\psi^*(X, x')]}$$

$$= \frac{-\hat{\lambda} \mathbb{E}\left[ \sum_{i \in [n+1]} \hat{v}_{S^{rand}, i} / \hat{\lambda} \cdot \psi_W^*(\hat{\pi}(X_i), W') \right]}{\mathbb{E}[\psi_W^*(\hat{\pi}(X), W')]} \qquad \text{by Lemma 4}$$

$$= \frac{-\mathbb{E}\left[ \sum_{i \in [n+1]} \hat{v}_{S^{rand}, i} \psi_W^*(\hat{\pi}(X), W') \right]}{\mathbb{E}[\psi_W^*(\hat{\pi}(X), W')]} \qquad \text{by the structure of } \hat{g}_{S^{rand}, \psi^*}$$

## C.3 PROOF OF THEOREM 2

By the definitions in Theorem 2, for all $i \in [n]$

$$W_i \sim P_W;$$
$$X_i \mid W_i \sim P_{X|W};$$
$$Y_i \mid X_i \sim P_{Y|X}.$$

In this procedure, we say $Y$ is conditionally independent of $W$ given $X$. In practice, the latent variables $\{W_i\}_{i \in [n+1]}$ are unobserved. Firstly, for the joint distribution, we have $\{(W_i, X_i, Y_i)\}_{i \in [n]}$ independent of $(W_{n+1}, X_{n+1}, Y_{n+1})$. In this framework, we consider the covariate shifts such that the tilt function $f(X) = \mathbf{1}\{\arg\max_{k' \in [K]} \pi_{k'}(X) = k\}$ for a fixed $k$. Therefore,

$$X_{n+1}, W_{n+1} \sim \frac{f(x)}{\mathbb{E}[f(X)]} P_{(X,W)};$$

$$Y_{n+1} \mid X_{n+1} \sim P_{Y|X}.$$

This gives that

$$X_{n+1} \sim \int_W \frac{f(x)}{\mathbb{E}[f(X)]} P_{X|W} P(W) dW = \frac{f(x)}{\mathbb{E}[f(X)]} \int_W P_{X|W} P(W) dW$$

$$= \frac{f(x)}{\mathbb{E}_X[f(X)]} dP_X(x) := dP_f(x)$$

Under this setting, we have for any set $C$ under the distribution $dP_f(x)$

$$\mathbb{E}\left[\mathbf{1}\{Y_{n+1} \in C(X_{n+1}) - (1-\alpha)\}\right]$$

$$= \int \left(\mathbf{1}\{Y_{n+1} \in C(X_{n+1}) - (1-\alpha)\}\right) \frac{f(X_{n+1})}{\mathbb{E}[f(X)]} dP_X P_{Y|X}$$

$$= \frac{\mathbb{E}[f(X_{n+1})\left(\mathbf{1}\{Y_{n+1} \in C(X_{n+1}) - (1-\alpha)\}\right)]}{\mathbb{E}[f(X)]}$$

Note that in the oracle setting, the prediction set is constructed directly using the true embedding $\pi(\cdot)$. In this case, the RKHS class $\mathcal{F}^*$ used in our quantile regression is defined over the latent space induced by $\pi$, and we can therefore apply Lemma 4 with such a $\mathcal{F}^*$. By the Lemma 4, we see the numerator equals zero since the function $f$ selected does not depend on the RKHS part. Therefore, we have

$$\frac{\mathbb{E}[f(X_{n+1})\left(\mathbf{1}\{Y_{n+1} \in C(X_{n+1})\}\right)]}{\mathbb{E}[f(X)]}$$

$$= \frac{\mathbb{P}(Y_{n+1} \in C(X_{n+1}), T(X_{n+1}) = k)}{\mathbb{P}(T(X_{n+1}) = k)}$$

$$= \mathbb{P}(Y_{n+1} \in C(X_{n+1}) \mid \arg\max_{k' \in [K]} \pi_{k'}(X_{n+1}) = k) = 1 - \alpha$$

Next, we establish group-conditional coverage when groups are defined via the estimated low-rank embedding $\hat{\pi}(\cdot)$.

**Corollary 6** *Fix $K \geq 2$ and let the latent mixture weights $\{W_i \in \Delta^{K-1}\}_{i=1}^n$ satisfy $W_i \overset{i.i.d.}{\sim} P_W$, with observations $\{X_i \mid W_i\}_{i=1}^n \overset{i.i.d.}{\sim} P_{X|W}$. Suppose we have an estimated embedding $\hat{\pi} : \mathcal{X} \to \mathbb{R}^K$ for $\pi(\cdot)$ which is defined in Theorem 2, and define $\hat{T}(X) := \arg\max_{k \in [K]} \hat{\pi}_k(X)$. Assume Assumptions 1 and 2 hold, and that $\mathbb{P}(\hat{T}(X) = k) > 0$ for all $k \in [K]$. Let $\hat{C}^*_{\text{rand}}(\cdot)$ be the randomized conformal set calibrated using the linear feature map*

$$\Phi^*(X) = \left(\mathbf{1}\{\hat{T}(X) = 1\}, \ldots, \mathbf{1}\{\hat{T}(X) = K\}\right)^\top.$$

*Then, for every $k \in [K]$,*

$$\mathbb{P}\left(Y_{n+1} \in \hat{C}^*_{\text{rand}}(X_{n+1}) \mid \hat{T}(X_{n+1}) = k\right) = 1 - \alpha. \tag{59}$$

**Proof.** The proof follows the same argument as Theorem 2. The only difference is that the tilting function is now taken to be

$$f(x) = \mathbf{1}\{\hat{T}(x) = k\}, \qquad \text{for a fixed } k \in [K],$$

so the reweighted distribution corresponds to conditioning on the estimated dominated group $\hat{T}(X) = k$. All other steps remain identical. Since the RKHS class $\mathcal{F}^*$ is now defined with respect

to the estimated embedding $\hat{\pi}(\cdot)$, we applying Lemma 4 with this tilting function yields the desired group-conditional guarantee. To connect representation-conditional guarantee in equation 59 to equation 11, we apply the law of total probability such that, for any set $\hat{C}(X)$,

$$\mathbb{P}\big(Y \in \hat{C}(X) \,\big|\, T(X)\big)$$

$$= \sum_{k=1}^{K} \mathbb{P}\Big(Y \in \hat{C}(X), \hat{T}(X) = k\big|\, T(X)\Big)$$

$$= \sum_{k=1}^{K} \mathbb{P}\Big(Y \in \hat{C}(X)\big|\hat{T}(X) = k,\, T(X)\Big) \cdot \mathbb{P}\Big(\hat{T}(X) = k\big|T(X)\Big)$$

$$= \sum_{k=1}^{K} \mathbb{P}\big(Y \in \hat{C}(X)\big|\hat{T}(X) = k\big)\mathbb{P}\big(\hat{T}(X) = k\big|T(X)\big)$$

$$+ \sum_{k=1}^{K} \big[\mathbb{P}\big(Y \in \hat{C}(X)\big|T(X), \hat{T}(X) = k\big) - \mathbb{P}\big(Y \in \hat{C}(X)\big|\hat{T}(X) = k\big)\big] \cdot \mathbb{P}\big(\hat{T}(X) = k\big|T(X)\big).$$

The term $\mathbb{P}\big(Y \in \hat{C}(X)\big|T(X), \hat{T}(X) = k\big) \equiv \mathbb{P}\big(Y \in \hat{C}(X)\big|\hat{T}(X) = k\big)$ for all $k$ if the conformity score is sufficient for $T$ given $\hat{T}$, i.e., $S \perp T(X) \mid \hat{T}(X)$. In this case, representation-conditional coverage transfers to true-group coverage. However, the condition, $S \perp T(X) \mid \hat{T}(X)$, is generally not hold, since score $S$ depends on $(X, Y)$ and $X$ depends on the unobserved variable $W$ beyond $\hat{T}$. It does hold under the ideal alignment $\hat{T}(X) = T(X)$ a.s., which is the setting in Theorem 2. The condition $\hat{T}(X) = T(X)$ a.s. holds when $\hat{T}(X)$ is essentially the Bayes-optimal argmax classifier or a marginal condition holds as shown in Lemma 10.

$\blacksquare$

**Remark 7** *While the RKHS class $\mathcal{F}^*$ is specified using the estimated low-rank embedding $\hat{\pi}(\cdot)$, the tilt function $f = \mathbf{1}\{\hat{T}(X) = k\}$ used for group-conditional coverage must be fixed with respect to the calibration sample. To ensure this, the low-rank projection $\hat{\pi}(\cdot)$ and the induced partition $\hat{T}(\cdot)$ are computed using an algorithm that is invariant under re-orderings of the input data. In this way, $f$ is data-dependent but non-random relative to the calibration set and test point, which preserves the exchangeability conditions for our finite-sample guarantee.*

*This construction can be viewed as an empirical proxy for a population-level tilt determined by the latent structure. The posterior mean embedding $\pi(X) = \mathbb{E}[W \mid X]$ serves as the ideal population target, as it is a deterministic, stable summary of the latent variable $W$, thus yielding the most efficient prediction sets under the settings in Theorem 2. Our estimated embedding $\hat{\pi}(\cdot)$ and groups $\hat{T}(\cdot)$ approximate this structure from data. From the Corollary 6 and the adaptivity of the tilt function $f$ to $\hat{\pi}(\cdot)$, the coverage guarantee remains robust to the errors in $\hat{\pi}(\cdot)$.*

### C.4 SOME TECHNICAL PROOFS

**Lemma 8** *Fix $K \geq 2$ and let $\pi_i = (\pi_{ik})_{k=1}^{K}$ be the true embedding representative and $\hat{\pi}_i = (\hat{\pi}_{ik})_{k=1}^{K}$ with $\pi_{ik}, \hat{\pi}_{ik} \in (0, 1)$. Define*

$$\theta_{ik} := \log \pi_{ik} - \frac{1}{K} \sum_{\ell=1}^{K} \log \pi_{i\ell}, \qquad \hat{\theta}_{ik} := \log \hat{\pi}_{ik} - \frac{1}{K} \sum_{\ell=1}^{K} \log \hat{\pi}_{i\ell},$$

*and write vectors $\theta_i = (\theta_{ik})_{k=1}^{K}$, $\hat{\theta}_i = (\hat{\theta}_{ik})_{k=1}^{K}$. Let $r_{ik} := \pi_{ik} - \hat{\pi}_{ik}$ and $\Delta\pi_{ik} := r_{ik}/\hat{\pi}_{ik}$, and define the centered vector*

$$\tilde{\Delta}\pi_{ik} := \Delta\pi_{ik} - \frac{1}{K} \sum_{\ell=1}^{K} \Delta\pi_{i\ell} \quad (k = 1, \ldots, K), \qquad \tilde{\Delta}\pi_i := (\tilde{\Delta}\pi_{ik})_{k=1}^{K}.$$

*Let $\Delta_{1,ij} = 2\langle \hat{\theta}_i - \hat{\theta}_j, \, \tilde{\Delta}\pi_i - \tilde{\Delta}\pi_j \rangle + \|\tilde{\Delta}\pi_i - \tilde{\Delta}\pi_j\|_2^2$. Assume $\max_{i,k} |\Delta\pi_{ik}| \leq \frac{1}{2}$, then*

$$\|\theta_i - \theta_j\|_2^2 - \|\hat{\theta}_i - \hat{\theta}_j\|_2^2 = \Delta_{1,ij} + \Delta_{2,ij}, \tag{60}$$

*for an absolute constant $C$ with $|\Delta_{2,ij}| \leq C\Big( \big( \max_k |\Delta\pi_{ik}| \big)^3 + \big( \max_k |\Delta\pi_{jk}| \big)^3 \Big)$. For the Gaussian kernel $\psi_{ij} := \exp\big( -\gamma \|\theta_i - \theta_j\|_2^2 \big)$ and $\hat{\psi}_{ij} := \exp\big( -\gamma \|\hat{\theta}_i - \hat{\theta}_j\|_2^2 \big)$, we have*

$$\psi_{ij} = \hat{\psi}_{ij} \big( 1 - \gamma \Delta_{1,ij} + O\big( |\Delta_{2,ij}| + \gamma^2 \Delta_{1,ij}^2 \big) \big). \tag{61}$$

**Proof.** Write $\pi_{ik} = \hat{\pi}_{ik}(1 + \Delta\pi_{ik})$. Then

$$\theta_{ik} - \hat{\theta}_{ik} = \log(1 + \Delta\pi_{ik}) - \frac{1}{K}\sum_{\ell=1}^K \log(1 + \Delta\pi_{i\ell}).$$

For $|u| \leq \frac{1}{2}$, $\log(1 + u) = u - \frac{1}{2}u^2 + r(u)$ with $|r(u)| \leq 2|u|^3$. Hence

$$\theta_{ik} - \hat{\theta}_{ik} = \tilde{\Delta}\pi_{ik} - \frac{1}{2}\left( \Delta\pi_{ik}^2 - \frac{1}{K}\sum_{\ell=1}^K \Delta\pi_{i\ell}^2 \right) + \tilde{r}_{ik}, \qquad |\tilde{r}_{ik}| \leq 2\left( \max_k |\Delta\pi_{ik}| \right)^3.$$

Let $q_i := \theta_i - \hat{\theta}_i - \tilde{\Delta}\pi_i$ where $q_i$ collects the centered quadratic and remainder terms; then $\|q_i\|_2 \lesssim (\max_k |\Delta\pi_{ik}|)^2$. Consequently,

$$\theta_i - \theta_j = (\hat{\theta}_i - \hat{\theta}_j) + (\tilde{\Delta}\pi_i - \tilde{\Delta}\pi_j) + (q_i - q_j),$$

and expanding the squared norm yields equation 60 with

$$\Delta_{2,ij} = 2\langle \hat{\theta}_i - \hat{\theta}_j, \, q_i - q_j \rangle + 2\langle \tilde{\Delta}\pi_i - \tilde{\Delta}\pi_j, \, q_i - q_j \rangle + \|q_i - q_j\|_2^2,$$

which is bounded as stated by Cauchy–Schwarz and the displayed bounds on $q_i, q_j$.

For the kernels, write with $\hat{dist}_{ij} := \|\hat{\theta}_i - \hat{\theta}_j\|_2^2$ and $\Delta dist_{ij} := \|\theta_i - \theta_j\|_2^2 - \hat{dist}_{ij}$,

$$\psi_{ij} = \hat{\psi}_{ij} \exp(-\gamma \Delta dist_{ij}) = \hat{\psi}_{ij}\Big( 1 - \gamma \Delta dist_{ij} + O(\gamma^2 \Delta dist_{ij}^2) \Big),$$

and substitute equation 60 to obtain equation 61. ∎

**Lemma 9 (Kernel perturbation against a fixed value)** *Fix $K \geq 2$. Let $\Delta\pi_{ik} := (\pi_{ik} - \hat{\pi}_{ik})/\hat{\pi}_{ik}$ and the centered version $\tilde{\Delta}\pi_{ik}$ as in Lemma 8. Fix any $w \in \mathbb{R}^K$ and define the Gaussian kernels*

$$\psi_i(w) := \exp\big( -\gamma \|\theta_i - w\|_2^2 \big), \qquad \hat{\psi}_i(w) := \exp\big( -\gamma \|\hat{\theta}_i - w\|_2^2 \big).$$

*Assume $\max_k |\Delta\pi_{ik}| \leq \frac{1}{2}$. Then, writing $\Delta_{1,i}(w) := 2\langle \hat{\theta}_i - w, \, \tilde{\Delta}\pi_i \rangle + \|\tilde{\Delta}\pi_i\|_2^2$, we have the distance expansion*

$$\|\theta_i - w\|_2^2 - \|\hat{\theta}_i - w\|_2^2 = \Delta_{1,i}(w) + \Delta_{2,i}, \qquad |\Delta_{2,i}| \leq C\Big( \max_k |\Delta\pi_{ik}| \Big)^3, \tag{62}$$

*for an absolute constant $C$. Consequently,*

$$\Delta_i(w) := \big| \psi_i(w) - \hat{\psi}_i(w) \big| \leq \hat{\psi}_i(w)\left( \gamma \big| \Delta_{1,i}(w) \big| + C\big( (\max_k |\Delta\pi_{ik}|)^3 + \gamma^2 \Delta_{1,i}(w)^2 \big) \right). \tag{63}$$

**Proof.** Write $\pi_{ik} = \hat{\pi}_{ik}(1 + \Delta\pi_{ik})$. Using the proof in Lemma 8, we have now

$$\|\theta_i - w\|_2^2 - \|\hat{\theta}_i - w\|_2^2 = 2\langle \hat{\theta}_i - w, \tilde{\Delta}\pi_i \rangle + \|\tilde{\Delta}\pi_i\|_2^2 + \Delta_{2,i},$$

with

$$\Delta_{2,i} = 2\langle \hat{\theta}_i - w, q_i \rangle + 2\langle \tilde{\Delta}\pi_i, q_i \rangle + \|q_i\|_2^2,$$

which obeys $|\Delta_{2,i}| \leq C(\max_k |\Delta\pi_{ik}|)^3$ by Cauchy–Schwarz and the bounds on $q_i$. This proves equation 62. For the kernels, let $\hat{dist}_i(w) := \|\hat{\theta}_i - w\|_2^2$ and $\Delta dist_i(w) := \|\theta_i - w\|_2^2 - \hat{dist}_i(w)$. Then

$$\psi_i(w) = \exp\big( -\gamma(\hat{dist}_i(w) + \Delta dist_i(w)) \big) = \hat{\psi}_i(w)\Big( 1 - \gamma \Delta dist_i(w) + O(\gamma^2 \Delta dist_i(w)^2) \Big).$$

Substitute equation 62 for $\Delta dist_i(w)$ and take absolute values to obtain equation 63. ∎

**Lemma 10** *Let $m(X) := \pi_{(1)}(X) - \pi_{(2)}(X)$ be the (pointwise) top-1 margin, where $\pi_{(1)} \geq \pi_{(2)} \geq \cdots$ are the order statistics of $\{\pi_k(X)\}_{k=1}^K$. If*

$$\|\hat{\pi}(X) - \pi(X)\|_\infty < \tfrac{1}{2} m(X) \quad a.s.,$$

*then $\hat{T}(X) = T(X)$ a.s.*

**Proof.** Let $k := T(X)$, so $\pi_k(X) - \pi_\ell(X) \geq m(X)$ for all $\ell \neq a$. Then

$$\hat{\pi}_k - \hat{\pi}_\ell = (\pi_k - \pi_\ell) + (\hat{\pi}_k - \pi_k) - (\hat{\pi}_\ell - \pi_\ell) \geq m(X) - 2\|\hat{\pi} - \pi\|_\infty > 0,$$

so $\hat{T}(X) = k$. ∎

### C.4.1 Approximate conditional validity under embedding error

**Lemma 11** *Let $W'$ be drawn according to the true neighborhood law $W' \mid \pi(X_{n+1}) \sim \psi_W^*(\pi(X_{n+1}), \cdot)$. Assume the conditions in Lemma 9 are all satisfied, then*

$$\mathbb{P}(Y_{n+1} \in \hat{C}_{rand}^*(X_{n+1}) \mid W' = w') = 1 - \alpha - \frac{\mathbb{E}[\sum_{i \in [n+1]} \hat{\upsilon}_{S^{rand},i} \psi^*(w', \hat{\pi}(X_i))]}{\mathbb{E}[\psi^*(w', \hat{\pi}(X))]} + Err(w'). \tag{64}$$

*where*

$$|Err(w')| \leq \frac{\Delta(w')}{\mathbb{E}[\psi_W^*(\pi(X), w')]} + \frac{\mathbb{E}[\psi_W^*(\hat{\pi}(X), w')]}{\mathbb{E}[\psi_W^*(\pi(X), w')]} \cdot \frac{\Delta(w')}{\mathbb{E}[\psi_W^*(\hat{\pi}(X), w')]}, \tag{65}$$

*with $\Delta_i(w') = \left|\psi_W^*(\hat{\pi}(X_i), w') - \psi_W^*(\pi(X_i), w')\right|$. and $\Delta(w') := \mathbb{E}[\Delta_i(w')]$.*

**Proof.** Starting from the displayed decomposition in Theorem 1,

$$\mathbb{E}\left[\mathbf{1}\{Y_{n+1} \in \hat{C}_{rand}^*(X_{n+1})\} - (1 - \alpha) \mid W'\right]$$

$$= \frac{\mathbb{E}\left[\psi_W^*(\pi(X_{n+1}), W') \cdot (\mathbf{1}\{Y_{n+1} \in \hat{C}_{rand}^*(X_{n+1})\} - (1 - \alpha))\right]}{\mathbb{E}[\psi_W^*(\pi(X), W')]}.$$

If we replace the true $\pi(X_i)$ by the estimated $\hat{\pi}(X_i)$, define $N(W') := \mathbb{E}[\psi_W^*(\pi(X), W') \cdot Z(X, Y)]$, $D(W') := \mathbb{E}[\psi_W^*(\pi(X), W')]$, and $\hat{N}(W') := \mathbb{E}[\psi_W^*(\hat{\pi}(X), W') \cdot Z(X, Y)]$, $\hat{D}(W') := \mathbb{E}[\psi_W^*(\hat{\pi}(X), W')]$ with $Z(X, Y) := \mathbf{1}\{Y \in \hat{C}_{rand}^*(X)\} - (1 - \alpha) \in [-1, 1]$. A standard ratio perturbation yields

$$\left|\frac{\hat{N}(W')}{\hat{D}(W')} - \frac{N(W')}{D(W')}\right| \leq \frac{|\hat{N}(W') - N(W')|}{D(W')} + \frac{|\hat{N}(W')|}{\hat{D}(W')} \cdot \frac{|\hat{D}(W') - D(W')|}{D(W')},$$

since $D(W'), \hat{D}(W') > 0$. Next, with $\Delta_i(W') := \left|\psi_W^*(\hat{\pi}(X_i), W') - \psi_W^*(\pi(X_i), W')\right|$ and $\Delta(W') := \mathbb{E}[\Delta_X(W')]$, we have

$$|\hat{N}(W') - N(W')| = \left|\mathbb{E}\left[(\psi_W^*(\hat{\pi}(X), W') - \psi_W^*(\pi(X), W')) Z(X, Y)\right]\right| \leq \mathbb{E}[\Delta_X(W')] = \Delta(W'),$$

and similarly $|\hat{D}(W') - D(W')| \leq \Delta(W')$. Using $|\hat{N}(W')| \leq \hat{D}(W')$ (because $|Z| \leq 1$) gives

$$\mathbb{E}\left[\mathbf{1}\{Y_{n+1} \in \hat{C}_{rand}^*(X_{n+1})\} - (1 - \alpha) \mid W'\right]$$

$$= \frac{\mathbb{E}\left[\psi_W^*(\hat{\pi}(X_{n+1}), W') \cdot (\mathbf{1}\{Y_{n+1} \in \hat{C}_{rand}^*(X_{n+1})\} - (1 - \alpha))\right]}{\mathbb{E}[\psi_W^*(\hat{\pi}(X), W')]} + Err(W')$$

$$= \frac{-\mathbb{E}\left[\sum_{i \in [n+1]} \hat{\upsilon}_{S^{rand},i} \psi_W^*(\hat{\pi}(X), W')\right]}{\mathbb{E}[\psi_W^*(\hat{\pi}(X), W')]} + Err(W'), \quad \text{Lemma 4}$$

where the general bound is in equation 65. If $\Delta_i(w') \to 0$, then $Err(w') \to 0$ as well. Therefore, equation 10 closely approximates the conditional guarantee with respect to the true latent representation. ∎

### C.4.2 COVERAGE GAP ESTIMATION

The idea behind estimating the coverage gap $\hat{\lambda} \frac{\mathbb{E}\left[\langle \hat{g}_{rand,\psi^*}, f_{\psi^*} \rangle_{\psi^*}\right]}{\mathbb{E}[f(X)]}$ is to leverage results from $n$-sample quantile regression, applied specifically to the calibration data points. As shown in Proposition 2 of Gibbs et al. (2023), the estimation error in their setting (using raw covariates) can be bounded by $O(\sqrt{d \log n / n})$. We adapt their arguments to the latent-space setting, where the feature map satisfies $\|\Phi^*(X)\|_1 = 1$. The following result, Proposition 4, provides a sharper bound on this estimation error under our setting.

To simplify the notation, let

$$\mathcal{L}_n(g_{\psi^*}, \eta) := \frac{1}{n} \sum_{i \in [n]} \ell_\alpha(S_i - \Phi^*(X_i)^\top \eta - g_{\psi^*}(X_i))$$

$$\mathcal{L}_\infty(g_{\psi^*}, \eta) := \mathbb{E}\left[\ell_\alpha(S_i - \Phi^*(X_i)^\top \eta - g_{\psi^*}(X_i))\right]$$

denote the empirical and population losses with low-rank projection $\hat{\pi}(\cdot)$.

Recall the closed form solution in equation 6 shows the estimated coefficients are functions of $\lambda$. For a fixed $\lambda$, we denote the solution class parameterized by $\lambda$ as

$$\mathcal{F}_{\lambda,\psi^*} = \{g_{\psi^*} : g_{\psi^*}(x) = \frac{1}{\lambda} \sum_{i \in [n+1]} \upsilon_i \psi^*(x, X_i)\} \tag{66}$$

Define the objective

$$\tilde{\mathcal{L}}_n(g_{\psi^*}, \eta) := \mathcal{L}_n(g_{\psi^*}, \eta) + \lambda \cdot \|g_{\psi^*}\|_{\psi^*}^2$$

$$\tilde{\mathcal{L}}_\infty(g_{\psi^*}, \eta) := \mathcal{L}_\infty(g_{\psi^*}, \eta) + \lambda \cdot \|g_{\psi^*}\|_{\psi^*}^2$$

which is strictly convex in $g_{\psi^*}$ and $\eta$. Let $(\hat{g}_{n,\psi^*}, \mathcal{B}_n), (g^*_{\infty,\psi^*}, \mathcal{B}^*_\infty) \in \mathcal{F}_{\lambda,\psi^*} \times 2^{\mathbb{R}^K}$, denote the minimizers of $\min_{(g_{\psi^*}, \eta) \in \mathcal{F}^*} \tilde{\mathcal{L}}_n(g_{\psi^*}, \eta), \min_{(g_{\psi^*}, \eta) \in \mathcal{F}^*} \tilde{\mathcal{L}}_\infty(g_{\psi^*}, \eta)$, respectively.

Note we write $g(x) = \Phi^*(x)^\top \eta + g_{\psi^*}(x)$ with arbitrary $(g_{\psi^*}, \eta)$. Let

$$g^*_\infty(x) = \Phi^*(x)^\top \eta^*_\infty + g^*_{\infty,\psi^*}(x) \text{ for } \eta^*_\infty \in \mathcal{B}^*_\infty$$

$$\hat{g}_n(x) = \Phi^*(x)^\top \hat{\eta}_n + \hat{g}_{n,\psi^*}(x) \text{ for } \hat{\eta}_n \in \mathcal{B}_n$$

Let $e(g, g^*_\infty) = \tilde{\mathcal{L}}_\infty(g_{\psi^*}, \eta) - \tilde{\mathcal{L}}_\infty(g^*_{\infty,\psi^*}, \mathcal{P}_{\mathcal{B}^*_\infty} \eta)$.

**Assumption 3 (Population strong convexity)** *Let $d(g_{\psi^*}, \eta) := \inf_{\eta^*_\infty \in \mathcal{B}^*_\infty} \|\eta - \eta^*_\infty\|_2 + \|g_{\psi^*} - g^*_{\infty,\psi^*}\|_{\psi^*}$ denote the distance from $(g_{\psi^*}, \eta)$ to the nearest population minimizer. Suppose $S \mid X$ has positive density on $\mathbb{R}$ and is continuous. If $d(g_{\psi^*}, \eta) \leq \epsilon_l$ for some constant $\epsilon_l > 0$, then there exists some constant $C_l > 0$ such that*

$$e(g, g^*_\infty) \geq C_l d(g_{\psi^*}, \eta)^2$$

This assumption is mild under the some assumptions on the distribution of $S \mid X$ since $\nabla^2_\eta \mathcal{L}_\infty = \mathbb{E}[P_{S|X}(0)XX^\top]$ Tan et al. (2022).

**Assumption 4** *There exist some constants $c_f, c_\pi, c_{f,S} > 0$ such that*

$$\sup_{f \in \mathcal{F}^*} \sqrt{\mathbb{E}[|f(X_i)|^2]} \leq c_f \mathbb{E}\left[|f(X)|\right], \quad \mathbb{E}[|f(X_i)|S_i^2] \leq c_{f,S} \mathbb{E}\left[|f(X)|\right]$$

$$\inf_{\eta:\|\eta\|_2=1, \eta \in \mathbb{R}^d} \mathbb{E}[|\Phi^*(X)^\top \eta|] \geq c_\pi, \mathbb{E}[\|\Phi^*(X_i)\|_2^2] \leq c_0,$$

$$\sup_{f \in \mathcal{F}^*} \mathbb{E}[|f(X_i)|\|\Phi^*(X_i)\|_2^2] \leq c_1 \mathbb{E}[|f(X_i)|].$$

*Furthermore, we assume that $\mathbb{E}[|S_i^2|] < \infty$ and $\sup_x \psi^*(x,x) = 1$.*

This assumption is stronger than Assumption 1 in Gibbs et al. (2023), which requires the following moment bounds:

$$\mathbb{E}[\|\Phi^*(X_i)\|_2^2] \leq c_0 d, \ \sup_{f \in \mathcal{F}^*} \mathbb{E}[|f(X_i)|\|\Phi^*(X_i)\|_2^2] \leq c_1 \mathbb{E}[|f(X_i)|]d$$

In contrast, we assume a bounded-norm feature map in the latent space, specifically $\|\Phi^*(X)\|_2^2 \leq c_0$ which does not grows with feature dimension $d$. In particular, when $\Phi^*(X)$ is an indicator vector over a finite partition, in which case $\|\Phi^*(X)\|_1 = 1$ as well.

**Proposition 4** *Suppose the assumptions 3, 4 are satisfied. Under the settings in Lemma 4. Define the $n$-sample kernel quantile regression estimate with a fixed $\lambda$*

$$(\hat{g}_{n,\psi^*}, \hat{\eta}_n) = \arg \min_{g_{\psi^*} \in \mathcal{F}_{\lambda,\psi^*}, \eta \in \mathbb{R}^K} \frac{1}{n} \sum_{i \in [n]} \ell_\alpha(S_i - g_{\psi^*}(X_i) - \Phi^*(X_i)^\top \eta) + \lambda \|g_{\psi^*}\|_{\psi^*}^2,$$

*and for any $\epsilon > 0$, let*

$$\mathcal{F}_\epsilon^* := \{f(\cdot) = f_{\psi^*}(\cdot) + \Phi^*(\cdot)^\top \eta \in \mathcal{F}^* : \|f_{\psi^*}\|_{\psi^*} + \|\eta\|_2 \leq 1, \mathbb{E}[|f(X)|] \geq \epsilon\}.$$

*Then,*

$$\sup_{f \in \mathcal{F}_\epsilon^*} \left| 2\lambda \frac{\langle \hat{g}_{n,\psi^*}, f_{\psi^*} \rangle_{\psi^*}}{\frac{1}{n} \sum_{i=1} |f(X_i)|} - 2\lambda \frac{\mathbb{E}[\langle \hat{g}_{S_{n+1},\psi^*}, f_{\psi^*} \rangle_{\psi^*}]}{\mathbb{E}[\frac{1}{n} \sum_{i=1} |f(X_i)|]} \right| \leq O_\mathbb{P}\left( \sqrt{\frac{\log n}{n}} \right)$$

**Proof.** By the results in Section 4.1.2 in Boucheron et al. (2005) and $\|\Phi^*(X)\|_2^2 \leq c_0$, we know that $\{f_{\psi^*}(\cdot) + \Phi^*(\cdot)\eta : \|f_{\psi^*}\|_{\psi^*} + \|\eta\|_2 \leq 1\}$ has Rademacher complexity at most $O(\sqrt{1/n})$. Following the proof for Proposition 2 in Gibbs et al. (2023), we can show

1. Let $\mathcal{E}_2 = \{\|\eta - \mathcal{P}_{\mathcal{B}_\infty^*} \eta\|_2 \leq \epsilon_1, \|g_{\psi^*} - g_{\infty,\psi^*}^*\|_{\psi^*} \leq \epsilon_2 : \epsilon_1, \epsilon_2 > 0\}$. We have

$$\mathbb{E}\left\{ \sup_{\eta, g_{\psi^*} \in \mathcal{E}_2} |\mathcal{L}_n(g_{\psi^*}, \eta) - \mathcal{L}_n(g_{\infty,\psi^*}^*, \mathcal{P}_{\mathcal{B}_\infty^*} \eta) - \left( \mathcal{L}_\infty(g_{\psi^*}, \eta) - \mathcal{L}_\infty(g_{\infty,\psi^*}^*, \mathcal{P}_{\mathcal{B}_\infty^*} \eta) \right)| \right\}$$

$$\leq O((\epsilon_1 + \epsilon_2)\sqrt{1/n})$$

2. $\sup_{f \in \mathcal{F}_\epsilon^*} \left| \frac{1}{n} \sum_{i \in [n]} f(X_i) - \mathbb{E}[\frac{1}{n} \sum_{i \in [n]} |f(X_i)|] \right| = O_\mathbb{P}(\sqrt{1/n})$

3. $\sup_{f_{\psi^*} \in \mathcal{F}_{\lambda,\psi^*}} \lambda \left| \mathbb{E}[\langle \hat{g}_{S_{n+1},\psi^*}, f_{\psi^*} \rangle_{\psi^*}] \right| = O(1)$

4. $\sup_{f_{\psi^*} \in \mathcal{F}_{\lambda,\psi^*} : \|f_{\psi^*}\|_{\psi^*} \leq 1} \lambda \left| \langle \hat{g}_{n,\psi^*}, f_{\psi^*} \rangle_{\psi^*} - \mathbb{E}[\langle \hat{g}_{S_{n+1},\psi^*}, f_{\psi^*} \rangle_{\psi^*}] \right| \leq O_\mathbb{P}(\sqrt{\frac{\log(n)}{n}})$

Using the claims above, we thus get the desired results through some calculations. ∎

**Remark 12** *Under the setting in Theorem 1, the tilt function $f^{w'}(x) = \psi_W^*(\hat{\pi}(x), w')$ emphasizes coverage in a neighborhood around the fixed point $w'$ in the latent space. As shown in Proposition 4, this coverage gap $\frac{\mathbb{E}[\sum_{i \in [n+1]} \hat{v}_{srand,i} \psi_W^*(W', \hat{\pi}(X_i))]}{\mathbb{E}[\psi_W^*(W', \hat{\pi}(X))]}$ admits a data-driven approximation $\frac{\sum_{i \in [n]} \hat{v}_{n,i} \psi_W^*(w', \hat{\pi}(X_i))}{\frac{1}{n} \sum_{i \in [n]} \psi_W^*(w', \hat{\pi}(X_i))}$ where $W' = w'$ is fixed and $\{\hat{v}_{n,i}\}_{i \in [n]}$ are the empirical coefficients from $\hat{g}_{n,\psi^*}$.*

# D ADDITIONAL EXPERIMENTS

## D.1 SYNTHETIC EXPERIMENTS

In this section, we provide additional details on the synthetic experiments and provide further experiment results.

In all of our experiments, we generate $X_i \in \mathbb{R}^p$ from a mixture of $K = 3$ latent distributions. Specifically, we first generate $X_i$ from a multinomial distribution, $mX_i \sim \text{Multinomial}(N, \sum_{k \in [K]} W_i(k)\zeta_k)$ with $W_i = w_i$ fixed and total count $N = 1000$. For each sample in the training and calibration sets, we generate $W_i \sim Dir([2,1,1])$ and randomly shuffle the elements to create a distribution that is more symmetric across vertices. Here, the density is higher in the central part of the simplex. For test samples, we generate from the same distribution but do not shuffle, to create a high concentration near one vertex of the 2-dimensional simplex (Figure 5).

We sample the latent component $\zeta_k \in \mathbb{R}^p$ from a uniform distribution and normalize it so that $\sum_{j \in [p]} \zeta_k(j) = 1$ for each $k \in [K]$. We estimate $\pi(X_i) = \mathbb{E}[W_i \mid X_i]$ with pLSI (Section B.1) and use $\hat{\pi}(X_i)$ as inputs of SpeedCP, CondCP, PCP, and RLCP. For SpeedCP and CondCP, we choose $\Phi^*(X_i) = (1, \mathbf{1}\{\arg\max_k \hat{\pi}_k(X_i) = 1\}, \ldots, \mathbf{1}\{\arg\max_k \hat{\pi}_k(X_i) = K-1\})$ using the estimated latent embeddings $\hat{\pi}(X))^\top$. The response $Y_i$ is generated from a nonlinear function of $Y_i \sim N(\sin(2\pi \cdot W_i(1)) + (W_i(2))^2 + W_i^\top \eta, 0.1^2)$ and $\eta_j \sim Unif(1, 10)$ for $j = 1, 2, 3$ and normalized. In this setting, we aim to see whether each conformal method can guarantee 0.9 coverage uniformly across the simplex, especially in boundaries (areas close to one vertex). We report our results based on 50 independent runs of data generation. At each run, we split the data into 600 training points, 300 calibration points, and 100 test points.

In Table 2, we report the computation time for two different predictors. We can see that SpeedCP is faster compared to CondCP and PCP, which are the state-of-the-art conformal prediction methods that account for the local or latent data structure. RLCP is fast but fails to attain target miscoverage level as discussed in Section 3 of the main manuscript. In Figure 4, we show the coverage conditional on the latent space of $\hat{\pi}(X)$ when the predictor is a neural network. The same plot for the linear regression predictor is shown in Figure 1. In both plots, we observe that SpeedCP achieves 0.9 across the simplex most uniformly.

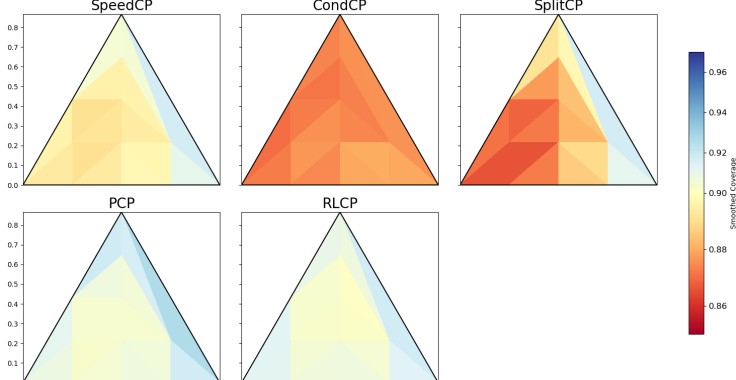

Figure 4: Mean coverage on fine-gridded partitions on the latent space (a 2D simplex) when $\hat{\mu} = $ NN. The results are aggregated over 50 random generations. SpeedCP shows the most uniform 0.9 (pale yellow) coverage across the simplex.

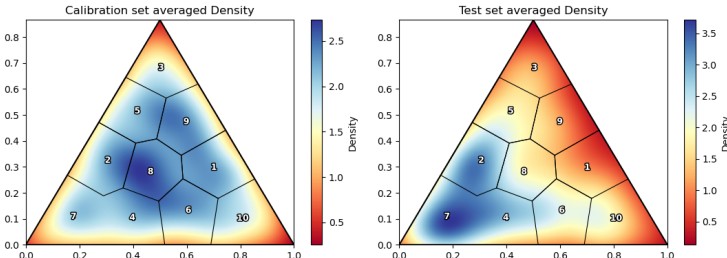

Figure 5: Averaged calibration and test density over 50 random generations of data. We use kmeans followed by Voronoi tessellation to partition the latent simplex into 10 bins.

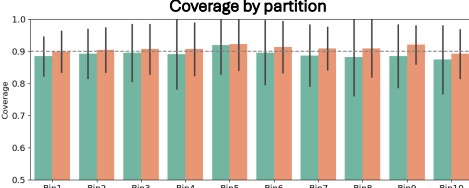
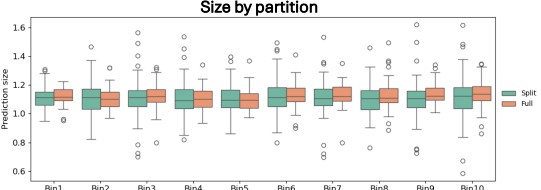

Figure 6: Conditional miscoverage and prediction set size for each fixed partition on the latent space. We observe small overestimation of coverage when we use the full calibration set instead of setting aside a dataset for tuning $(\gamma, \lambda)$. Overall, the two methods are marginally different.

### D.1.1 COVERAGE ACROSS DIFFERENT SAMPLE SIZE $n$

We experiment with different values of the sample size $n$ in Table 5. As the calibration set grows larger, we generally expect the coverage guarantee to be the same while the prediction set size decreases because the estimation error and the uncertainty of the kernel quantile estimator diminishes. Interestingly, we observe a slight increase in prediction size at $n = 2000$ followed by a decrease at n=5000. This fluctuation is likely due to finite-sample variability in both the estimated latent embeddings and the cross-validated hyperparameters. The computation time increases with $n$, which is consistent with the computational complexity of SpeedCP, which is approximately $O(n^3)$. Overall, for moderately large $n$, the method still remains computationally feasible.

Table 5: Mean prediction set size and computation time for SpeedCP (linear regression predictor).

| $n$ | Prediction set size | Computation time (seconds) |
|---|---|---|
| 1000 | $1.376 \pm 0.07$ | $12.363 \pm 4.10$ |
| 2000 | $2.053 \pm 0.11$ | $23.956 \pm 7.03$ |
| 5000 | $1.030 \pm 0.02$ | $191.255 \pm 29.36$ |

### D.1.2 CHOICES OF DIFFERENT $\Phi^*(X)$

We also discuss how conditional coverage changes with different choices of $\Phi^*(X)$ of our function class $\mathcal{F}^*$ equation 17. When running a RKHS-based quantile regression on the scores, $\Phi^*(X)^\top \eta$ acts as the linear component with the design matrix $\Phi^*(X)$ and parameters $\eta$. $\Phi^*(\cdot)$ allows flexible modeling of different types of conditional coverage. For example, in this synthetic experiment, we can consider four different $\Phi^*(X)$ based on the estimated latent embedding $\hat{\pi}(X)$,

1. Taking $\Phi^*(X) = 1$ yields the marginal coverage.

2. Taking $\Phi^*(X) = \hat{\pi}(X)$ yields mixture- conditional coverage, where we guarantee coverage linearly reweighted with $\hat{\pi}(X)$.

3. (What we used) Taking $\Phi^*(X) = (\mathbf{1}, \mathbf{1}\{\hat{T}(X) = 1\}, \ldots, \mathbf{1}\{\hat{T}(X) = K - 1\})^\top$ or said $\Phi^*(X) = (\mathbf{1}\{\hat{T}(X) = 1\}, \ldots, \mathbf{1}\{\hat{T}(X) = K\})^\top$ where $\hat{T}(X) = \arg\max_{k \in [K]} \hat{\pi}_k(X)$ yields topic-conditional coverage, where the topic is defined as the latent distribution with the highest mixture proportion weight.

Through our experiments we observed that in high-dimensional settings, coverage using SpeedCP is primarily affected by the RKHS component, $f_{\Psi^*}$ rather than the linear term. If more prior information is available on the conditional distribution, and the goal is to achieve more precise conditional coverage at level $1 - \alpha$, one may instead calibrate scores using a function class restricted to the linear term, as in Gibbs et al. (2023). However, the inclusion of the RKHS component can lead to smaller prediction sets even without those additional prior structures. Further investigation is needed to determine whether choosing $\Phi^*(X)$ as the indicators of topics, or the latent embeddings, improves performance under varying covariate dimensionality $p$ or the signal-to-noise ratio in $X$.

### D.1.3 USING CALIBRATION SET FOR TUNING $(\gamma, \lambda)$

In our experiments, we use the calibration set for selection of $(\gamma, \lambda)$ instead of setting aside a separate validation set for efficiency. We agree that, in the current implementation, $(\gamma, \lambda)$ is selected by cross-validation using the calibration data, so the chosen pair is technically data-dependent. Therefore, the dependence will introduce some bias and small finite-sample distortion of coverage. However, the selected pair $(\lambda, \gamma)$ converges in probability to a deterministic value when the calibration set size is large. Recent work on adaptive coverage policies shows that it is empirically valid to use the calibration set itself to select regularization parameters via leave-one-out or cross-validation (Theorem 2.6 in Gauthier et al. (2025)). Gibbs et al. (2023) also show, the selection of $\lambda$ using the calibration set does not affect the coverage significantly.

To assess whether the calibration set can be reliably used for hyperparameter tuning, we compare it against a split strategy in which half of the calibration set is used for tuning and the remaining half for calibration, avoiding the potential issues discussed earlier. As shown in Figure 6, the two approaches have similar conditional coverage and prediction set sizes, with the full-calibration procedure exhibiting only a slight overestimation of coverage. Also in Figure 7, we observe that the chosen $(\gamma, \lambda)$ pairs from the two approaches are also similar.

### D.1.4 UNIFORM COVERAGE ON ANY $(\gamma, \lambda)$

We assess whether the uniform coverage guarantee assumed in Section 2.2 holds across all choices of $(\gamma, \lambda)$. We get the approximate joint hyperparameter by gathering results from running $\lambda$-path on each $\gamma$ in the $\gamma$ grid. We then select 25 pairs and run $S$-path to get coverage and prediction set size. We observe in Figure 7 that the coverage holds uniformly 0.9 across the pairs, affirming that the coverage is not affected by the choice of $(\gamma, \lambda)$. However, we observe that the prediction set size differs by the choice of $(\gamma, \lambda)$. Our cross validation approach chooses $(\gamma, \lambda)$ in the region where prediction set size is small.

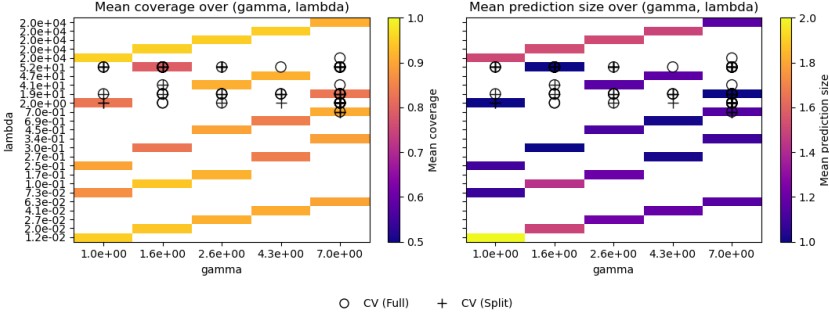

Figure 7: Marginal coverage and prediction set size of 25 pairs of $(\gamma, \lambda)$ on the joint hyperparameter space. We also show selected $(\gamma, \lambda)$'s using $k$-fold cross validation on calibration set (full) or validation set (split).

### D.1.5 EFFECT OF RANDOMIZATION ON $S$

As described in Section 2, we use the randomized cutoff $S^{rand}(X_{n+1}) = \sup\{S \mid \hat{v}_{S,n+1} < U\}$, where $U \sim Unif(-\alpha, 1-\alpha)$, to construct prediction sets. In practice, this randomization introduces little variability. This is because the $S$-path starts with a small value, and along the path, the $S$ values along the path are generally smaller than $\hat{g}_S$, leading to $\hat{v}_{S,n+1} = -\alpha$. Figure 8, we observe that the standard deviation of $S_i^{rand}$'s for each run(seed) is small, confirming this behavior.

## D.2 REAL DATA EXPERIMENT

### D.2.1 ARXIV ABSTRACTS

We sample 5000 abstracts from ArXiv metadata (Clement et al., 2019) in mathematics, statistics, and computer science categories. The processed abstract-word count matrix has a vocabulary size

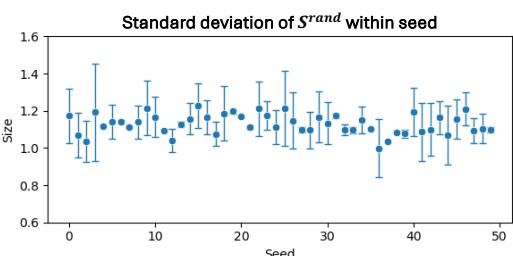

Figure 8: Mean and standard deviation of $S_i^{rand}$'s for each seed.

of $p = 11,516$. We project the abstracts onto $K = 5$ latent mixture proportions, $\hat{\pi}(X_i)$, using pLSI, the topic modeling approach described in Section B.1. We use $\hat{\pi}(X_i)$ as inputs for all methods.

For SpeedCP and CondCP, we additionally set the linear representation $\Phi^*(X_i)$ as an one-hot encoding of the topic: $\Phi^*(X) = \big( \mathbf{1}\{\hat{T}(X) = \texttt{Geometry}\}, \mathbf{1}\{\hat{T}(X) = \texttt{Algebra}\}, \mathbf{1}\{\hat{T}(X) = \texttt{ML}\}, \mathbf{1}\{\hat{T}(X) = \texttt{Vision}\}, \mathbf{1}\{\hat{T}(X) = \texttt{Quantum}\}\big)^\top$. Figure 9 displays the top words for each estimated topic, while Figure 10 shows the proportion of documents in each estimated topic. At a resolution of $K = 5$, the topics are readily interpretable and correspond to distinct subfields within mathematics, statistics, and computer science. pLSI estimates soft assignments $\hat{\pi}(X_i) \in \mathbb{R}^5$, representing mixture proportions over the topics, which we use as inputs to SpeedCP, CondCP, PCP, and RLCP.

The goal is to construct prediction intervals that achieve nominal level 0.9 across topics. CondCP is omitted because, in our experiments, it did not finish within the allotted time budget (30 hours). This occurred consistently across the larger datasets we evaluated. We present topic-conditional coverage and prediction set size in Table 6. To illustrate performance under a poor predictor, we choose linear regression of citation counts on raw word frequencies, which fails to extract any meaningful associations between words and citation counts. As a result, RLCP produces overly wide prediction intervals and PCP fails to uncover any latent mixture structure of $S|\hat{\pi}(X)$ and becomes equivalent to SplitCP. In contrast, SpeedCP leverages kernel smoothing, resulting in tighter and more accurate prediction intervals.

Table 6: Mean coverage across topics and prediction set size of ArXiv dataset.

| Method | Target coverage ($1 - \alpha = 0.9$) | | | | | Size | Time (seconds) |
|---|---|---|---|---|---|---|---|
| | Geometry | Algebra | ML | Vision | Quantum | | |
| **SpeedCP** | **0.880** $\pm$**0.02** | **0.890** $\pm$**0.05** | 0.730 $\pm$0.34 | **0.920** $\pm$**0.02** | 0.822 $\pm$0.11 | 15.835 $\pm$3.05 | 8.682 $\pm$3.10 |
| SplitCP | 0.877 $\pm$0.02 | 0.876 $\pm$0.04 | 0.659 $\pm$0.35 | 0.926 $\pm$0.02 | 0.762 $\pm$0.08 | 15.661 $\pm$1.17 | $< 0.01$ |
| PCP | 0.877 $\pm$0.02 | 0.876 $\pm$0.04 | 0.659 $\pm$0.35 | 0.926 $\pm$0.02 | 0.762 $\pm$0.08 | 15.661 $\pm$1.17 | 17.501 $\pm$0.54 |
| RLCP | 0.935 $\pm$0.02 | 0.958 $\pm$0.03 | **0.956** $\pm$**0.16** | 0.923 $\pm$0.02 | **0.962** $\pm$**0.04** | 42.493 $\pm$45.308 | 1.184 $\pm$0.01 |

### D.2.2 MOLECULE GRAPHS

We provide additional results of the molecule dataset example in Section 3. For each dataset, we subsample 2000 molecule graphs at each run with 50 runs in total, and split into 1000/500/500 training, calibration, and test points. Using the 1000 molecule graphs, we train a GIN predictor $\hat{\mu}(\cdot)$ to extract the 64-dimensional last layer and compute conformal scores $S_i = |\hat{\mu}(X_i) - y_i|$. In this experiment, we consider the intercept for the linear term, $\Phi^*(X_i) = 1$ and $\pi(X_i)$ as the PC score. In Figure 11, we plot the Voronoi partitions on which we measure the coverage (Figure 12) as well the mean prediction set (Figure 13) at level $\alpha = 0.1$.

Our method, SpeedCP, and SplitCP construct the smallest prediction sets overall. However, while SplitCP applies a single global cutoff across the entire PC space, SpeedCP adapts to the local structure of the embeddings. For instance, in the QM9 dataset we find that SpeedCP produces slightly larger prediction sets in sparser regions of the PC space (e.g., partitions 2, 4, and 6), which allows it to maintain consistent 0.9 coverage across all partitions.

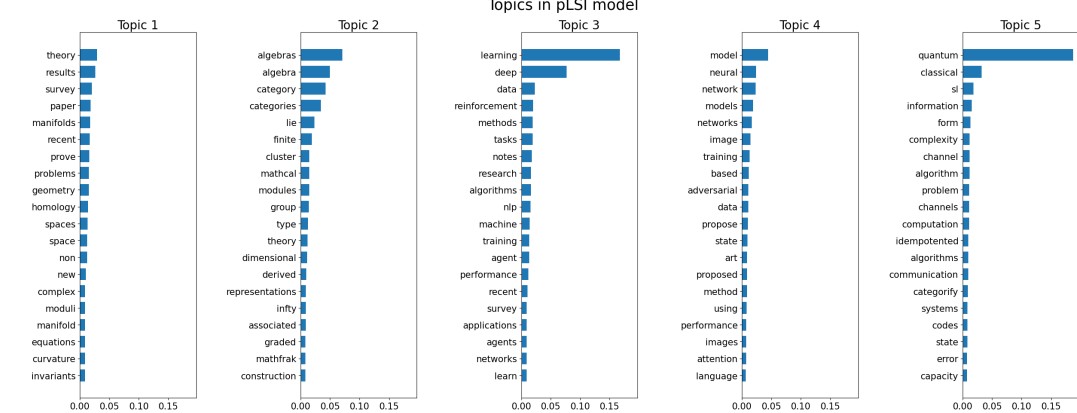

Figure 9: Latent topics of ArXiv abstracts identified by probabilistic latent semantic indexing (pLSI), a topic modeling approach. We plot the top 20 words with the largest weights for each topic. We name each topic as *Geometry, Algebra, Machine Learning, Computer Vision,* and *Quantum theory* based on the top words.

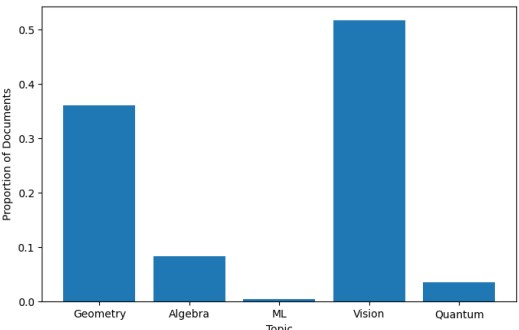

Figure 10: Distribution of the most likely topic over all abstracts with $n = 5000$.

### D.2.3 BRAIN TUMOR MRI

We train a CNN classifier $\hat{\mu}(\cdot)$ on 2,000 images and extract the 256-dimensional NN features from the last layer. We report the performance of the CNN classifier $\hat{\mu}(\cdot)$ in Figure 14, which shows the evaluation metrics on the training and validation sets.

Using another 2,400 images for calibration, we compute conformal scores $S_i = |\hat{\mu}(X_i) - y_i|$ and apply our RKHS path-following quantile regression in the latent space to obtain thresholds at level $\alpha = 0.1$.

In this experiment, we evaluate both *marginal* coverage and *per-label* (predicted-label) coverage $\mathbb{P}(Y_{n+1} \in \hat{C}^*_{\text{rand}}(X_{n+1}) \mid \hat{\mu}(X_{n+1}) = \hat{y})$ using 600 test images over 50 simulation trials. We exclude CondCP from the analysis because a single simulation takes over 50,000 seconds and the algorithm fails to converge. For comparison, we perform calibration using the 256-dimensional neural network features directly as the embedding $\hat{\pi}(\cdot)$. To further reduce dimensionality, we apply a post hoc PCA to rank 8 on these features; the resulting principal components define $\hat{\pi} : \mathcal{X} \to \mathbb{R}^8$.

**Using 256-dim features from NN.** We include illustrative results corresponding to Table 4 from the main paper. Empirically, the cutoffs produced by SplitCP and RLCP are effectively identical in our high-dimensional setting. Intuitively, RLCP's locality weights become uninformative in high dimensions (the distance metric loses discriminative power), so RLCP reduces to uniform weighting over the calibration set, recovering the SplitCP cutoff.

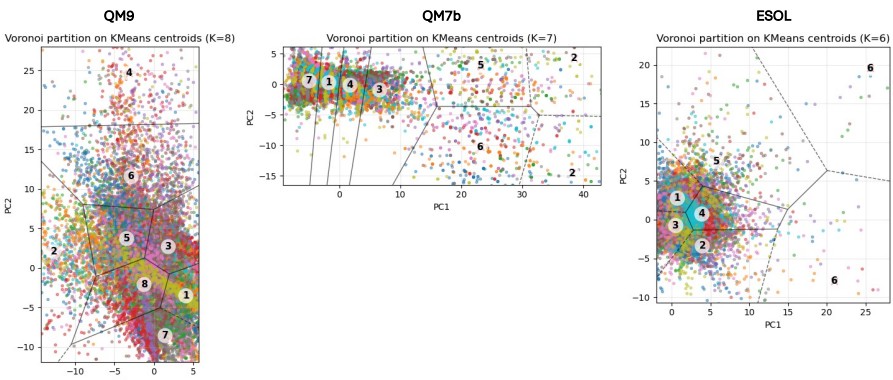

Figure 11: Voronoi tessellation of the PC space. We plot PC representation of graph embeddings where each color denotes each random subsample of the dataset.

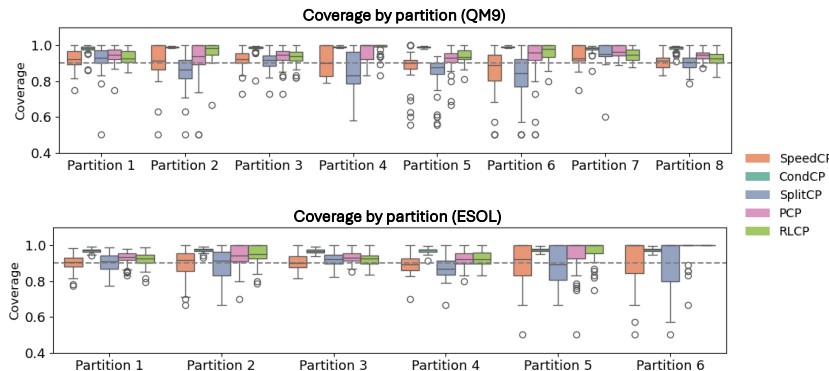

Figure 12: Coverage on fixed partitions of the PC space for QM9 and ESOL. We use PCA on the last layer embeddings of GNN with $K = 3$ dimensions. The dashed line denotes the target coverage rate $1 - \alpha = 0.9$.

**Using principle components.** To further reduce dimensionality, we extract features from the neural network and project them onto a low-rank embedding via PCA with $K = 8$, fitted on the first 2,000 training samples. SplitCP attains similar coverage but requires more conservative sets in lower-dimensional space, whereas our method delivers narrower sets with near-nominal predicted-label coverage. RLCP and PCP tend to over-cover, particularly for the healthy class as well, and exhibit unstable cutoffs with high variance and frequent near-zero values (see Table 9). Consequently, even after dimensionality reduction, RLCP and PCP produce overly conservative conditional coverage.

Compared to results using higher-dimensional features, the low-rank projection further reduces the cutoff without compromising conditional guarantees (comparing Table 4 with 8), thereby yielding narrower prediction sets.

### D.3 DETAILS ON COMPUTATION RESOURCES

All experiments were conducted on a cloud-based computing cluster. Each job was allocated 4 CPU cores and 4 GB of memory. No GPUs were used. For CondCP, we used the MOSEK solver in CVXPY to solve the underlying convex optimization problems. All code was implemented in Python3 and run in a consistent computing environment to ensure reproducibility.

Table 7: Summary statistics of conformal cutoffs (marginal and by predicted label) using the 256-dim features from NN as input space for conformal prediction.

| Method | Mean | Std | Min | Max |
|---|---|---|---|---|
| **Marginal** | | | | |
| **SpeedCP(1)** | 0.2662 | 0.0908 | 0.0012 | 0.9985 |
| **SpeedCP($\Phi^*$)** | 0.2828 | 0.0820 | 0.0025 | 0.9714 |
| SplitCP | 0.3482 | 0.0091 | 0.3271 | 0.3660 |
| RLCP | 0.3482 | 0.0091 | 0.3271 | 0.3660 |
| PCP | 0.2310 | 0.2899 | 0.0000 | 0.9984 |
| $\hat{\boldsymbol{y}} = $ healthy | | | | |
| **SpeedCP(1)** | 0.2500 | 0.0954 | 0.0012 | 0.9938 |
| **SpeedCP($\Phi^*$)** | 0.2662 | 0.0819 | 0.0025 | 0.9533 |
| SplitCP | 0.3482 | 0.0091 | 0.3271 | 0.3660 |
| RLCP | 0.3482 | 0.0091 | 0.3271 | 0.3660 |
| PCP | 0.2818 | 0.2904 | 0.0000 | 0.9984 |
| $\hat{\boldsymbol{y}} = $ tumor | | | | |
| **SpeedCP(1)** | 0.2758 | 0.0866 | 0.0963 | 0.9985 |
| **SpeedCP($\Phi^*$)** | 0.2925 | 0.0805 | 0.0952 | 0.9714 |
| SplitCP | 0.3482 | 0.0091 | 0.3271 | 0.3660 |
| RLCP | 0.3482 | 0.0091 | 0.3271 | 0.3660 |
| PCP | 0.2012 | 0.2855 | 0.0000 | 0.9984 |

Table 8: Mean coverage and prediction set size across predicted labels in the MRI dataset under the PCA-based model.

| Method | Target coverage ($1 - \alpha = 0.9$) | | | Prediction set size | | | Time (seconds) |
|---|---|---|---|---|---|---|---|
| | Marginal | Healthy | Tumor | Marginal | Healthy | Tumor | |
| **SpeedCP(1)** | 0.910 ±0.01 | **0.901** ±**0.02** | 0.915 ±0.01 | 0.239 ±0.07 | **0.230** ±**0.07** | 0.244 ±0.08 | 286.1 ±14.2 |
| **SpeedCP($\Phi^*$)** | 0.905 ±0.02 | **0.898** ±**0.03** | **0.900** ±**0.02** | 0.247 ±0.08 | 0.241 ±0.08 | 0.251 ±0.08 | 294.5 ±20.9 |
| SplitCP | **0.901** ±**0.01** | 0.893 ±0.02 | 0.906 ±0.01 | 0.350 ±0.00 | 0.350 ±0.00 | 0.350 ±0.00 | < 0.01 |
| PCP | 0.906 ±0.02 | 0.925 ±0.03 | 0.895 ±0.02 | **0.230** ±**0.27** | 0.279 ±0.26 | **0.200** ±**0.26** | 130.1 ± 28.9 |
| RLCP | 0.916 ±0.01 | 0.926 ±0.02 | 0.911 ±0.02 | 0.359 ±0.38 | 0.388 ±0.37 | 0.342 ±0.38 | 2.095 ±0.13 |

Table 9: Summary statistics of conformal cutoffs (marginal and by predicted label) using PCA-based model. **SpeedCP($\Phi^*$)** calibrates scores with a linear term that includes predicted labels, whereas **SpeedCP(1)** uses an intercept-only term.

| Method | Mean | Std | Min | Max |
|---|---|---|---|---|
| **Marginal** | | | | |
| **SpeedCP(1)** | 0.2391 | 0.0738 | 0.0654 | 0.8641 |
| **SpeedCP($\Phi^*$)** | 0.2470 | 0.0805 | 0.0442 | 1.2279 |
| SplitCP | 0.3505 | 0.0087 | 0.3315 | 0.3729 |
| RLCP | 0.3594 | 0.3797 | 0.0000 | 0.9984 |
| PCP | 0.2301 | 0.2672 | 0.0000 | 0.9984 |
| $\hat{\boldsymbol{y}} = $ healthy | | | | |
| **SpeedCP(1)** | 0.2300 | 0.0697 | 0.0654 | 0.7414 |
| **SpeedCP($\Phi^*$)** | 0.2409 | 0.0785 | 0.0442 | 1.2279 |
| SplitCP | 0.3506 | 0.0088 | 0.3315 | 0.3729 |
| RLCP | 0.3883 | 0.3711 | 0.0000 | 0.9984 |
| PCP | 0.2788 | 0.2654 | 0.0000 | 0.9984 |
| $\hat{\boldsymbol{y}} = $ tumor | | | | |
| **SpeedCP(1)** | 0.2445 | 0.0756 | 0.1486 | 0.8641 |
| **SpeedCP($\Phi^*$)** | 0.2506 | 0.0815 | 0.0615 | 1.2225 |
| SplitCP | 0.3505 | 0.0087 | 0.3315 | 0.3729 |
| RLCP | 0.3420 | 0.3838 | 0.0000 | 0.9984 |
| PCP | 0.2009 | 0.2641 | 0.0000 | 0.9984 |

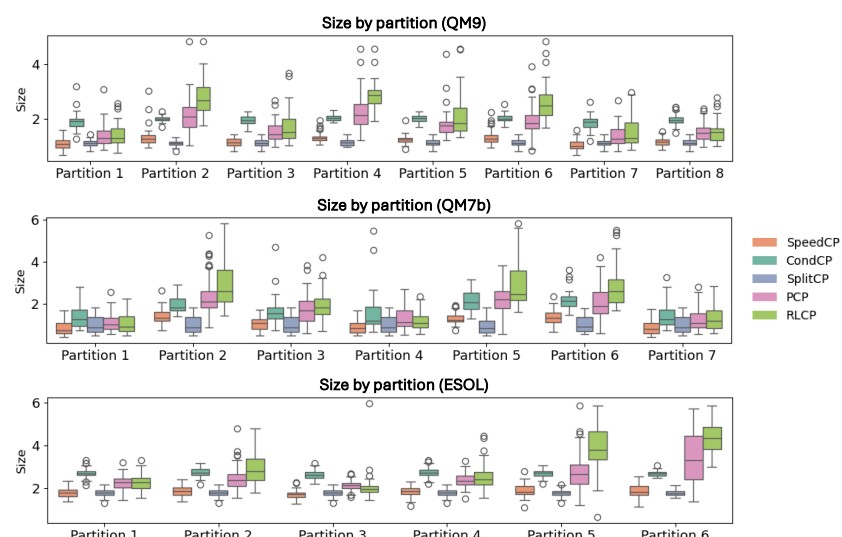

Figure 13: Prediction set size on fixed partitions of the PC space for each molecule dataset. We use PCA on the last layer embeddings of GNN with $K = 3$ dimensions.

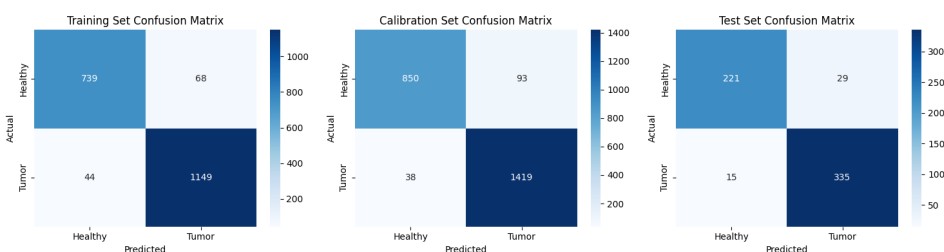

Figure 14: Evaluation of the CNN classifier on the Brain Tumor MRI dataset.

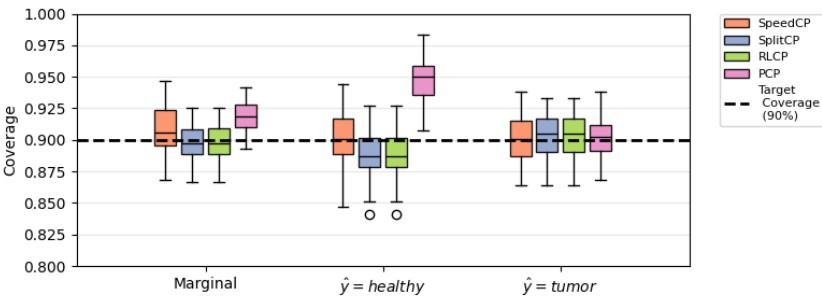

Figure 15: Predicted-label conditional coverage on the Brain Tumor MRI test set under the PCA-based model. Calibration is performed using the linear feature map $\Phi^*(X) = \left(\mathbf{1}\{\hat{\mu}(X) = \text{healthy}\}, \mathbf{1}\{\hat{\mu}(X) = \text{tumor}\}\right)^\top$ under the 256-dim features layer from NN.

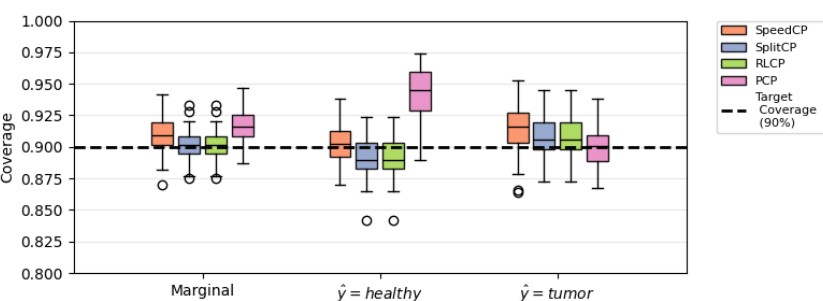

Figure 16: Predicted-label conditional coverage on the Brain Tumor MRI test set by calibrating with the intercept only $\Phi^*(X) = 1$ under the 256-dim features layer from NN.

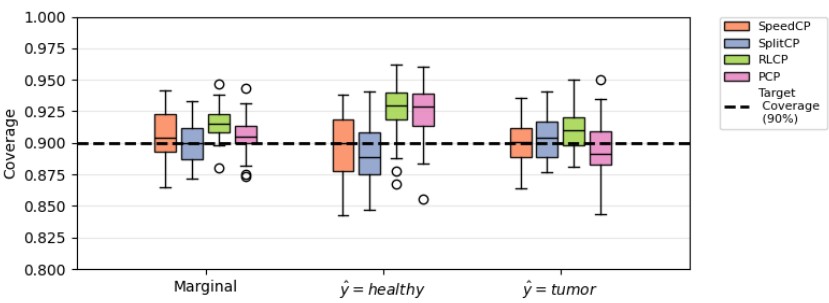

Figure 17: Predicted-label conditional coverage on the Brain Tumor MRI test set under the PCA-based model. Calibration is performed using the linear feature map $\Phi^*(X) = \left(1, \mathbf{1}\{\hat{\mu}(X) = \texttt{healthy}\}, \mathbf{1}\{\hat{\mu}(X) = \texttt{tumor}\}\right)^\top$.

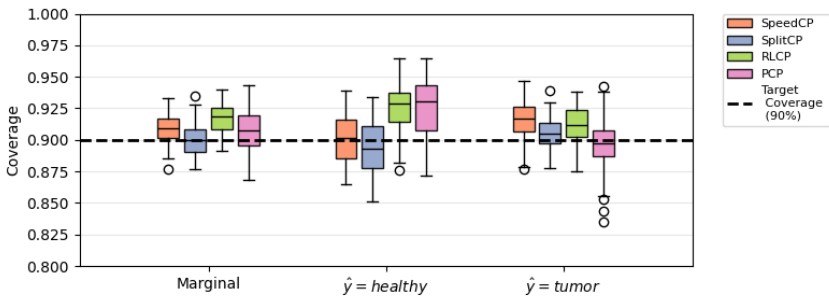

Figure 18: Predicted-label conditional coverage on the Brain Tumor MRI test set by calibrating with the intercept only $\Phi^*(X) = 1$.

