# OpenReview forum: "SpeedCP: Fast Kernel-based Conditional Conformal Prediction"
_ICLR.cc/2026/Conference — Submitted to ICLR 2026_

### Official Review · Reviewer_qs9h · 2025-10-28

**Soundness:** 3
**Presentation:** 3
**Contribution:** 3
**Rating:** 8
**Confidence:** 4

**Summary:**

This paper improves the practical implementation of the RKHS-based conformal prediction framework of Gibbs, Cherian, and Candès (2023). That earlier framework guarantees approximate conditional coverage over a class of covariate shifts \mathcal{F}, which was shown in Gibbs et al. to be equivalent to a relaxed form of conditional coverage. In their method, Gibbs et al. construct a quantile regression model to estimate the (1-alpha) quantile of nonconformity score S(x*, y*) as a function of new test data x* within an RKHS defined by \mathcal{F}. While searching for the model in the space, they solve a regularized optimization problem with a regularization parameter \lambda. To ensure exchangeability of the calibration set and the new data point to achieve theoretical coverage, this optimization needs to be repeated for different possible candidate values of S(x*, y*). Further, it's ideal to tune \lambda, so this optimization would be repeated for different values of \lambda. These repetitions make the computation expensive.
The key contribution of the proposed SpeedCP is to accelerate the aforementioned computation by finding a solution path over \lambda and S, specifically, by reducing computation to the tracking and computation concerning an (often much smaller than size n) elbow set. In addition, SppedCP allows low-rank latent embeddings of X to handle high-dimensional covariates that further reduces computational cost when p is large. Theoretical results on conditional validity in terms of the low-rank embedding is provided.

**Strengths:**

Well written (literature review, motivation, logical flow, clarity, rigorosity)

**Weaknesses:**

The calibration set was further split into K folds, to help find best (\lambda, \gamma). This procedure was only very lightly mentioned in the end of sec.2.1.1. Was this expensive? Was the cost reported/included in the empricial examples? (E.g. when SpeedCP was claimed to be 50 times faster in figure 1, and in table 4)

I'd appreciate a clearer presentation (say a table) that shows the samples size (training and calibration), the dimension of x (say, p), the dimension of the embedding, (and perhaps the computing cost of SpeedCP) for all examples. This will provide a good practical guidance for future users. How large a p is too large, and how much should we reduce the dimension. The pro and con of reducing p.  Was the embedding of dim 256 or further reduced for SpeedCP in the Brain MRI example?

I don't get the arxiv example, e.g., it can be made clear in the beginning what is the purpose of the application including what is the output, number of citations? What is this supposed to mean:"For the predictor, we choose linear regression of citation counts on raw word frequencies, which fails to extract any meaningful associations between words and citation counts."

minor: covariate shift was not defined explicitly. Consider introducing it in a way that readers do not have to seek its definition in Gibbs et. al. 2023.

**Questions:**

See the weakness section for some Questions.

**Details Of Ethics Concerns:**

The statements in page 10 are well written and look truthful to me.

---

> ### Author Response · Authors · 2025-11-24
> **We clarified cross-validation cost, added tables summarizing dataset and embedding dimensions, revised the arXiv section and covariate-shift descriptions for clarity, and ensured the manuscript now provides practical guidance on tuning, data specification, and methodological intent.**
>
> **W1: Computation cost for cross-validation.**
> > We appreciate this comment and added a detailed description of our cross-validation procedure in Appendix B.5 and Appendix D.1. All reported runtimes already include CV time, and SpeedCP remains 30–50× faster than CondCP even though we tune both $\lambda$ and $\gamma$ while CondCP only tunes $\lambda$; in our simulations, CV accounts for roughly 85\% of the total runtime, but the overall procedure is still very fast compared to CondCP.
>
> **W2: Table for summarizing the data specifications and dimensions for MRI dataset.**
> > Thank you for this helpful suggestion. We agree that summarizing the data specifications and embedding dimensions in a single table would greatly improve the clarity and practical usefulness of the paper. In the revised manuscript, we have added a table for each dataset in Table 1. Regarding the reviewer’s question about the effect of reducing  $p$: in practice, SpeedCP does not require the raw covariate dimension to be small, because the conformal calibration is performed on the scalar conformity scores. What matters for conditional validity is the quality of the embedding $\hat\pi(x)$. Reducing $p$ through a representation map (e.g., PCA, neural embeddings, topic or mixture models) typically improves group homogeneity (based on users' preferences) and thus leads to more efficient prediction sets. The trade-off is that overly aggressive dimension reduction can obscure meaningful structure. For the Brain MRI experiment in particular: the raw imaging data are extremely high dimensional, so we first extract 256-dimensional features using a pretrained encoder, and use those 256-dimensional embeddings directly. On the other hand, we could also use PCA to further reduce the dimension to $K=8$, which we did in Appendix D.2.3. The resulting rank of PCA does not affect the coverage guarantee significantly.
>
>
> **W3: Specifications for arxiv dataset.**
> > We agree with the reviewer and have revised the manuscript to clearly state the purpose of the arXiv experiment, including what the method is predicting and what the outputs represent (predicted citation counts). We have also clarified that the experiment intentionally uses a weak predictor, such as linear regression on raw word frequencies, to illustrate how different calibration methods behave when the underlying model has very limited predictive power. The revised text in Appendix D.2.1 now explains this motivation explicitly and avoids the ambiguous phrasing in the original version.
>
> **W4: Clarification for covariate shift.**
> > We thank the reviewer for pointing this out. We agree that the notion of covariate shift should be defined directly in the paper rather than use it in the proof only. In the revision, we will add a clear, self-contained definition when the concept first appears. We will also briefly explain how this framework motivates the use of importance-weight tilting functions
> $f$ in Section 2.2 and how it connects to the conditional guarantee we analyze. This clarification will allow readers to follow the theory without needing to consult external references such as Gibbs et al. (2023).

---

> > ### Comment · Reviewer_qs9h · 2025-11-25
> > **The answers to my questions are satisfactory**
> >
> > The answers to my questions are satisfactory.

---

### Official Review · Reviewer_HaCa · 2025-11-01

**Soundness:** 3
**Presentation:** 3
**Contribution:** 2
**Rating:** 6
**Confidence:** 3

**Summary:**

The paper introduces SpeedCP, a novel framework for conditional conformal prediction that addresses the computational limitations of prior work while providing stronger coverage guarantees than marginal methods. The authors build upon the RKHS-based framework of Gibbs et al. (2023), which provides approximate conditional coverage guarantees but suffers from high computational costs. SpeedCP's key idea is a solution-path algorithm that efficiently computes the full regularization path of the RKHS conformal optimization problem at essentially the same cost as a single kernel quantile fit. The method simultaneously tunes hyperparameters for smoothness control and data-adaptive calibration. To extend applicability to high-dimensional settings, the authors integrate their approach with low-rank latent embeddings that capture conditional validity in a data-driven latent space.

**Strengths:**

The solution-path approach for RKHS-based conformal prediction appears to be interesting. The $\lambda$-path and S-path formulations are well-designed and theoretically sound. The paper provides solid theoretical foundations with finite-sample guarantees for approximate conditional coverage in both standard and high-dimensional settings. Theorems 1 and 2 characterize the coverage properties under different assumptions.  In addition. the integration with low-rank latent embeddings addresses the difficulty of applying conditional conformal prediction to high-dimensional settings.

**Weaknesses:**

The coverage guarantees in Theorem 1 involve a coverage gap that depends on the embedding quality, which is usually unknown is practice.
This raises the concern that whether the proposed method can guarantee coverage in pratice.

There is only limited comparison to existing ethods:  the paper compares against CondCP, SplitCP, PCP, and RLCP, it would benefit from comparisons to more recent methods like those in Xu et al. (2024) or Shahrokhi et al. (2025) mentioned in the references and perhaps recent methods based on normalizing flows.

The method involves selecting a two-dimensional $\gamma, \lambda)$ grid. It is not clear how sensitive/robust the method is with respect to
the selection.

**Questions:**

The paper describes cross-validating over a grid of $(\gamma, \lambda)$ values, but doesn't discuss how sensitive the results are to this choice. How the conditional coverage and prediction interval size vary with different $(\gamma,\lambda)$ values, particularly in the high-dimensional settings?

Is Theorem 1 for a fixed pair of $(\gamma, \lambda)$? Does it  depend on the randomness in the selected $(\gamma, \lambda)$ values?

Same question as above for Theorem 2.

---

> ### Author Response · Authors · 2025-11-24
> **In summary, we clarify that Theorem 1 is an interpretive, estimable decomposition of conditional coverage difficulty (not a strict asymptotic guarantee), that SpeedCP targets a different scalar, representation-conditional setting than Xu et al. (2024) and Shahrokhi et al. (2025), that coverage is empirically robust across (lambda, gamma).**
>
> **W1: Concern about coverage gap in Theorem 1:**
> > We appreciate the reviewer’s question. The key point is that Theorem 1 is not to show the asymptotic or exact coverage guarantee of our method. The coverage gap in Theorem 1 quantifies the difficulty of achieving conditional coverage in a given direction of the tilt function $f$. Specifically, if we have small number of effective data near the given embeddings $w'$, so the residual coefficient from the RKHS projection is large in magnitude which indicates the conditional coverage around $w'$ is hard to guarantee. Even though this gap is not directly observable, it can be directly estimated by $\frac{\sum_i\hat \upsilon_{n, i} \psi^\star_W(w', \hat\pi(X_i))}{n^{-1}\sum_i\psi^\star_W(w', \hat\pi(X_i))}$ where $W'=w'$ is fixed and $\hat \upsilon_{n,i}$'s are the empirical coefficients from  $\hat g_{n, \psi^\star}$ (from Proposition 4). The exact or asymptotic condition coverage is hard to achieve without strong assumptions (Barber et al. 2021; Guan 2023). Theorem 1 provides an explicitly estimable decomposition of the deviation, with each component arising from measurable quantities of the method, rather than relying on any assumed prior structural conditions.
>
>
> **W2: Comparison with Xu et al 2024 and Shahroki et al 2025:**
> > We appreciate the suggestion to compare with Xu et al. (2024), Shahrokhi et al. (2025), and related flow-based methods. These approaches, however, target a different problem than SpeedCP. Xu et al. (2024) focuses on constructing geometric prediction sets using specialized covariance structures. Shahrokhi et al. (2025) study deep generative models (like LLMs) and build prediction sets for sampling. Their primary goal is to improve the prediction set under strong modeling or distribution assumptions. By contrast, SpeedCP focuses on conditional-coverage across latent covariate structure, where the predictor assumes a fixed scalar outcome, rather than a generative model. For these reasons, a direct empirical comparison with Xu et al. (2024) and Shahrokhi et al. (2025) would not be very informative. We agree that extending SpeedCP to their settings with deep generative models is an interesting future direction.
>
>
> **W3: Robustness of $(\lambda, \gamma)$:**
> > We thank the reviewer for raising this point. In the revised manuscript, we included additional robustness experiments in Appendix D.1.5 that vary the $(\gamma, \lambda)$ over a fixed grid. Right panel of Figure 7 in the revision shows that the coverages remain uniformly close to the nominal level (0.9) across all evaluated values of $(\gamma, \lambda)$. Therefore, the coverage guarantee is robust to the choice of tuning parameters. However, the choice of $(\gamma, \lambda)$ does vary the tightness of the resulting prediction sets. Since some choices of $(\gamma, \lambda)$ produce smoother or more adaptive quantile estimates, they yield tighter sets (also observed in right panel of Figure 7). Our algorithm uses the $\lambda$-path together with the cross-validation to select a pair that balances smoothness and flexibility, resulting in tighter intervals while preserving the formal coverage guarantee.
>
>
> **Q1: The relationship between $(\lambda, \gamma)$ and Theorems:**
> > We thank the reviewer for the question. Both Theorem 1 and Theorem 2 are stated for a fixed pair of $(\lambda, \gamma)$ (including the selected $(\hat\lambda, \hat\gamma)$ from the algorithm). The theoretical guarantees do not rely on any randomness from hyperparameter selection and are valid for every deterministic choice of $(\lambda, \gamma)$ in the grid. In our implementation, cross-validation is used only as a practical method for improving efficiency (prediction-set size), not for establishing validity of coverage. Therefore, using the CV with calibration set is not perfect aligned with the theorem. To avoid confusion, in the revision we added experiments in Appendix D.1.4 and D.1.5 to show that the fixed values of $(\lambda, \gamma)$ and CV values of $(\lambda, \gamma)$ does not effect the empirical coverage too much.
>
>
> Leying Guan. Localized conformal prediction: A generalized inference framework for conformal prediction. Biometrika, 110(1):33–50, 2023.
> Rina Foygel Barber, Emmanuel J Candes, Aaditya Ramdas, and Ryan J Tibshirani. The limits of distribution-free conditional predictive inference. Information and Inference: A Journal of the IMA, 10(2):455–482, 2021.

---

### Official Review · Reviewer_vevS · 2025-11-01

**Soundness:** 3
**Presentation:** 2
**Contribution:** 2
**Rating:** 2
**Confidence:** 4

**Summary:**

Submission SpeedCP tackles conditional (not just marginal) coverage in conformal prediction by enforcing the guarantee over a rich RKHS class $\mathcal{F}^\*=\\{f\_{\psi^\*}+\Phi^\* \eta\\}$. To make similarity meaningful in high dimensions, the kernel $\psi^\*(x\_1, x\_2)$ is defined on a low-rank embedding $\hat{\pi}(X) \in \mathbb{R}^K$ (e.g., LDA/PCA), with distance computed in centered log-ratio space. Prediction sets are obtained by solving a regularized RKHS quantile regression on calibration points plus the test point with an imputed score $S$ (4), yielding the representer form $\hat{g}\_S(x)=\Phi^*(x)^{\top} \hat{\eta}\_S+\frac{1}{\lambda} \sum\_i \hat{\nu}\_{S, i} \psi^\*(x, X\_i)$ and the elbow/left/right partition.


Computationally, the authors proposed a pair of solution-path algorithms, i.e. a ${\lambda}$-path for smoothness selection on calibration data only, and an S-path that traces the test score until it reaches the conformal threshold.
Between "events", coefficients evolve affinely in $\lambda$ (resp. in S). Algorithm 1 couples $\lambda$-path cross-validation over ($\gamma, \lambda$) with per-test S-paths, offering substantial speedups.

On theory, a randomized tilted-expectation identity (Lemma 4) extends Gibbs to this RKHS/representer scaling, expressing coverage deviation as $-\hat{\lambda} \mathbb{E}\langle\hat{g}\_{\psi^\*}, f\_{\psi^\*}\rangle$. A localized guarantee (thm 1) conditions on a random neighborhood $W^{\prime}$ drawn from a density kernel over the latent space, giving an explicit coverage ratio (10); a group-conditional guarantee (thm 2) holds for features that encode latent components under a topic-alignment condition.

**Strengths:**

- The motivation of conditional conformal prediction in RKHS appears reasonable

- On my verification, the technical development is overall correct and sound. The piecewise linearity of the derived path is a nice property for both implementers and theorists

- Empirically, SpeedCP attains more uniform coverage with smaller sets and reports observed speedups over CondCP on a series of provided benchmarks

**Weaknesses:**

Major weaknesses:

- The paper oversells the algorithmic novelty relative to what’s already in the ML toolbox. The $\lambda$-path derivation, KKT structure, and  active set partitions are garden-variety kernel quantile-regression path-following; this is the same playbook [1] (and additionally [2]) used, and the paper just carry along an extra linear term $\Phi^*$ and then code the usual event-driven updates. The $\lambda$-path piece is incremental, basically a straightforward adaptation. The S -path is a natural, low-effort extension (change the homotopy parameter from $\lambda$ to $S$ and keep the same active-set bookkeeping). To me the only novelty is the use of those paths inside the RKHS-CP pipeline. It also should be noted that even just the kernel solution path itself is by no means a novel concept in ML/optimization, see e.g. [3,4]


- Hyperparameter selection leaks calibration and targets the wrong objective. The paper tunes $(\gamma, \lambda)$ by ${k}$-fold cross-validation on the calibration set to minimize pinball loss (Algorithm 1, sec. 2.1.1). That 1) uses the same calibration points later used for conformal calibration, and 2) optimizes a regression loss rather than the quantity that matters-coverage (or the coverage gap in (10)). There is no formal theorem showing that this two-stage selection preserves the stated finite-sample coverage properties. This philosophy is exactly the kind of selection-induced calibration drift CP tries to avoid.




Minor concerns:

- The central theoretical claim (Theorem 1) is an identity that expresses the coverage error as a ratio involving unknown terms, specifically an RKHS‑weighted residual term over a kernel density in the latent space. There's no non‑asymptotic bound that forces this gap to be small, nor any verifiable a‑priori control (i.e. users can’t evaluate without extra modeling assumptions)

- The entire work assumes the columns of $\Phi^*$ are linearly independent. The assumption itself is standard and reasonable, however, the usage of it is highly problematic. First, several experiments specify $\Phi^\*$ with intercept + one-hot indicators (synthetic, MRI), which is rank-deficient by construction. Importantly, you actually need a (stronger) elbow-level rank condition ( $|E| \geq d$ and $\operatorname{rank}(\Phi\_E^{\*})=d$) along the entire path; this is neither stated nor enforced. I mean, the work initialize the $\lambda$-path at the largest value for which "at least two points are in the elbow". Two points is not enough unless you have $d \leq 2$. For general $d$, one need $|E| \geq d$ and $\operatorname{rank}(\Phi\_E^{{ }^\*})=d$ so that $(\Phi\_E^{^*T } \Phi\_E^{^\*})^{-1}$ exists at every segment.

- The simulations fix exactly one base model per dataset. The authors never test whether SpeedCP's gains persist if you swap $\hat{\mu}$ for a stronger or different architecture on the same task (e.g., transformer text models instead of naive bag-of-words linear regressor; alternative GNNs on molecules). That omission weakens the generality claim because cp methods are supposed to be model-agnostic

Bib:

[1] Li, Youjuan, Yufeng Liu, and Ji Zhu. "Quantile regression in reproducing kernel Hilbert spaces." Journal of the American Statistical Association 102.477 (2007): 255-268.

[2] Takeuchi, Ichiro, Kaname Nomura, and Takafumi Kanamori. "Nonparametric conditional density estimation using piecewise-linear solution path of kernel quantile regression." Neural Computation 21.2 (2009): 533-559.

[3] Gu, Bin, Ziran Xiong, Xiang Li, Zhou Zhai, and Guansheng Zheng. "Kernel path for ν-support vector classification." IEEE Transactions on Neural Networks and Learning Systems 34, no. 1 (2021): 490-501.

[4] Zhai, Zhou, Heng Huang, and Bin Gu. "Kernel path for semisupervised support vector machine." IEEE Transactions on Neural Networks and Learning Systems 35, no. 2 (2022): 1512-1522.

(COI: i'm not one of the authors of these citations)

**Questions:**

- Theorem 2 demands perfect topic alignment $\hat{T}(X)=T(X)$ *almost surely*, which is equivalent to requiring a bayes-optimal argmax classifier on the latent mixture with exact zero error. Handwaving that this holds "under a margin condition" doesn't make it realistic. Without that, the group-conditional guarantee collapses. May the authors have more comment over this?


- the work rely on randomized cutoffs S_rand via $U$~$Unif(-\alpha, 1-\alpha$) (see eq. (9)) to avoid *inflated coverage*. That makes predictions non-deterministic across runs; could the authors explore the variance induced by this step instead of reporting means over 50 runs?

- in experiment, the work excludes CondCP *entirely* on the arXiv and MRI experiments for computational difficulty, then claim superiority; that’s not a real fair comparison




While reading, I noticed a few minor issues that don't affect the evaluation, e.g.

- l79, a new path algorithms: remove s

- l83, the authors used "event" point which is not defined until page 4

- l134, preliminary notations: usually an uncountable noun

- eq4: an additional $^2$; consider to rearrange the footnote

- l218, We proceed to constructing: construct

- Theorem 1: you assume a density kernel $\psi\_W^\*$ with $\psi\_W^\*(\hat{\pi}(x\_1), \hat{\pi}(x\_2))=\psi^\*(x_1, x_2)$. Since you define $\psi^\*(x\_1, x\_2)= \exp \\{-\gamma d_\pi(\hat{\pi}(x\_1), \hat{\pi}(x\_2))\\}$, to interpret $\psi\_W^\*$ as a probability density in its second argument; please state explicitly that $\psi^\*$ is either normalized already or that the equality holds up to a constant that cancels in the ratio on the RHS of eq.10

- l823, wider prediction intervel: interval

- From the Lagrangian (21) and stationarity condition (22), you correctly get $v_{S, i}=\sigma_i-\tau_i$, $\sigma_i=1-\alpha-\kappa_i$, $\tau_i=\alpha-\rho_i$. But in (24), for positive residual $S_i-g_S\left(X_i\right)>0$ you wrote $\sigma_i=\alpha$, $\tau_i=0$. It should be $\sigma_i=1-\alpha, \tau_i=0$ (values are swapped with negative residual)

- l1299, Then We have: we

- p.28, proof of Lemma 4: when you borrowed results from Gibbs et al. (2023), the $-\mathbb{E}[f(X\_{n+1}) \hat{v}\_{n+1}]= -2 \mathbb{E}[\lambda\langle\hat{g}, f\rangle]$ is only correct for their scaling. (no factor 2 as your own objective uses $\frac{\lambda}{2}\\|g\|\^2$)

For completeness, the authors should consider incorporating portions of appendix A into the main body, especially definitions and key related work.

---

> ### Author Response · Authors · 2025-11-25
> **Our response clarifies that our main contribution is to exploit the affine kernel path structure together with low-rank representations to obtain fast, representation-conditional coverage, and to reinterpret Theorem 1 as an estimable diagnostic of localized coverage difficulty and Theorem 2 as an oracle efficiency limit rather than a practical assumption.**
>
> **W1: novelty beyond the algorithm.**
> > We thank the reviewer for their extremely insightful and detailed feedback. We agree that the novelty of this paper does not lie in the conditional conformal prediction setting, nor in proposing a new kernel solution path. Neither of these great ideas are ours. Instead, our contribution is to integrate the ideas of kernel solution path from Li et.al. 2007 [1] and Takeuchi 2009 [2] to the conditional conformal prediction framework, thereby leading to our two main contributions:
>
> > Coverage guarantee when using low-rank representations: we use the framework of Gibbs et.al. 2023 to develop theoretical guarantee of the kernel-based conditional conformal prediction when we use learned low-rank representation $\hat{\pi}(X)$ such as latent group indicators or neural network embeddings. This is different from a classical analysis performed on the raw covariates $X$. Our finite-sample coverage bounds thus accounts for the dependence introduced by unobserved priors W and errors given by $\hat{\pi}(X)$.
>
> > Affine path structure in $S$ and $\lambda$: under the kernel-based conditional conformal prediction framework, we prove in Propositions 1,2 that the dual kernel variables $\hat{\nu}_{S,i}$'s and the linear variables $\hat{\eta}_S$ are affine and monotone in both $S$ and $\lambda$. This enables us to construct prediction sets by adapting the solution path for $S$, as opposed to the binary search procedure used by Gibbs et al.(2023), who establish the monotonicity (their Theorem 4) but not the stronger affine relationship. This yields substantial gain in the computation time (60-fold improvement in Table 2). Exploiting this property is the main reason such path-based methods are of practical interest. Although the underlying ideas are not new, we would argue that bridging them and showing that the framework can be adapted effectively and deliver strong empirical performance in the regimes of interest constitutes a meaningful and practically relevant advance.
>
> **Minor concern 1: how to evaluate the coverage gap in Theorem 1?**
> > We appreciate the reviewer’s comments. The intent of Theorem 1 is not to claim that the coverage gap is a priori small, nor that it provides a directly computable upper bound on the coverage error. Instead, the format of Theorem 1 provides a structural decomposition of the conditional coverage gap, which quantifies why and in what direction conditional coverage is difficult given a chosen tilt $f$. Achieving exact conditional coverage under localization is statistically impossible in general without strong structural assumptions. Instead, Theorem 1 gives practitioners guidance on how to choose a tilt $f$ that reduces the risk of large gaps.
>
> > Under the setting in Theorem 1, the tilt function $f^{w'}(x)=\psi^\star_W(\hat\pi(x), w')$ emphasizes coverage in a neighborhood around the fixed point $w'$ in the latent space (as detailed in the proof in Appendix C.2). As shown in Appendix C.4.2, this coverage gap $\frac{E[\sum_{i\in[n+1]}\hat \upsilon_{S^{rand}, i} \psi^\star_W(W', \hat\pi(X_i))]}{\mathbb{E}[\psi^\star_W( W', \hat\pi(X))]}$ admits a data-driven approximation $\frac{\sum_{i\in[n]}\hat \upsilon_{n, i} \psi^\star_W(w', \hat\pi(X_i))]}{\frac{1}{n}\sum_{i\in[n]}\psi^\star_W(w', \hat\pi(X_i))]}$ where $W'=w'$ is fixed and $\hat \upsilon_{n,i}$'s are the empirical coefficients from  $\hat g_{n, \psi^\star}$ (from Proposition 4). While there is no exact closed-form bound for the gap, the format of the gap provides a practically estimable diagnostic of the difficulty to achieve localized conditional coverage around a given point $w'$.
>
> **Q1: is it required $\hat T=T$ in practice?**
> > We appreciate this insightful comment and clarify the role of $\hat T= T$ in Theorem 2. This assumption is not required in practice and our method does not rely on Bayes-optimal classification of the latent mixture. Theorem 2 is intended only as an oracle, population-level result that describes what happens if the learned representation exactly recovers the true latent structure. For better presentation, we revised the Theorem 2 in the main paper.
>
> > In practice, neither $W$ nor $\pi(X)$ is observed, so we instead condition on data-driven projection, $\hat\pi(X)$, and its induced partition $\hat T(X)$. Our finite-sample guarantee holds for any learned partition as long as the tilt is defined using $f(X)=\mathbf{1}\{ \hat T(X)=k\}$. The accuracy of $\hat T$ relative to $T$ does not affect validity, but might affect the prediction set size. A representation closer to the true latent structure yields tighter, more meaningful prediction sets, while a coarser representation still maintains coverage but may yield less efficient sets. This is precisely why Theorem 2 analyzes true-group coverage using the natural population embedding $\pi(X)=\mathbb{E}[W\mid X], as it represents the ideal limit that our learned embedding seeks to approximate for maximal efficiency.

---

> ### Author Response · Authors · 2025-11-25
>
> **W2**:
> > We thank the reviewer for raising this important point about hyperparameter selection. **Our finite-sample coverage guarantee applies to every fixed pair $(\lambda,\gamma)$ in the grid, but prediction set size may differ across the grid.** (We have added additional experiments in Appendix D.1.4 to illustrate this point.) The tuning step is therefore not part of the validity argument, but to improve efficiency and yield tighter prediction sets. In the original implementation, $(\lambda,\gamma)$ was selected using cross-validation on the calibration set, which introduces mild data dependence. To address this, we revised the method to include a separate set for hyperparameter tuning. At the same time, we clarify in Appendix D.1.3 that using the calibration set for tuning can be acceptable in small-sample regimes, which is more efficient.  Recent work on adaptive coverage policies shows that it is empirically valid to use the calibration sample itself to select regularization parameters via cross-validation (Theorem 2.6 in Gauthier et al. 2025). Also, the quantile model is used to construct a good thresholding function, not to directly control coverage, and pinball loss is just a practical way to choose a smoother estimator within this class. The “selection-induced calibration drift” arises only when one searches over models by optimizing the coverage itself, which forces degenerate solutions; our method avoids this entirely. We note that the coverage gap itself is determined by the complexity of the RKHS and the difficulty of obtaining conditional coverage with tilt $f$, not by the regression loss.
>
> **Minor concern 2**:
> > We clarify the design matrix $\Phi^\ast$ and address the concern about rank deficiency. The reviewer correctly notes that an ``intercept + full one-hot encoding’’ would be rank-deficient; however, this is not the design used in our experiments. In all synthetic and MRI settings, there are $K$ groups, and the design matrix is an intercept plus $K-1$ one-hot variables, which has rank $K$. Our original text was intended to mean that the design supports both marginal and group-conditional coverage, but we agree it was misleading and have corrected it. We also agree that the closed-form path update requires $|E|\ge d$ and  $\mathrm{rank}(\Phi^\ast_E)=d$. At initialization, however, we set $\lambda^0=\infty$, place a single point $i_0$ into the elbow,
> and zero out all coefficient entries of $\Phi^\ast$ except the intercept for identifiability.  We then choose $\lambda^1$ so that the next point enters while satisfying all KKT conditions.  In this regime, the solutions $\hat{\upsilon}_i(\lambda^0)$ and $\hat{\upsilon}_i(\lambda^1)$ are deterministic and do not require inverting $(\Phi^\ast_E)^\top \Phi^\ast_E$, so the rank condition
> is not needed at this stage. For all subsequent $\lambda$-updates, where the reviewer’s concern does apply, we use the stabilized matrix $(\Phi^\ast_E)^\top \Phi^\ast_E + rI$ with $r = 10^{-6}$ until $|E|\ge d$.  This guarantees well-posed updates even when $\Phi^\ast_E$ is temporarily rank-deficient.
>
> **Minor concern 3**:
> > Conformal prediction is indeed model-agnostic, and it is natural to ask whether the results hold for SpeedCP with different $\hat\mu$. In the revised manuscript (we added Table 2), we include additional synthetic experiments in which we set the predictor as a neural network, which has smaller residuals. These experiments allow us to compare CP methods under systematically improved predictive power. As expected, when the underlying predictor becomes stronger, all methods see a modest improvement in tightness of the prediction set size. However, the key finding is that SpeedCP consistently maintains more uniform conditional coverage (we added Figure 4), whereas other baselines exhibit larger fluctuations in sparse regions even when equipped with a stronger predictor.
>
> **Q2**:
> > We agree that, in principle, randomizing the cutoff $S^{\mathrm{rand}}$ could introduce variability across runs. In practice, however, this randomization contributes negligible variability because the $S$-path for a test point begins at a small value, and throughout the path, $S$ values are typically strictly smaller than $\hat{g}$, which forces $\hat{\upsilon}_{S,(n+1)} = -\alpha$. As a result, the effect of the randomized cutoff is minimal. We have added an experiment demonstrating this behavior in Appendix D.1.5 of the revised version.
>
> **Q3**:
> > We apologize for the confusion. We attempted to run all five methods, including CondCP, on both the arXiv and MRI datasets, but CondCP failed to finish within a 6-hour time budget, whereas all other methods completed in under 10 minutes. We have updated Appendix D.2.1 to make these runtime limitations explicit and to clarify that CondCP was omitted from the tables because it did not finish, not because it was not attempted.
> - Gauthier, E., et al. (2025). Backward Conformal Prediction.

---

### Official Review · Reviewer_yCxV · 2025-11-04

**Soundness:** 3
**Presentation:** 2
**Contribution:** 2
**Rating:** 4
**Confidence:** 4

**Summary:**

This paper proposes a general nonparametric method for constructing conformal prediction sets with specific conditional guarantees. The authors complement the algorithm with a theoretical analysis. The experimental study shows some benefits of the proposed method.

**Strengths:**

- Extending conditional properties of conformal prediction approaches is of high interest
- The proposed algorithm looks sound as well as Theorem 1 (I have some questions with respect to Theorem 2; see below)
- Experimental study is generally well executed

**Weaknesses:**

- The worst-case complexity of the algorithm is high. I would recommend doing an experimental study, varying the size of the dataset to see what the pattern is.

- Theorem 2 assumes that $\hat{T} = T$. It is looks like extremely strong assumption to recover this mapping precisely.

- In many experiments, FastCP is closely matched by simple SplitCP (especially, Figures 2 and 3). It is unclear what the benefit of FastCP is, then.

**Questions:**

- What is the map $\Phi^*$ introduced on page 3? It seems that you do not explain what it is.

- What is the benefit of FastCP over SplitCP? Why the differences are small between them?

---

> ### Author Response · Authors · 2025-11-25
>
> **W1: Varying sample size $n$**:
> > We thank the reviewer for raising this important point. We agree that the theoretical worst case complexity $O((n+d)^3)$ can be large, since updates may involve operations on $(n+d)^2$ matrices. However, in practice, the empirical $\lambda$ and $S$-paths are much shorter since the size of the elbow set $|E|$ is generally much smaller than $n$, and the path contains at most $2(n+1)$ break-points.Following the reviewer’s suggestion, we added an experiment varying the sample size $n$ in Appendix D.1.1 to provide a clear empirical picture of the method’s scaling behavior. Runtime increases from about 12 seconds (n=1000) to 24 seconds (n=2000) and 191 seconds (n=5000), which shows that the computation remains feasible in practice.
>
> **W2: Assumption on $\hat{T}=T$**:
> > We appreciate the reviewer’s concern. We agree that the assumption
> $\hat T=T$ a.s. is extremely strong and is not intended as a practical requirement. Theorem 2 uses this assumption only to study an idealized situation where we could condition on the true latent partition. In this idealized setting, the learned-group conditional guarantee collapses to true-group conditional coverage, if the estimated representation were to perfectly align with this structure. We have revised Theorem 2 to avoid the misunderstanding that the learned-group conditional coverage $\mathbb{P}(Y\in\hat C(X)\mid \hat T(X))$ requires the assumption $\hat T=T$ a.s..
>
> > In practice, however, we never observe $W$ or $T$, so we cannot condition on the true latent groups. All algorithms and experiments in the paper work with the learned representation. Our learned-group conditional guarantee throughout the paper does not assume
> $\hat T=T$ a.s.. Specifically, we define the tilt function on the learned group, $f=\mathbf{1}\set{\hat T(X)=k}$, instead of $f=\mathbf{1}\set{ T(X)=k}$, to reweight the baseline covariate law $P_X$, and to define the target test population for which we calibrate coverage. In this case, the design yields the finite-sample conditional coverage on the learned-group directly. Remarkably, our procedure provides finite-sample learned-group conditional coverage for any mapping $\hat \pi(\cdot)$. The alignment between $\hat T$ and $T$ affects the size and effectiveness of the prediction sets. When $\hat T$ is close to the true partition, the groups are more homogeneous and the resulting sets tend to be narrower and more interpretable. To connect $\mathbb{P}(Y\in\hat C(X)\mid \hat T(X))$ to the latent true-group coverage $\mathbb{P}(Y\in\hat C(X)\mid T(X))$ under the design with $f=\mathbf{1}\set{\hat T(X)=k}$, we still need the assumption $\hat T=T$ a.s. (In the revision, we added this analysis in the Appendix C.3, Corollary 6 and Remark 7).
>
> **W3 & Q2: Comparison of SpeedCP and SplitCP**:
> > We appreciate the reviewer’s observation. We agree that in Figure 2, the performance of SpeedCP sometimes matches to that of SplitCP; this is expected because, in our synthetic setup, the calibration mixture weights are sampled uniformly over the simplex, so the conditional structure is also roughly uniform, and a single global threshold can perform reasonably well in most bins. However, the test distribution is skewed toward one vertex, making the other vertices sparse. In these sparse regions, SplitCP struggles because its global threshold is dominated by the dense interior of the simplex and does not accurately reflect the local score distributions near the sparse-data vertex. Therefore, SplitCP tends to undercover in bins 3 and 5 where we have sparse test and calibration samples. We have added an explanation for this in the revision, in Section 3.
>
> > Importantly, the goal of SpeedCP is not to uniformly outperform SplitCP in all designs, including this synthetic example where the calibration distribution is nearly uniform, but to provide a general, principled framework that (i) correctly calibrates local score distributions even when local calibration data are scarce, and (ii) remains valid under explicit covariate shift through the tilting function $f$. The advantages of SpeedCP become more pronounced in harder, real-data settings such as QM7b in Figure 3, where the conditional structure is significantly more complex and the calibration distribution is far from uniform. In these regimes, SpeedCP yields improved conditional coverage.
>
> **Q1: Explanation of the linear map $\Phi^*$**:
> > We thank the reviewer for pointing this out. $\Phi^\ast$ is a user-specified feature map built on the projection $\hat\pi(\cdot)$, analogous to the map $\Phi$ introduced in Section 1. Specifically, $\Phi^\ast:\mathcal{X}\to\mathbb{R}^d$ encodes the covariates used to parameterize the tilt functions $f$ within the RKHS, and its exact form depends on the setting. In our experiments, $\Phi^*$ is instantiated as an intercept + $K-1$ group dummy variables, which supports both marginal and group-conditional analyses. We have added these details in our revision.

---

### Meta-Review · Area_Chair_QcxX · 2026-01-12

**Summary:**

Some limited algorithmic novelty: Reviewer vevS argued that the \lambda-path and active-set updates closely follow established kernel quantile-regression path-following techniques, and the S-path is a natural extension, with the main novelty being the integration of these tools into the Gibbs-style conditional conformal pipeline. The rebuttal largely accepts this framing and repositions the contribution as an engineering/integration advance rather than a new solution-path method.  Applications of solution-path (homotopy continuation) have been also applied in similar conformal prediction settings.

On the theory side, reviewers yCxV and vevS highlighted that Theorem 2’s latent-structure result relies on an unrealistic oracle alignment assumption (effectively Bayes-optimal recovery), which the authors subsequently reframe as an efficiency limit rather than a practical guarantee; related concerns also remain about whether Theorem 1 provides a usable guarantee versus a decomposition with an uncontrolled gap. Finally, while the authors provide empirical scaling and argue that paths are short in practice, worst-case behavior of solution-path methods can still be unfavorable (potentially many path segments), and the paper does not provide a robust theoretical characterization beyond typical-case arguments. In my knowledge, solution-path usually suffers from exponential worst case complexity. However, as the authors mentioned, in practice the algorithm is quite fast and worst-case appears in quite adversarial setup.

**Reviewer Concerns:**

Beside some writing/clarity issues, here are some point that are (partially standing)

- Path-following/KKT/active-set machinery is viewed as standard kernel quantile-regression solution paths; S-path as a natural extension. Rebuttal reframes novelty as integration into CP, but “incremental” perception may remain.
- Tuning vs validity CV selection of (\lambda,\gamma) on calibration (and pinball-loss tuning) raised concerns about post-selection finite-sample coverage. A tuning split helps, but the revision needs a clear “guaranteed protocol vs heuristic variant” separation.
-  Theorem 1 is seen as a decomposition/identity with an uncontrolled gap, not a bound ensuring small error; Theorem 2 relies on an unrealistic oracle alignment assumption (even if reframed as an oracle limit).
- Computational complexity: Scaling experiments help, but worst-case path complexity remains potentially large; claims should stay “practically fast” rather than implying strong worst-case guarantees. Plus, one could add numerical instabilities
- In some plots SpeedCP is close to SplitCP; benefits need to be clearly demonstrated in regimes where conditional structure/sparsity makes a difference.

**Reviewer Scores:**

- Reviewer yCxV: might increase (their main questions scaling with n, Theorem 2 assumption framing, \Phi^* definition, and SplitCP comparison, were directly addressed). The score could remain neutral
- Reviewer vevS: likely no change (the rebuttal concedes novelty framing and proposes fixes, but the core objections about incremental novelty, tuning/validity, and theory framing would probably remain).
- Reviewer HaCa: no change (the rebuttal responds to the coverage-gap interpretation and robustness/sensitivity questions, which were their main concerns).
- Reviewer qs9h: likely no change (they indicated their questions were satisfactorily answered). However, the score is quite high and not strongly backed in the reviews, so the overall score is inflated.

---

### Decision · Program_Chairs · 2026-01-26

Reject